# Healthy forests safeguard traditional wild meat food systems in Amazonia

André Pinassi Antunes[1,2,3,4,5 ✉], Pedro de Araujo Lima Constantino[1,3], Julia E. Fa[6,7,8], Daniel P. Munari[1,9], Thais Q. Morcatty[1,3,10,11 ✉], Michelle C. M. Jacob[12], Bruce W. Nelson[2], Mariana Franco Cassino[2,13], Elildo A. R. Carvalho[1,5], Amy Ickowitz[7], Lauren Coad[7,14], Richard E. Bodmer[15,16], Pedro Mayor[1,3,4,5,17], Cecile Richard-Hansen[18], João Valsecchi[1,3,11], João V. Campos-Silva[1,2,4], Juarez C. B. Pezzuti[1,19,20], Miguel Aparício[2], Eduardo M. von Muhlen[4], Marcela Alvares Oliveira[1,3,21], Milton J. de Paula[1,22], Natalia C. Pimenta[1,23], Marina A. R. de Mattos Vieira[1,24], Marcelo A. Santos Junior[25,26], André V. Nunes[1,27], Jean P. Boubli[28], Luan M. G. Suruí[29], Eneias C. S. Paumari[30], Abimael V. C. Paumari[30], José Lino V. S. Paumari[30], Germano C. Paumari[30], Ana Paula L. R. Katukina[31], Dzoodzo Baniwa[32], Valencio S. M. Baniwa[32], Walter S. L. Baniwa[32], Abel O. F. Baniwa[32], Armindo B. Baniwa[32], Isaías J. S. Baniwa[32], Yaukuma Waura[33], Jairo Silvestre Apurinã[34], Valdir S. S. Apurinã[34], Josiane O. G. Tikuna[35], Elias P. A. L. Tikuna[35], José L. Kaxinauá[36], Kussugi B. Kuikuro[37], Jorge T. Penaforth Kaixana[38], George H. Rebelo[1,2], Dione Torquato[1,39], Vanessa S. F. Apurinã[34,40], Miguel Antúnez[3,15,41], Pedro E. Perez-Peña[42], Tula G. Fang[15], Pablo E. Puertas[43], Rolando M. Aquino[44], Louise Maranhão[1,11], Guillaume Longin[45], Cíntia K. M. Lopes[46,47] & Hani R. El Bizri[1,3,7,11]

Amazonia is the largest[1] and the most species-rich tropical forest region on Earth[2], where hundreds of Indigenous cultures and thousands of animal species have interacted over millennia[3,4]. Although Amazonia offers a unique context to appraise the value of wildlife as a source of food to millions of rural inhabitants, the diversity, geographic extent, volumes and nutritional value of harvested wild meat are unknown. Here, leveraging a dataset comprising 447,438 animals hunted across 625 rural localities, we estimate an annual extraction of 0.57 Mt of undressed animal biomass across Amazonia, equivalent to 0.34 Mt of edible wild meat. Just 20 out of 174 taxa account for 72% of all animals hunted and 84% of the overall biomass extracted. We show that this amount of wild meat can meet nearly half of protein and iron dietary requirements for rural peoples, along with a substantial portion of their needs for B vitamins (18–126%) and zinc (23%). However, wild meat productivity is likely to have decreased by 67% in nearly 500,000 km² of highly deforested areas of Amazonia. Furthermore, the availability of wild meat per capita decreases significantly in areas with higher human population, greater proximity to cities, and more extensive deforestation. These findings highlight the urgent need to preserve the forest to safeguard biodiversity and traditional wild meat food systems, which will be essential for ensuring Amazonian peoples' well-being and achieving several of the United Nations Sustainable Development Goals[5].

Meat derived from wild animals (hereafter wild meat) is a critical source of dietary protein and income for up to 150 million households across the global south[6], making their well-being inextricably linked to the health of the ecosystems that support wildlife habitats and sustain this vital component of traditional food systems[7,8]. Amazonia, the largest and most biodiverse tropical system on Earth[1,2], is home to almost 11 million rural inhabitants, including Indigenous, traditional and small-scale farming peoples (Extended Data Fig. 1 and Methods). Collectively referred to here as Amazonian peoples, these groups rely on hunting, fishing, foraging, swidden agriculture and animal husbandry[9].

Wild tetrapods—amphibians, reptiles, birds and mammals—along with freshwater fish, constitute vital components of their diets and cultural traditions[10].

Amazonia's unparalleled biocultural diversity provides a unique opportunity to assess wildlife as a critical food source at a near-continental scale. Previous studies have documented hunting practices in specific localities or sub-regions, but their limited geographic scope has hindered understanding of broader spatial patterns and the cumulative impacts of large-scale environmental change. Our dataset—spanning multiple Amazonian cultures and regions over six

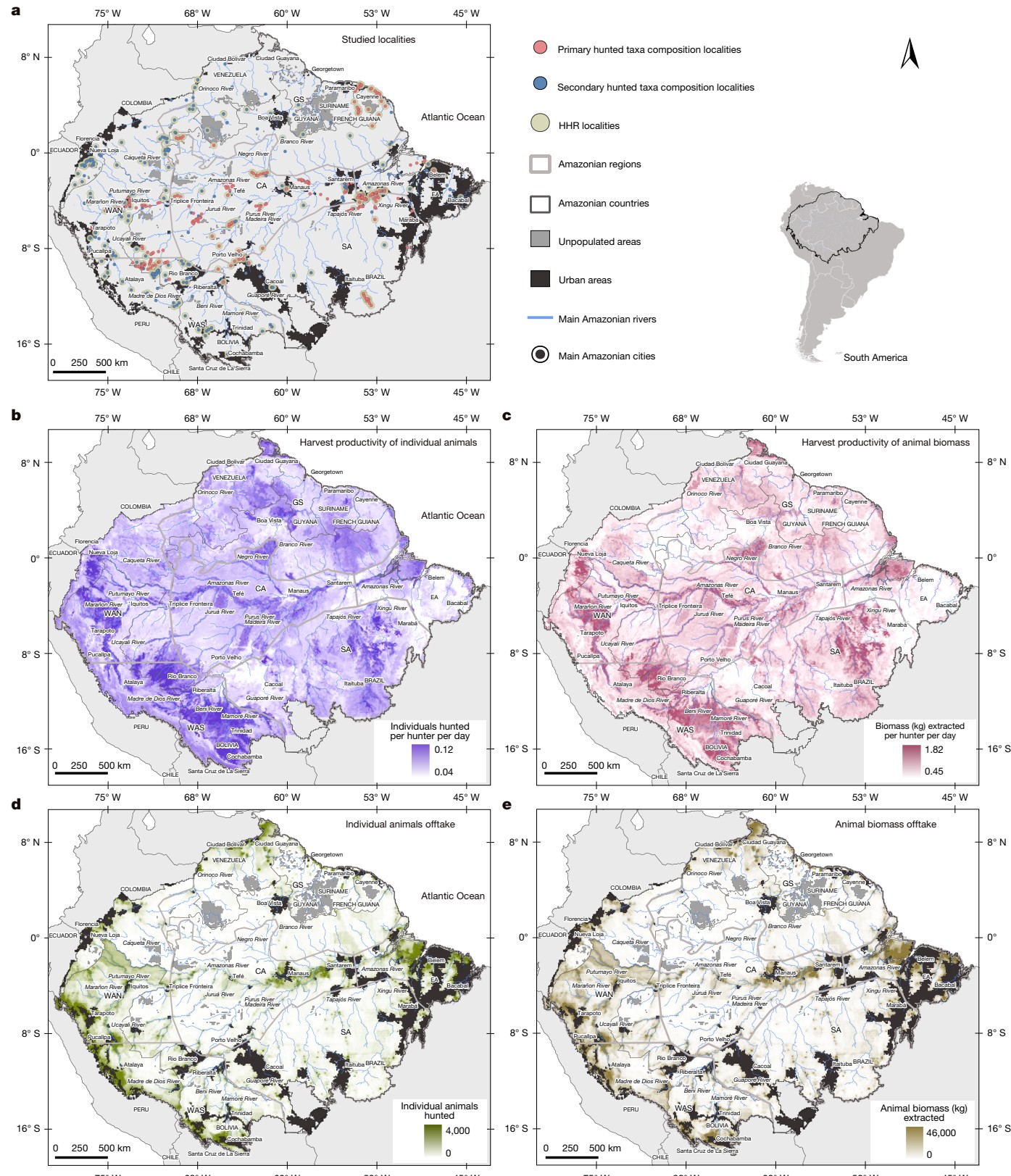

**Fig. 1** | See next page for caption.

decades—offers the first spatially explicit, large-scale analysis of hunted animal diversity and the interplay of environmental and anthropogenic factors influencing wild meat productivity, availability, consumption and nutritional contributions. Results show that deforestation and urbanization can reduce wild meat productivity and simplify the composition of hunted taxa, threatening traditional food systems and the nutritional well-being of Amazonian peoples. Safeguarding Amazonia is thus critical for biodiversity conservation, for maintaining ecosystem functions, and for supporting human health[11], thereby advancing multiple United Nations Sustainable Development Goals (SDGs)[5].

Fig. 1 | **Distribution of studied localities and predicted heat maps of individuals and biomass HP and offtake. a**, Distribution of the 625 studied localities with primary data (342), secondary data (290) and information on HHR (301) and TSOP (590). **b**, Individuals HP—how the overall individual animals HHR (number of individuals hunted per hunter per day) varies with environmental and anthropogenic factors. **c**, Biomass HP—how the overall animal biomass HHR (biomass extracted in kg per hunter per day) varies with environmental and anthropogenic factors. **d**, Individuals offtake—total number of individuals hunted per year. **e**, Biomass offtake—total biomass in kg extracted per year. Individuals and biomass HP predictions represent the potential overall individuals and Biomass HHR of each 10 × 10 km spatial cell, whereas offtake predictions represent the potential catch of individuals and biomass because consider the product between the predicted HP and the number of hunters for each pixel. Amazonian regions: Guiana Shield (GS), northwestern Amazonia (WAN), central Amazonia (CA), southwestern Amazonia (WAS), southern Amazonia (SA) and eastern Amazonia (EA). See Methods for detailed information on the spatial prediction process.

## Quantifying hunting in Amazonia

Our analyses draw on primary and secondary data collected between 1965 and 2024, which documents wild tetrapods (mammals, birds, reptiles and amphibians) hunted by Amazonian peoples. Our Marupiara Dataset includes a total of 447,438 individual hunts recorded across 625 rural localities (Fig. 1a and Methods). From this dataset, we derived two key metrics: (1) hunter harvest rate (HHR), as the average number of animals hunted per hunter per day in each locality; and (2) taxon-specific offtake proportion (TSOP) for 174 hunted taxa (classified into species, genera, family, order or subclass), as the proportion of animals hunted from a specific taxon relative to the total animals hunted in each locality.

We utilized random forest models[12] to spatially predict HHR and TSOP metrics based on 12 environmental and anthropogenic variables, including ecosystem productivity and integrity, environmental conditions, habitat type, topography, human pressure and cultural identity (Methods and Supplementary Methods 1–3). For HHR modelling, we included a variable to account for the hunting registration effort, specifically the number of days over which hunted animals were recorded. This modelling approach allowed us to estimate HHR for all pooled taxa and for each of the 174 hunted taxa individually. In this context, we define harvest productivity (HP) as the potential number of individuals and biomass that can be harvested per hunter across Amazonia, based on how overall HHR for all taxa pooled varies with environmental and anthropogenic factors (Fig. 1b,c). The total number of individual animals (individuals offtake) and biomass (biomass offtake) harvested for all taxa pooled (Fig. 1d,e) and for each taxon separately were spatially predicted by combining the predicted HHR rasters and the derived raster of the number of rural hunters across Amazonia. All metrics and spatial modelling procedures are described in Methods. Our estimates are presented as percentages and mean ± s.d. (0.10–0.90 quantiles).

## Harvest productivity

We estimated that there are 1.93 ± 0.08 (1.83–2.04) million rural hunters in Amazonia (Extended Data Fig. 1 and Methods). Our spatial models suggest that a rural Amazonian hunter harvests on average 0.07 ± 0.02 (0.05–0.10) animals or 0.94 ± 0.40 (0.56–1.45) kg biomass per day, amounting to 24.8 ± 8.2 (17.8–35.5) animals or 345.0 ± 145.1 (204.1–531.1) kg per year.

Random forest spatial models predicted higher HP in forested regions of western Amazonia, along the main course of the Amazon River and its major tributaries, as well as in parts of the Guyana Shield and southern western Amazonia (Fig. 1a,b). These areas are characterized by fertile soils, high primary productivity, low to moderate elevations, well-preserved forests and relative isolation from large urban centres. Higher HP was also predicted in historically inhabited territories managed by the Waorani, Itonama, Movima, Warina, Kandoshi-Shapra and Yurucare Indigenous peoples of western Amazonia (Supplementary Methods 2). Full details on the relationships between HHR and environmental and anthropogenic factors, including variable importance, are provided in the Extended Data Fig. 2 and Supplementary Data 1.

Our predicted HP patterns are consistent with broader Amazonian ecosystem dynamics, reflecting the geological history of the region and the resulting variation in soil fertility, forest productivity and turnover, and species and functional composition. Enriched by Andean erosion, the younger and more fertile soils of western Amazonia and floodplains sustain highly productive forests that allocate proportionally more resources to plant reproductive processes such as flower and fruit production rather than photosynthesis[13]. This enhanced reproductive output supports larger and more diverse animal populations[14,15] (Supplementary Data 2), thereby contributing to the higher HP observed in these regions. Notably, our results highlight the complex interplay among environmental and anthropogenic factors that shape wildlife harvesting patterns across Amazonia, with cultural identity emerging as a key predictor of HP (Extended Data Fig. 2 and Supplementary Data 1).

Although our findings provide robust insights, some limitations warrant consideration. Several studies lack clear descriptions of hunter selection criteria—whether participants were chosen randomly or focused on primary hunters in each community—and often do not specify whether reported data represent all, or only a subset, of hunting activities. Moreover, whereas hunting records span from 1965 to 2024, spatial covariates are not consistently available for the entire period. To address this, we matched HHR and TSOP records to the closest corresponding time frame (Methods) and included a variable to account for variation in recording effort across studies (Methods), which emerged as an important predictor in our HHR models. Despite potential constraints in capturing fine-scale local variability, the breadth of our dataset and the consistency of spatiotemporal covariates provide a solid basis for a comprehensive assessment of wildlife harvesting dynamics across Amazonia.

## Wildlife offtake

Our spatial predictions of individual and biomass offtake reveal that human population density is a major driver of harvest intensity, with the highest numbers of animals being taken in densely populated rural areas surrounding major Amazonian cities (Fig. 1d,e). Although predicted HP was lower in these areas (Fig. 1b,c), large rural populations result in more hunters and thus greater overall harvest pressure. By contrast, sparsely inhabited interfluves show markedly lower offtake levels (Fig. 1d,e), largely reflecting the scarcity or absence of rural settlements and hunters (Extended Data Fig. 1c,d), despite high variability in predicted HP across these landscapes (Fig. 1b,c).

We estimate that 46.4 ± 10.3 (35.7–56.7) million animals are harvested annually across Amazonia, corresponding to 0.57 ± 0.17 (0.41–0.75) Mt of undressed wild meat. These values fall within the range of previous, non-spatially explicit estimates (0.15–1.29 Mt) in Amazonia[16,17].

## Hunted taxa composition and dominance

The diversity of wildlife hunted and consumed in Amazonia includes at least 490 species (175 mammals, 264 birds, 40 reptiles and 11 amphibians), here grouped into 174 taxa (Fig. 2, Extended Data Figs. 3–5 and Supplementary Tables 1 and 2). These hunted taxa span 6 orders of magnitude in body size (Fig. 2, Extended Data Fig. 3, Supplementary Table 2 and Supplementary Data 3) and represent roughly 10% of the approximately 4,788 tetrapods known to occur in Amazonia, based

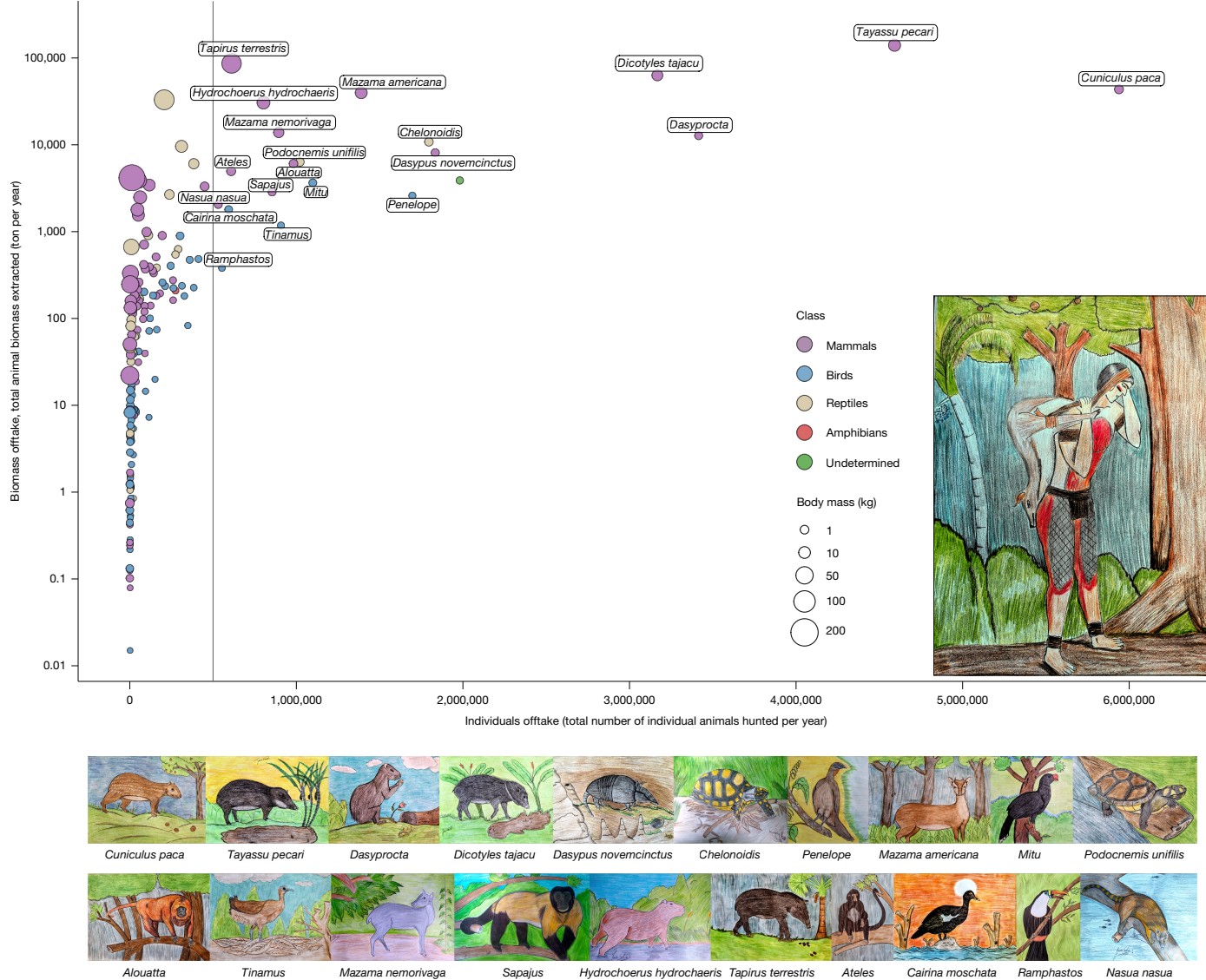

**Fig. 2 | The relationship between the annual offtake of the number of individuals hunted and biomass extracted for 174 animal taxa.** Size of dots is proportional to the average taxon-specific body mass and colours represent animal classes. Animal drawings, made by the Indigenous artist and author Jairo Silvestre Apurinã, represent 20 key hunted taxa in order of the total number of individuals hunted, from top left (highest) to bottom right (lowest). The grey vertical line indicates the threshold of 500,000 animals hunted, which was used to define key dominant hunted taxa.

on data from the International Union for Conservation of Nature and Natural Resources (IUCN)[18].

Mammals were the most hunted group, accounting for 66.5% of all individuals hunted, followed by birds (21.6%), reptiles (11.2%) and amphibians (0.006%). The most harvested groups were ungulates (24.4%) and large rodents (23.7%), followed by primates (8.7%), guans and curassows (8.2%), armadillos (5.6%), aquatic birds (4.9%), river turtles (4.6%), tortoises (4.1%) and terrestrial birds, including tinamous, trumpeters and wood quails (4.1%). Mammals contributed most of the biomass (85.5%), followed by reptiles (11.9%), birds (2.5%) and amphibians (< 0.001%). The highest biomass harvested came from ungulates (61.2%), followed by large rodents (15.5%), caimans (6.5%), primates (3.3%), river turtles (3.3%), armadillos (2.4%), tortoises (1.9%) and cracids (1.4%). These offtake patterns on the composition of hunted animals groups are shown in Extended Data Fig. 6.

Only 20 taxa surpassed the threshold of 500,000 animals hunted annually, yet these accounted for 71.7% of all individual animals hunted and 83.6% of the total biomass extracted (Fig. 2). This group included 14 mammals (1 cingulate, 1 carnivore, 3 rodents, 3 primates and 5 ungulates), 4 birds and 2 chelonians, collectively representing 63 animal species, 15 of which are threatened with extinction according to the IUCN Red List of Threatened Species[18] (Supplementary Table 3). The white-lipped peccary (*Tayassu pecari*) and the tapir (*Tapirus terrestris*) account for about 40% of the total biomass extracted. Harvest patterns for these key hunted taxa are shown in Fig. 3.

The lowland paca (*Cuniculus paca*) was the most hunted species by number of individuals in all sub-regions, generally followed by white-lipped peccary. However, regional differences are marked (Fig. 3 and Supplementary Tables 4 and 5). Ungulates dominate harvests in the Guyana Shield, and western and southeastern Amazonia, whereas large rodents are more targeted in central and eastern Amazonia. Large primates, such as woolly monkeys (*Lagothrix* spp.), spider monkeys (*Ateles* spp.), howler monkeys (*Alouatta* spp.) and capuchin monkeys (*Sapajus* spp.), are more commonly hunted in western Amazonia, whereas armadillos are more frequent in eastern and southeastern Amazonia.

Habitat type also shapes hunting patterns. In flooded forests, Amazonian peoples hunt higher proportions of river turtles, tortoises, caimans, howler monkeys, capybara, manatees and waterfowl (ducks, cormorants and egrets). By contrast, those living in upland terra firme forests, which cover around 86% of Amazonia[19], hunt proportionately

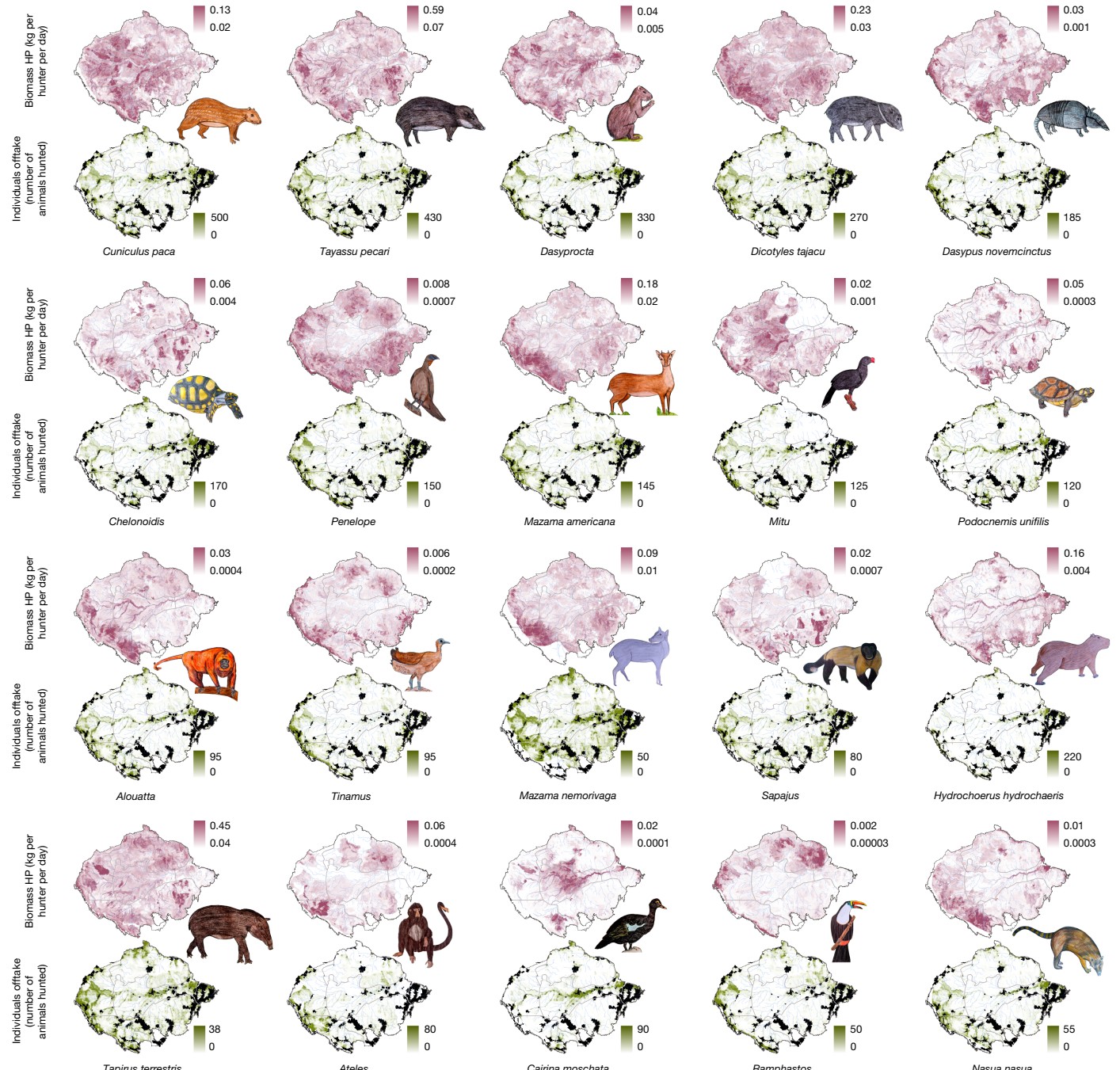

**Fig. 3 | Wild meat extraction heat maps showing predicted animal biomass HP and individual animals offtake across Amazonia of the 20 dominant wild meat taxa.** The 20 dominant hunted taxa were the only ones that surpassed the threshold of 500,000 animals hunted annually. See Methods for detailed information on the spatial prediction process. Animal drawings by Jairo Silvestre Apurinã.

more tinamous, trumpeters, wood quails, guans and armadillos (Supplementary Tables 6 and 7). Many Amazonian peoples have historically settled in transitional zones between flooded and upland areas[20], taking advantage of seasonally complementary resources and adapting management strategies accordingly[21].

Cultural identity exerts a strong influence on both HP and TSOP across taxa. With 511 distinct Indigenous peoples[22] speaking at least 335 languages[23], the remarkable cultural diversity of the Amazonian region reflects millennia of dynamic, reciprocal relationships with nature[3,4]. Hunting preferences illustrate this diversity: most Aruak peoples highly value ungulates (V.S.S.A. and D.B., personal observations), yet those in Xingu River deliberately avoid them (Y.W. and K.B.K., personal observation), whereas those in southwestern Amazonia focus

mostly on primates and large terrestrial birds[24]. Conversely, several Indigenous peoples in the northwestern Amazonia prize the often overlooked slender-legged tree-frogs (*Osteocephalus* spp.). These culturally mediated differences highlight the depth and complexity of hunting dynamics, calling for region-specific investigations and a transdisciplinary approach that moves beyond simple environmental or cultural determinism.

Although our models capture TSOP variation well for key hunted taxa, they are less reliable for underrepresented species (Supplementary Data 4). Hunting practices have also shifted over the past six decades in response to cultural and technological shifts. The widespread adoption of rifles has increased hunting efficiency, particularly for large species[25], and the growing use of flashlights has intensified pressure

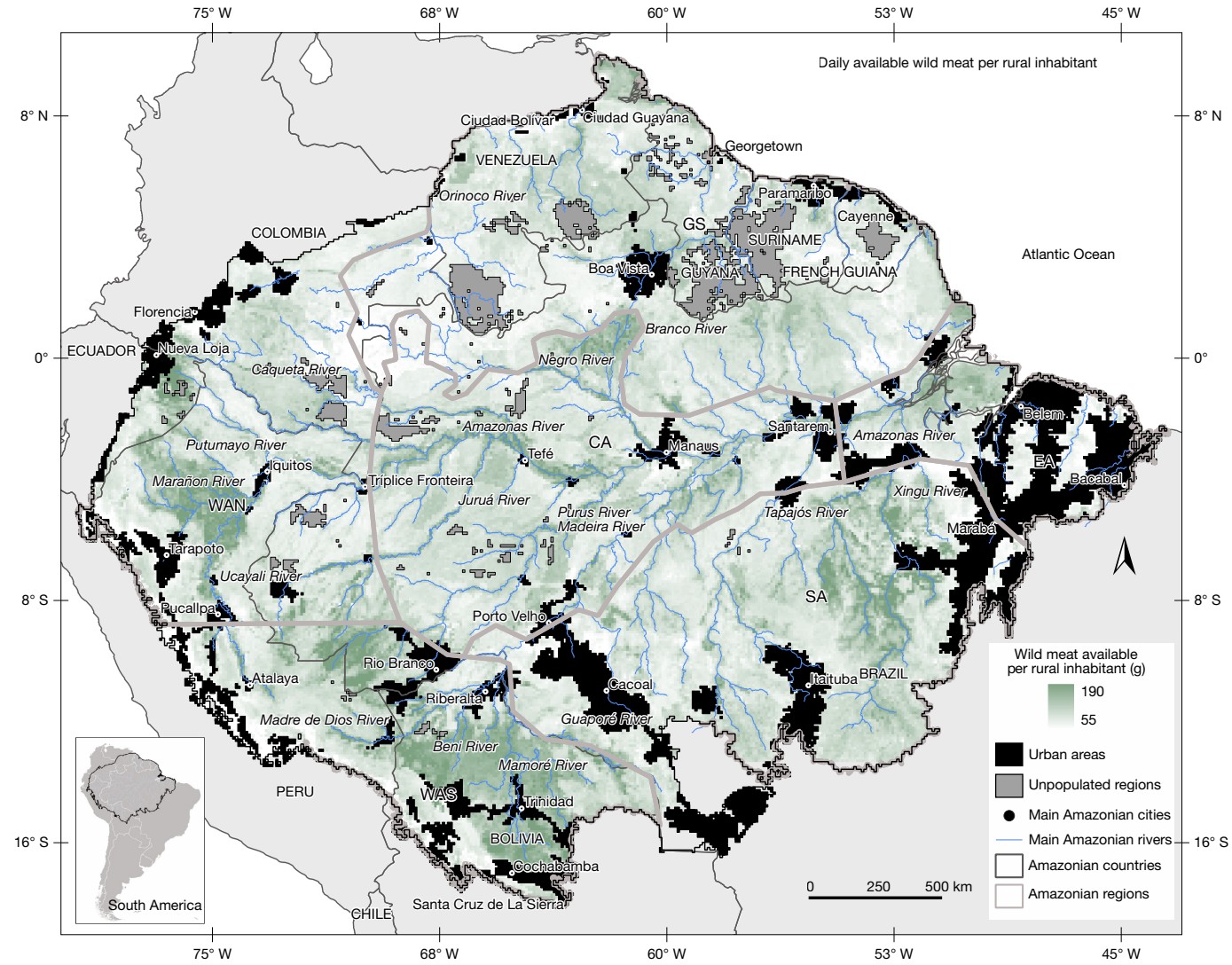

**Fig. 4 | Predicted heat map of the available wild meat per rural inhabitant across Amazonia.** The spatial distribution of available wild meat per rural inhabitant was derived by multiplying the predicted raster of daily animal biomass offtake by 0.585—the proportion of edible wild meat relative to undressed biomass—to obtain daily edible wild meat production across Amazonia, and subsequently dividing it by the APPS raster, which spatially explicit the number of rural inhabitants per spatial cell.

on nocturnal animals such as the paca[26], now the most hunted species in Amazonia. Furthermore our models do not incorporate recently documented natural population cycles of white-lipped peccaries[27], which are likely to influence hunting patterns given this species' central importance to Amazonian hunters.

## Overall wild meat production

We estimated that the annual edible wild meat production in Amazonia amounts to 0.34 ± 0.09 (0.24–0.44) Mt, which represents 58.5% of the total undressed biomass (Methods). As much as 86.4% of the wild meat produced in Amazonia—equivalent to 0.30 ± 0.08 (0.22–0.38) Mt—is derived from the key 20 taxa.

The annual monetary value of this wild meat production is approximately US$2.2 ± 0.6 (1.6–2.8) billion, based on 2024 beef market prices. Accurately assessing the economic value of wild meat remains challenging owing to the often informal or illegal nature of the trade in most of Amazonia. However, this hidden economic value of wild meat production suggests that traditional hunting is a significant ecosystem service for Amazonian peoples, providing affordable, high-quality nutrition and reducing meat expenditures.

## Nutritional value of wild meat

We estimated that, on average, 101.0 ± 43.6 (59.1–157.5) g of edible wild meat are available per rural inhabitant per day across Amazonia (Fig. 4), equating to 36.8 ± 15.9 (21.6–57.5) kg per person per year. This estimate falls within the range of results from previous research in the region[16], which found per capita wild meat availability ranging from 21 to 191.6 (kg per person per year).

The level of available wild meat per rural inhabitant estimated here has the potential to meet the dietary reference intakes (DRIs) for vitamin B₁₂ across much of Amazonia, while also significantly contributing to DRIs for protein, iron, zinc and other essential B vitamins and minerals (Table 1). The irreplaceable nutritional value of wild meat is particularly critical for 10.87 million rural inhabitants in Amazonia, providing a highly bioavailable source of protein, all essential amino acids, and vital micronutrients, which are often less accessible in plant-based foods[28,29]. Higher wild meat availability was linked to better health among children not only in the Amazon[30] but also in the Congo Basin[31], including higher haemoglobin levels in children and increased household iron and zinc intake[32]. This is particularly noteworthy in regions where micronutrient deficiencies

**Table 1 | Estimated percentages of daily dietary requirements for energy, macronutrients and micronutrients provided by wild meat in Amazonia**

| Macronutrients and energy | Dietary requirements furnished by wild meat (%) |
|---|---|
| Protein | 50.7±13.6 (36.5–66.6) |
| Total fat | 5.6±1.5 (3.3–8.7) |
| Calories | 5.7±1.5 (3.9–7.7) |
| **Micronutrients** | **Dietary requirements furnished by wild meat (%)** |
| Vitamin B$_{12}$ | 126.4±33.8 (59.6–218.7) |
| Iron | 39.2±10.5 (22.7–61.1) |
| Vitamin B$_2$ | 36.8±9.8 (23.9–52.7) |
| Vitamin B$_3$ | 33.5±9.0 (20.4–50.5) |
| Selenium | 31.6±8.5 (22.9–41.4) |
| Zinc | 23.3±6.2 (11.5–39.2) |
| Vitamin B$_1$ | 18.1±4.8 (10.5–28.4) |

The percentages of the daily dietary requirements of energy, macronutrients and micronutrients furnished by wild meat to Amazonian peoples are correlated to the predicted values of the average availability of edible wild meat per rural inhabitant per 10 × 10 km spatial cell. See Methods for detailed information on the spatial prediction process.

are widespread, compounded by malaria, intestinal parasites and genetic disorders[33].

## Threats to wild meat food systems

Our spatial analyses revealed the crucial effect of ecosystem integrity on HP in Amazonia. In regions with over 70% deforestation, covering 0.80 M km², we observed a 74.7% decline in the number of individual animals harvested per hunter and a 67.3% reduction in harvested biomass per hunter (Extended Data Fig. 7).

Available wild meat per rural inhabitant was significantly lower in regions with: (1) higher numbers of rural inhabitants; (2) closer proximity to urban centres; and (3) greater deforestation levels (Extended Data Fig. 8). In these more degraded areas, our spatial estimates of TSOP indicate a shift in hunting patterns, with ecological generalists such as the nine-banded armadillo (*Dasypus novemcinctus*), capybara (*Hydrochoerus hydrochaeris*), guans (Cracidae, *Penelope* spp.) and pigeons (Columbidae) becoming proportionally more hunted than in better-conserved forests (Supplementary Table 8). By contrast, large atelid primates such as the woolly, spider and howler monkeys, which are quite vulnerable to the synergistic effects of deforestation and overhunting, are much less hunted in degraded areas (Supplementary Table 8).

Large-scale agriculture, land grabbing, logging, mining, infrastructure development and urbanization have led to deforestation[34], increasingly undermining Amazonian peoples' reliance on biodiversity. The combined effects of deforestation and wildlife overharvesting have produced simplified animal assemblages, with lower species richness and fewer large-bodied species[35]. These pressures are further compounded by recent climatic changes, including more frequent floods, droughts and large-scale fires[34], all of which threaten both habitats and many key hunted taxa[36]. We recommend that further studies investigate the future impacts of deforestation, wildfires and climate change on animal populations, the supply of wild meat and the territorial dynamics of hunting grounds.

We also predicted higher offtake and lower HP near urban centres, raising concerns about the sustainability of hunting in these areas. High meat demand in densely populated peri-urban areas, coupled with declining wildlife populations[37], may shift rural diets toward cheaper domestic meats like chicken[38], which generally contains four times less iron, two times less B vitamins, and lower levels of protein and zinc than wild meat[39].

Owing to its high cost and the significant logistical challenges of production, transport and storage in remote areas, beef is rarely consumed across most of Amazonia[38]. Paradoxically, cattle ranching remains the leading driver of deforestation in the region, and contributes to the loss of approximately 0.63 M km² of forest since 1978, primarily to supply domestic meat markets[40]. Replacing the estimated edible wild meat production of 0.34 Mt with beef, based on current cattle ranching yields in Amazonian traditional pastures[41] (0.2–0.8 kg ha$^{-1}$ day$^{-1}$), would require converting 7,603–63,803 km² of forest into pasture. According to previous estimates of emissions of approximately 18,000 tonnes of carbon dioxide per square kilometre of deforested Amazonia[42], this conversion would release between 140 and 1,160 Mt of CO2−equivalent to up to around 3% of global annual emissions. Even with this substantial environmental and climatic costs, domestic meat production still does not ensure equitable access for rural populations in remote areas.

## Managing wild meat food systems

Our study highlights the essential role of traditional hunting and wild meat access in advancing several SDGs of The United Nations Convention on Biological Diversity, supporting nutritional security and health, helping reduce malnutrition, strengthening traditional food systems and promoting sustainable wildlife use and ecosystem conservation across Amazonia (Supplementary Discussion 1). Although the sustainability of hunting in tropical forests and the risks of zoonotic diseases potentially linked to hunting have been intensively debated, we focus on understanding Indigenous and traditional hunting practices and wild meat access within the broader context of achieving the SDGs in Amazonia. We demonstrate that the health of the Amazon Forest is vital to sustaining traditional wild meat food systems and the nutrition of Amazonian peoples. From this evidence, we contend that illegitimate proposals to ban, restrict or replace wild meat without acknowledging its cultural and nutritional significance reflect a colonial mindset that undermines the autonomy and traditional food systems of Amazonian peoples.

We provide new insights into the complex interplay of environmental, cultural and human pressure factors shaping wildlife harvest patterns throughout Amazonia. The finer details of these interactions merit further locally focused research to avoid determinisms. Whereas the SDGs and the IUCN Red List of Threatened Species emphasize global policies and actions, we stress—echoing E. Ostrom[43]—that wildlife management initiatives must be tailored to local ecological, cultural, socioeconomic and vulnerability nuances, ideally focused on local and regional key hunted taxa and led by Amazonian peoples. Wildlife management initiatives shaped by Amazonian peoples' demands and cultural practices are more legitimate and more likely to remain viable long-term. Protected areas managed by Indigenous and traditional peoples maintain healthy populations of key hunted species[44], even those once commercially over-harvested[45]. Indigenous and traditional knowledge have an important role in determining the conservation status of animal populations[46] and understanding key ecological parameters[47], including their densities[48], reproductive rates[49] and population dynamics[27]. These factors, along with deforestation[35], commercial hunting pressures[50], climate change impacts[36] and HP, as shown in this research, strongly influence sustainable harvest potentials. Our predictive heat maps offer valuable spatial insights into HP, human–wildlife interactions and human nutrition in Amazonia, supporting more effective policies integrating conservation and public health.

There is evidence that demonstrates that the sustainability of traditional hunting in Amazonia has historically relied on the dispersed pattern of human occupation, low population density and limited spread of hunters across vast, conserved forests[50]. This mechanism guarantees the preservation of large spatial refuges for terrestrial species, in contrast to aquatic species[50], which often require spatial zoning

and local agreements to maintain healthy populations[51]. Although well-preserved forests combined with strong local governance can sustain harvests through source–sink hunting dynamics[45], our findings indicate that forest degradation increasingly threatens wild meat productivity, alters hunted species composition and erodes unique traditional food systems. In this context, establishing locally agreed hunting management frameworks may be essential for safeguarding the well-being of Amazonian peoples, ensuring the long-term sustainability of the wildlife on which they depend and reducing the risk of social conflict over hunting grounds.

Amazonian systems of knowledge and practice are grounded in ontologies that attribute agency, personhood and humanity to multiple beings, including animals[52], a perspective that must be taken seriously in ecological assessments of hunting and management systems[53]. For Amazonian peoples, relationships with wildlife are framed not as resource extraction but as social reciprocity governed by norms and ethical obligations[54]. Dietary restrictions and spatial avoidance act as sophisticated wildlife management mechanisms akin to species-specific protections, spatial zoning rules, and hunting bans[55]. These culturally embedded practices are effective tools in regulating wildlife harvests and represent the most legitimate management and conservation strategies in Amazonia[56]. Hunting is deeply intertwined with territoriality, as the mobility of hunters continuously redefines space through the creation and maintenance of path networks[57]. Wild meat therefore constitutes a vital food source and a social cornerstone, motivating Amazonian peoples to safeguard their territories, and in doing so, contribute to forest and biodiversity conservation. The health of Amazonian ecosystems and wildlife is inextricably linked to the well-being of Amazonian peoples, underscoring the importance of recognizing their land rights and supporting policies that enhance their autonomy and governance over their territories and biodiversity.

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

[1]Rede de Pesquisa em Conservação, Uso e Manejo da Fauna da Amazônia (RedeFauna), Manaus, Brazil. [2]Instituto Nacional de Pesquisas da Amazônia (INPA), Manaus, Brazil. [3]Comunidad de Manejo de Fauna Silvestre en América, Iquitos, Peru. [4]Instituto Juruá, Manaus, Brazil. [5]Centro Nacional de Pesquisa e Conservação de Mamíferos Carnívoros, Instituto Chico Mendes de Conservação da Biodiversidade, Atibaia, Brazil. [6]Department of Natural Sciences, Faculty of Science and Engineering, Manchester Metropolitan University, Manchester, UK. [7]Center for International Forestry Research, World Agroforestry (CIFOR-ICRAF), Bogor, Indonesia. [8]Natural Sciences and Environment Hub, University of Gibraltar, Gibraltar, Gibraltar. [9]Wildlife Conservation Society Brazil Program, Manaus, Brazil. [10]Department of Geography, University College London, London, UK. [11]Instituto de Desenvolvimento Sustentável Mamirauá, Tefé, Brazil. [12]Laboratory of Biodiversity and Nutrition, Federal University of Rio Grande do Norte, Natal, Brazil. [13]Núcleo de Estudos da Amazônia Indígena (NEAI), Universidade Federal do Amazonas (UFAM), Manaus, Brazil. [14]ICCS, University of Oxford, Oxford, UK. [15]Museo de Culturas Indígenas Amazónicas, Fundamazonia, Iquitos, Peru. [16]Durrell Institute of Conservation and Ecology (DICE), University of Kent, Canterbury, Kent, UK. [17]Departament de Sanitat i Anatomia Animals, Universitat Autonoma de Barcelona, Barcelona, Spain. [18]OFB/DRAS/SEE, UMR Ecofog, Campus Agronomique Kourou, Kourou, French Guiana. [19]Núcleo de Altos Estudos Amazônicos, Universidade Federal do Pará, Belém, Brazil. [20]Departamento de Vertebrados, Museu Nacional do Rio de Janeiro, Rio de Janeiro, Brazil. [21]Centro Universitário Aparício Carvalho, Porto Velho, Brazil. [22]Instituto Federal de Educação, Ciência e Tecnologia do Pará, Altamira, Brazil. [23]Instituto Socioambiental, Manaus, Brazil. [24]Universidade Estadual de Campinas, Campinas, Brazil. [25]Fundação Vitória Amazônica (FVA), Manaus, Brazil. [26]Instituto Nacional de Pesquisas Espaciais (INPE), São José dos Campos, Brazil. [27]Instituto de Pesquisas Ecológicas (IPE), Campo Grande, Brazil. [28]School of Science Engineering and the Environment, University of Salford, Salford, UK. [29]Paiter Suruí Indigenous People, Cacoal, Brazil. [30]Paumari Indigenous People, Tapauá, Brazil. [31]Katukina Indigenous People, Tapauá, Brazil. [32]Baniwa Indigenous People, São Gabriel da Cachoeira, Brazil. [33]Waura Indigenous People, Gaúcha do Norte, Brazil. [34]Apurinã Indigenous People, Lábrea, Brazil. [35]Tikuna Indigenous People, São Paulo de Olivença, Brazil. [36]Huni Kuin Indigenous People, Tarauacá, Brazil. [37]Kuikuro Indigenous People, Gaúcha do Norte, Brazil. [38]Fundação Nacional do Povos Indígenas, São Paulo de Olivença, Brazil. [39]Conselho Nacional das Populações Extrativistas (CNS), Brasília, Brazil. [40]Coordenação das Organizações Indígenas da Amazônia Brasileira (COIAB), Manaus, Brazil. [41]Instituto del Bien Común, Paisaje Putumayo Amazonas, Loreto, Perú. [42]Instituto de Investigaciones de la Amazonía Peruana (IIAP), Loreto, Perú. [43]Asociación Centro de Innovación Científica Amazónica (CINCIA), Loreto, Perú. [44]Instituto de Ciencias Biológicas, Facultad de Ciencias Biológicas, Universidad Nacional Mayor de San Marcos, San Marcos, Perú. [45]Parc Amazonien de Guyane, Remire-Montjoly, French Guiana. [46]Programa de Pós-Graduação em Zoologia, Universidade Estadual do Pará, Belém, Brazil. [47]Museu Paraense Emílio Goeldi, Belém, Brazil. [✉]e-mail: aapardalis@gmail.com; t.morcatty@ucl.ac.uk

## Methods

### The Marupiara dataset

The name Marupiara is derived from a specific epithet in the Indigenous Tupi language and is traditionally associated with the figure of the good or virtuous hunter. We compiled data from primary and secondary sources on wild tetrapods (mammals, birds, reptiles and amphibians) that are hunted for food by Indigenous, traditional, and small-scale farming peoples in all nine Amazonian countries. Primary data were contributed by researchers involved in 12 long-term and short-term studies conducted in 342 communities in Brazil, Peru and French Guyana between 1991 and 2024 (Supplementary Methods 4). Secondary data were obtained from 203 peer-reviewed articles, technical reports, postgraduate dissertations and theses reporting hunting studies in 290 communities in all nine Amazonian countries between 1965 and 2021 (Supplementary Methods 5).

The composition of hunted taxa, documenting the number of individual animals hunted per taxon, year and locality, which was used to model the TSOP, was recorded in 590 localities. Whenever available, we also included data on hunting effort—specifically the number of hunters surveyed and the number of recording days—which was used to model the HHR. Hunting effort data were available for 301 georeferenced localities.

We also compiled information on the number of hunters and consumers in each locality and year to calculate the hunters-to-consumers ratio, which we used to estimate the number of rural hunters in Amazonia based on regional population figures. Each locality was georeferenced, and we recorded the cultural identity of hunter communities where available. We also collected data on the biomass of animals hunted. Further details on the derivation and application of these metrics are provided below.

The final Marupiara dataset comprises 21,397 records of hunted taxa composition, representing 447,438 individual animals hunted across 625 georeferenced communities in rural Amazonia from 1965 to 2024.

### Hunting monitoring schemes

Primary data obtained through hunting monitoring schemes were generally collected under the supervision of researchers and Indigenous, traditional and small-scale farming trained researchers. Three main data collection methods were utilized:

(1) Written surveys. Standardized forms were used to record detailed information on hunted taxa, the number of individual animals hunted, and the estimated biomass per taxon.
(2) Face-to-face interviews: In some communities, structured oral questionnaires were conducted with hunters. Trained interviewers asked hunters about their hunting activities and recorded their responses on paper or digital devices. These questionnaires captured the same information as the written surveys.
(3) Direct monitoring: In select cases, researchers accompanied hunters during their activities, recording real-time observations on species composition and offtake.

All data collection efforts were meticulously documented to ensure consistency, accuracy and comparability across communities and time periods. The approach was designed to be both culturally respectful—honouring the knowledge systems and practices of participating communities—and scientifically rigorous, thereby ensuring the reliability and integrity of the data compiled in the Marupiara dataset.

### Data selection and validation

Given the diversity of secondary data sources spanning nearly six decades, our dataset naturally varied in objectives, methodological rigour, reporting standards and cultural contexts. To ensure consistency, we focused on extracting comparable information across all studies.

We implemented a multi-step validation process to mitigate potential inaccuracies. Primary data served as a baseline for evaluating datasets and assessing methodological consistency, identifying discrepancies and evaluating the reliability of secondary sources. We prioritized studies that provided clear methodological descriptions, reproducible metrics or supplemental documentation and contacted the original authors or institutions when clarification was needed regarding sampling design and local conditions.

We scrutinized outliers and apparent inconsistencies by cross-referencing with more recent peer-reviewed sources or primary datasets from similar regions or time periods. When discrepancies were identified, we assessed whether they reflected genuine cultural differences, shifting hunting practices or methodological shortcomings. Records showing implausible biological or cultural values were excluded.

Throughout the validation process, we remained attentive to cultural and practical factors—such as hunting laws, local traditions and wildlife management strategies—which vary widely across regions and over time. To address this, we consulted local experts and researchers familiar with such nuances to confirm that data collection methods were culturally appropriate and to verify that the underlying assumptions of each dataset remained valid. Only datasets meeting our standards for scientific rigour and comparability were integrated into the analysis and annotated with metadata. We also excluded studies that reported less than four hunted taxa, as these offered limited insights into species composition. Records lacking clearly described or reliable hunting effort methodologies were removed from the final dataset.

### Taxonomic reclassification

We extracted the list of mammal, bird, reptile and amphibian species for the 625 localities from the IUCN spatial database[18], assuming that the taxonomic identity of all recorded hunted species aligns with the taxonomic and geographic distribution currently recognized by the IUCN[18]. We then reviewed all 1,789 original taxa from the Marupiara dataset to correct potential misclassifications, including outdated taxonomy, vernacular identifications, misidentifications or typographical errors. Taxonomic entries were updated to the most refined, species-specific refined classifications, with taxa categorized into species wherever possible.

Following this review, we identified 438 species, 51 higher taxa and an 'undetermined' category, resulting in at least 490 distinct species (Supplementary Table 1). The 'undetermined' category includes unidentified species or aggregations for which TSOP could not be reliably calculated. Higher taxa represent broader taxonomic groupings retained from the original sources, such as *Mazama* spp., which includes lower taxa like *Mazama americana*, *Mazama nemorivaga* and *Mazama gouazoubira*, which are already listed separately in the Marupiara dataset.

All allopatric species were aggregated at the genus level to improve analytical coherence, and taxa with very low sample sizes were grouped into broader categories at the family, order, or subclass level. This classification process resulted in 173 analytically focused hunted taxa, plus the 'undetermined' category, which were used in TSOP and related analyses (Supplementary Tables 1 and 2).

### Taxon-specific body mass and density

We compiled 4,019 observations on body mass for 477 animal species, representing all 173 hunted taxa. These data were sourced directly from the Marupiara dataset, from primary data and from the scientific literature. Based on these records, we estimated the average body mass for each of the 173 taxa (Fig. 2, Extended Data Fig. 3, Supplementary Table 2 and Supplementary Data 3). Additionally, we collected from published sources a total of 2,024 observations on population density for 330 hunted animal species, representing 139 of the 173 hunted taxa. For these 139 taxa, we estimated the average number of individuals per 100 km² (Supplementary Table 2 and Supplementary Data 5). Details on

the estimation procedures for both body mass and population density are provided in the Metrics Estimation section of the Methods.

## Spatial modelling and raster manipulation

All spatial analyses were performed at $10 \times 10$ km raster resolution and WGS (World Geodetic System) 84 Datum using the terra[58] and random-Forest[59] packages in R software[60]. Maps were produced in QGIS (https://qgis.org/en/site/), with the final edition in Inkscape (https://inkscape.org) and GIMP (https://www.gimp.org). We spatially predicted the 10%, 25%, 50%, 75% and 90% quantiles for all the spatial variables and calculated the mean, standard deviation and 90% confidence intervals from the pixel values of this five rasters stack.

Although relatively fine-grained given the vast extent of the Amazon biome, the $10 \times 10$ km resolution may still be insufficient to accurately capture fine-scale heterogeneity in habitat conditions, species distributions, and conservation needs. However, this scale provides a necessary balance between regional coverage and data availability, guaranteeing compatibility with widely used ecological and environmental datasets while allowing for broad-scale assessments that inform conservation planning. A finer resolution would significantly increase computational demands and face data limitations, particularly across such an extensive and heterogeneous region. The $10 \times 10$ km scale remains effective for identifying broader patterns, and its outputs can be complemented in the future by higher-resolution studies or downscaled using local ecological data to refine site-specific conservation actions.

## Amazonian geographical boundaries

We defined the geographical boundaries of Amazonia according to the Amazon Network of Georeferenced Socio-Environmental Information (RAISG), which combines the Amazon biome, its river basins, and relevant administrative regions[1]. We rasterized this polygon at a resolution of 0.0083, resulting in a total area of 8,179,389 km$^2$.

Amazonian regions are classified into: (1) Guiana Shield (GS); (2) northern part of western Amazonia (WAN); (3) central Amazonia (CA); (4) southern part of western Amazonia (WAS); (5) southern Amazonia (SA); and (6) eastern Amazonia (EA) (Fig. 1).

## Amazonian peoples and rural hunters

To spatially determine rural Amazonia, we included only regions 2–3 h away from a small city or town (100,000–250,000 urban inhabitants) as defined in the urban–rural catchment area raster[61] (Extended Data Fig. 1). Amazonian peoples were defined as all individuals living in rural Amazonia, frequently self-declared as Indigenous, traditional or small-scale farming peoples. To generate the Amazonian peoples population size (APPS) raster, we first removed all urban areas[61] from the raster of the total human population (people per pixel for 5 time points between 2000 and 2020 at 5-year intervals)[62] within Amazonia boundaries (Extended Data Fig. 1). This spatial filtering resulted in an estimated rural population of 10,870,022 individuals (APPS), corresponding to 34.2% of the 31,783,941 individuals living (total human population size) in Amazonia.

To obtain the rural hunters population size (RHPS) raster, we multiplied the APPS raster by 0.178 (median) and by 0.168, 0.173, 0.183 and 0.187 (the 0.10, 0.25, 0.75 and 0.90 quantiles), which represent the estimated proportion of hunters to consumers based on data from 72 localities of the Marupiara dataset with both information of number of hunters and number of consumers (Extended Data Fig. 1). See 'Metrics estimation' details on estimation. Based on these calculations, the total number of rural hunters in Amazonia was estimated at $1.93 \pm 0.08$ (1.83–2.04) million.

Wild meat trade and urban consumption data were excluded to maintain a focus on rural hunting practices and their ecological and nutritional implications. Including trade and urban consumption would have introduced complexities related to market dynamics, transportation

and intermediate processing, which are outside the scope of this study's traditional-focused approach. Additionally, reliable trade and urban consumption data are often sparse or inconsistent in Amazonia, making integrating them into the models challenging without introducing significant uncertainty. Although the spatial exclusion of urban areas may inadvertently omit individuals who rely on wild meat as part of their diet, the inclusion of peri-urban populations in our dataset allows for a more reliable assessment of the effects of urbanization on HP.

## Spatial variables

To analyse the factors shaping wild meat harvest in Amazonia, we selected the following set of environmental and anthropogenic spatial variables (Supplementary Methods 1–3):

**Enhanced vegetation index.** A dimensionless index that describes the difference between near-infrared and red reflectance of vegetation cover, normalized by their sum, corrected for some atmospheric conditions and canopy background noise, that can be used to estimate the density of green cover on an area of land, especially in areas with dense vegetation[63]. We included values for enhanced vegetation index (EVI) from 2000, 2005, 2010, 2015 and 2020.

**Annual gross primary productivity.** A MODIS-Terra digital database expressed in kgC m$^{-2}$ year$^{-1}$ (ref. 64) represents the total amount of carbon compounds produced by the photosynthesis of plants in an ecosystem over a given period of time[65]. We included values for annual gross primary productivity (GPP) in 2000, 2005, 2010, 2015 and 2020.

**Annual net primary productivity.** A MODIS-Terra digital database expressed in kgC m$^{-2}$ year$^{-1}$ (ref. 64) represents the carbon uptake plants retain in an ecosystem after accounting for plant respiration (net increase in biomass)[65]. We included values for annual net primary productivity (NPP) in 2000, 2005, 2010, 2015 and 2020.

**Soil fertility.** A digital database covering Amazonia, represented as the sum of exchangeable base cation concentration[66].

**Proportion of flooded areas.** A digital database of the proportion of flooded areas in the Amazon region to upland forests, built on the raster manipulation and combination of the products provided by Hess et al.[67] and Lehner & Dohl[68].

**Elevation.** A digital topographic database scaled in metres based on the NASA Shuttle Radar Topography Mission[69].

**Height above the nearest drainage.** A digital terrain model normalized to the elevation in metres of the drainage network[70].

**Historical distribution of Indigenous family languages.** A categorical digital database of the distribution of the Indigenous territories during the early accounts in Amazonia, built from the combination of the maps provided by Loukotka[71] and Eriksen[72] and classified into their respective Indigenous family language (Supplementary Methods 2).

**Current distribution of Indigenous and non-Indigenous peoples.** A categorical digital database built from the RAISG database of the distribution of the Indigenous lands[73], classified into Indigenous peoples (regions inside Indigenous lands) and non-Indigenous peoples (regions outside Indigenous lands). The current peoples' cultural identity (Indigenous or non-Indigenous) was compiled in the Marupiara dataset from primary and secondary studies.

**Current distribution of family languages.** A categorical digital database built from the RAISG database of the distribution of the Indigenous

lands[73], classified into their respective Indigenous family languages (Supplementary Methods 3). Regions outside the Indigenous Lands are tentatively classified as Latin or German languages. The cultural identity of the current people was compiled in the Marupiara dataset from primary and secondary studies.

**Proportion of habitat loss.** A digital database built on the raster manipulation of annual land cover mapping provided by the MapBiomas[74]. We included measures for the Proportion of habitat loss in 1985, 1990, 1995, 2000, 2005, 2010, 2015 and 2020. We manipulated the MapBiomas land use rasters to obtain rasters with the proportion of habitat loss.

**Urban–rural catchment areas.** Urban–rural catchment areas (URCA) is a digital database of the 30 urban–rural catchment areas showing the catchment areas around cities and towns of different sizes, in which each rural pixel is assigned to a defined travel time category[61].

**Hunting recording time span.** A variable that controls the effort to record hunting in each study, that is, the time range in days in which hunted animals were recorded. This metric was only used to model HHR. We included this variable since we assumed that different time spans of hunting surveys could have different accuracies for data on the animals hunted. We obtained measures of the hunting recording time span from the Marupiara dataset.

## Raster manipulation and processing
We reprojected all spatial variables' rasters to WGS 84 Datum and 10-km resolution, cropped them to the geographical boundaries of Amazonia[1], and replaced missing pixels with the interpolated value from the neighbour pixel. However, even performing this gap-filling technique reduced the original spatial inference from 81,790 to 80,459 10 × 10 km cells (Extended Data Fig. 1).

We extracted the values of all these spatial variables for the 301 georeferenced localities with HHR measures and 590 with TSOP measures (Fig. 1a).

## Temporal variation of spatial variables
None of the digital spatial variables fully cover the 1965–2024 period of the Marupiara dataset, preventing a year-by-year evaluation of their spatial and temporal effects on HHR and TSOP. To address this, we matched HHR and TSOP records to the closest available values of spatial variables in years where spatiotemporal data were available. This approach was feasible for modelling the effects of EVI, NPP and GPP on HHR and TSOP by incorporating measurements at five-year intervals. We obtained proportion of habitat loss data for 1985, 1990, 1995, 2000, 2005, 2010, 2015 and 2020. For EVI, NPP, and GPP, data were available from 2000 onwards at the same five-year intervals. HHR and TSOP values from 1965 to 1987 were assigned the 1985 habitat loss data; from 1988 to 1992, the 1990 data; from 1993 to 1998, the 1995 data, and so on, with post-2018 records assigned the 2020 data. A similar process was applied to EVI, NPP, and GPP, though for a shorter period: records from 1965 to 2002 were matched to the 2000 values, continuing at five-year intervals up to 2020. Despite the Marupiara dataset spanning 1965–2024, most hunting studies occurred around 2006 ± 9 years (90% quantiles: 1995–2017), aligning well with the available temporal coverage of the spatial variables.

## Overall individual animals HHR and overall individual animals HP
After cropping the Marupiara dataset to the geographical boundaries of Amazonia, we obtained georeferenced observations with information on the number of individual animals hunted per hunter per day—overall individual animals HHR. We extracted the values of all spatial variables for the 301 georeferenced localities (both from primary and secondary data) with HHR measures. We determined the overall individual animals HHR as the total number of animals hunted of all taxa in each locality

divided by the sum of hunters accountable for catching those animals during the monitored period in days. This includes days when hunters neither went hunting nor harvested any animals.

Firstly, we spatially predicted the overall individual animals HP by performing random forest models, where the overall individual animals HHR across Amazonia was a function of the enhanced vegetation index, gross primary productivity, net primary productivity, net primary productivity quality control, soil fertility, proportion of flooded areas, elevation, height above the nearest drainage, historical distribution of Indigenous family languages, current distribution of Indigenous and non-Indigenous peoples, current distribution of languages, proportion of habitat loss, urban–rural catchment areas, and hunting monitoring time span. We ran 30 models to the entire Amazonia with 70% of observations each and took the central and 75% and 90% quantiles of these 30 overall individual animals HP spatial models. The ranking importance of each predictor in the full model (that is, with 100% of the observations) is given in Supplementary Data 1. Individual animals HP (individuals HP) reflects how the overall individual animals HHR varies with environmental and anthropogenic factors (Fig. 1b,c).

## TSOP and taxon-specific individual animals HP
We obtained 590 georeferenced observations on the number of animals hunted of each taxon in each locality of the Marupiara dataset (Fig. 1a). We extracted the values of all spatial variables for these georeferenced localities. We assigned a value of zero (absence) when the hunted taxon was predicted to occur in that locality according to the IUCN spatial database, but no animals were hunted there. We then spatially predicted the individual animals HP for the 174 hunted taxa covered in our dataset. To accomplish that, we first calculated the TSOP of the 174 hunted taxa in the 590 localities as the number of animals hunted of a taxon in each locality divided by the total number of animals hunted in the same locality. We ran spatially explicit random forest models for the 174 taxa using their TSOP as functions of the same environmental and anthropogenic factors used to estimate the overall individual animals HP, except for the hunting recording time span not considered in the TSOP random forest models.

To prevent underestimation of the TSOP of some taxa, we removed localities that contained data for both the lower and the corresponding high taxa when modelling for the lower taxon. For example, localities that contained hunting data for both *M. americana* and *Mazama* spp. were removed from the analysis when the TSOP of *M. americana* was modelled.

Each of the 174 TSOP rasters was clipped to its respective geographic distribution obtained from the IUCN database[18]. To ensure that the proportions of the 174 TSOP rasters summed to 1 in each spatial cell, we normalized each raster by dividing it by the sum of all 174 TSOP rasters, improving the accuracy of TSOP spatial predictions.

We then multiplied the overall individual animals HP raster by each of the 174 TSOP rasters. This allowed us to spatially predict the individual animals HP for each of the 174 hunted taxa, resulting in a taxon-specific individual animals HP. The sum of the 174 taxon-specific individual animals HP equals the overall individual animals HP.

## Taxon-specific and overall animal biomass HP
By multiplying the 174 taxon-specific individual animals HP rasters by their respective estimated median and 0.10, 0.25, 0.75 and 0.90 quantiles of taxon-specific average body mass, we built the 174 taxon-specific animal biomass HP rasters (see Fig. 3). By summing all the 174 taxon-specific animal biomass HP rasters, we got the overall animal biomass HP raster (Fig. 1c).

## Overall HP
Overall HP regarding individual animals and animal biomass reflects how the overall individual animals HHR and animal biomass HHR vary

with environmental and anthropogenic factors throughout Amazonia (Fig. 1b,c).

## Taxon-specific and overall individual animals offtake
We spatially predicted the individual animals offtake–the annual number of animals hunted per taxon–for the 174 hunted taxa across Amazonia by multiplying the 174 rasters of taxon-specific individual animals HP (the estimated number of animals hunted per taxon per hunter per day in each pixel) by the RHPS raster (the estimated number of rural hunters in each pixel). To avoid overestimations in areas with high rural hunter density, we applied an upper limit on the number of animals hunted per taxon per pixel: For 139 taxa (primarily species-specific or allopatric genera), we truncated the maximum offtake values based on their taxon-specific average density, defined as the average number of individuals per 100 km² or 10 × 10 km (Supplementary Table 2 and Supplementary Data 5).

We then calculated the number of animals hunted per taxon per year in Amazonia by summing the values from the 174 taxon-specific individual animals offtake rasters. By aggregating these, we generated the overall individual animals offtake raster (Fig. 1d), from which we derived the total number of animals hunted per year by summing the pixel values across the entire Amazonia region.

## Taxon-specific and overall animal biomass offtake
The spatial prediction of the taxon-specific animal biomass offtake (the animal biomass in kg extracted of the 174 hunted taxa across Amazonia) was accomplished by multiplying each one of the 174 rasters of taxon-specific individual animals offtake by their respective estimated median and 0.10, 0.25, 0.75, and 0.90 quantiles of taxon-specific average body mass. By summing all the 174 taxon-specific animal biomass offtake rasters, we got the raster of the overall animal biomass offtake (Fig. 1d). From the sum of values of the pixels of overall animal biomass offtake raster, we calculated the total animal biomass extracted (in kg) per year in Amazonia.

## Proportion of edible wild meat to undressed biomass
We estimated the overall annual production of edible wild meat in Amazonia (see 'Metrics estimation' for details) using data on the proportion of consumable meat contained in animal carcasses reported in the literature[75,76]. Edible yield proportions were calculated separately for each of the 20 key hunted taxa and then pooled by major taxonomic groups: mammals (0.63 ± 0.11), birds (0.73 ± 0.06), chelonians (0.47 ± 0.14) and caimans (0.45 ± 0.04). These values were then multiplied by the estimated total undressed animal biomass offtake for the corresponding taxa or groups. Based on this approach, we estimated that edible wild meat represents approximately 58.5% of the total undressed biomass harvested annually across Amazonia.

## Available wild meat per rural inhabitant
Therefore, we multiplied the predicted raster of the daily animal biomass offtake by 0.585 to spatially predict the daily overall edible wild meat produced across Amazonia. Then, we divided the overall edible wild meat produced raster by the APPS raster, which includes the number of rural inhabitants per spatial cell, generating a raster of the available wild meat per rural inhabitant in each spatial cell (Fig. 4).

## Wild meat nutritional composition
Using the scarce data from the literature on the nutritional composition in meat for 26 taxa (22 species and 4 genera; Supplementary Table 9) in Amazonia[77–87], we estimated the average amount of energy and macro- and micronutrients in wild meat in Amazonia (Extended Data Table 1). We used 265 observations overall from these literature sources (for example, 59 for protein, 58 for total fat, 44 for energy, 28 for iron, 7 for zinc, 2 for selenium, 20 for vitamin $B_1$, vitamin $B_2$ and vitamin $B_3$ and 7 for vitamin $B_{12}$ (Supplementary Table 9). Out of the 20 dominant taxa, 12 had species-specific data on nutritional composition (Supplementary Table 9).

A limitation of our study stems from the reliance on approximate rather than species-specific nutritional composition data. This constraint is primarily due to the limited availability of detailed nutritional data in the existing scientific literature[88]. To address this challenge, we adopted the food-matching technique, a methodology endorsed by the Food and Agriculture Organization (FAO) for such scenarios[89].

## Daily amounts of energy and nutrients furnished by wild meat
We then produced 10 rasters of the daily amounts of energy and nutrients furnished by wild meat (that is, one for energy and nine for nutrients) for each spatial cell by multiplying the overall edible wild meat produced raster by the estimated average values of energy and macro- and micronutrients contained in Amazonian wild meat (Extended Data Table 1). See 'Metrics estimation' for details on estimation.

## Percentage of dietary requirements furnished by wild meat
To estimate the nutritional needs of micronutrients of the Amazonian peoples we used DRIs. DRIs are a set of recommendations for nutrient intake based on the latest scientific evidence and intended to guide the amounts of nutrients that are needed to maintain health. Due to the lack of specific nutritional data for the targeted population, such as weight and food consumption, we had to rely on general references measured in grams per day instead of grams per kilogram per day of a given nutrient. Consequently, we selected the estimated average requirement (EAR) values, measured in weight per day, as our primary choice. This approach was taken because the EAR reflects the average daily nutrient intake estimated to meet the needs of half of the healthy individuals within a specific age and gender group[90,91]. In cases where there was no EAR available, we used the adequate intake (AI) or recommended dietary allowance (RDA) as a guide for determining the appropriate recommendation of a nutrient. For total fat, in instances where DRIs were unavailable, we utilized the acceptable macronutrient distribution range (AMDR). Considering the midpoint of the range, we converted these values to a percentage of energy. Finally, for energy (that is, calories), we applied the estimated energy requirements (EER)[92]. The DRI values are presented in the Supplementary Tables 10 and 11.

First, we constructed a raster of the population size for each sex-age group (that is, children, women and men) by multiplying the AMPS raster with the proportion of each group relative to the total rural population in Amazonia. These proportions were derived from data available for north Brazil through the Brazilian population census[93]. Using this approach, we established spatially explicit demographic distributions reflecting rural Amazonia's population structure.

Then, we spatially predicted each sex-age group's minimum daily dietary requirements for seven micronutrients, two macronutrients, and energy. This was achieved by multiplying the average reference values for each nutrient or energy requirement with the corresponding sex-age population raster. The resulting minimum daily dietary requirements rasters for children, women, and men were then summed for each nutrient and energy to predict the overall minimum daily dietary requirements for Amazonian peoples across all spatial cells. As a result, these requirements were directly proportional to the number of rural inhabitants per pixel, ensuring that the analysis accurately reflected localized nutritional needs across the region.

Finally, the daily nutrient amounts furnished by the estimated wild meat production in Amazonia were evaluated against the daily micronutrient and macronutrient requirements to determine their adequacy for a nutritionally balanced diet for Amazonian peoples. This was done by comparing the corresponding spatial cell of a given nutrient/energy between the daily amounts furnished by wild meat and the minimum daily dietary requirements of Amazonian peoples. This process allowed

us to spatially explicit the levels of energy and nutrients supplied by wild meat to Amazonian peoples.

## Metrics estimation

To enhance the accuracy of our metrics (such as taxon-specific body mass, taxon-specific density, proportion of hunters to consumers, proportion of edible wild meat to undressed biomass and wild meat nutritional composition), we run each metric 1,000 times, drawing each estimated quantity from a normal statistical distribution defined by its corresponding mean and standard error. After obtaining 1,000 values for each metric, we took the mean, standard deviation and 90% quantiles of these 1,000 values to produce a 90% confidence interval.

## Research collaboration and ethics

The community-based hunting monitoring initiatives that contribute primary data to the Marupiara dataset were developed and implemented through close partnerships with Indigenous and traditional peoples. These initiatives should not be seen as conventional academic research, where external researchers extract data on biodiversity use. Instead, they are grounded in sociopolitical realities relevant to Indigenous and traditional communities and are designed to strengthen their territories and livelihoods, empower communities, support wildlife conservation and management, enhance food security, and protect cultural practices—always with respect for their priorities and autonomy. The data collected directly informs community-led decisions on sustainable wildlife and territorial management, with meaningful community engagement throughout the monitoring cycle. Results are transparently shared at the community level and used to guide practical actions on the ground. Each project is tailored to the specific needs of the communities involved, ensuring that Indigenous and traditional knowledge is respected, safeguarded and valued. Our collaborative approach includes training Indigenous and traditional hunters and researchers, fostering long-term capacity building and education. Wildlife monitoring programmes are conducted under government regulation, ensuring ethical data handling and confidentiality. All research activities are formally approved by Indigenous and traditional communities, as well as by relevant academic or governmental institutions overseeing Indigenous Lands, parks, and extractive reserves.

Data-sharing agreements were established among communities, researchers, and technicians, enabling informed local decision-making and advancing research on wildlife use, management, and conservation in Amazonia. Free, prior, and informed consent (FPIC) was obtained—either orally or in writing—from all communities participating in those initiatives, ensuring ethical engagement and respect for rights, welfare, and autonomy. While some early initiatives predated formal non-Indigenous and local ethics committees (for example, before the Nagoya Protocol, 2010), agreements were always culturally adapted, ranging from oral consensus to written contracts detailing research objectives, participant rights, and data use (Supplementary Methods 4).

Six independent ethics committees reviewed and approved all primary data methods, with the 12 contributing initiatives receiving clearance from institutional review boards across Brazilian, Peruvian, and French Guianese Amazonia (Supplementary Methods 4). These approvals, secured through universities and research institutions, guaranteed compliance with international ethical standards for research involving Indigenous and traditional peoples.

The primary data collection methods were approved by the main representative organizations of Indigenous peoples and traditional communities in the Brazilian Amazon—respectively, the Coordination of Indigenous Organizations of the Brazilian Amazon (COIAB) and the National Council of Extractive Populations (CNS). Both organizations have formally endorsed the content of this article and have committed to participating in the Evaluation Committee for future research utilizing the Marupiara dataset (Supplementary Methods 6 and 7).

## Reporting summary

Further information on research design is available in the Nature Portfolio Reporting Summary linked to this article.

## Data availability

The Marupiara dataset that was built and analysed in this study is not publicly available owing to sensitivity and privacy concerns related to hunters and their communities. Similarly, the R code supporting the analyses presented in this article was made available during the editorial process exclusively for review purposes. However, the authors can make the data available upon request for research purposes only, where the researcher provides a detailed written document outlining the study's objectives and a signed letter of commitment to the ethical and exclusive use of the data for the specified research. Each research proposal must receive explicit approval from the Evaluation Committee, which is composed of all dataset contributors, along with representatives from Coordination of Indigenous Organizations of the Brazilian Amazon (COIAB), National Council of Extractive Populations (CNS) and the RedeFauna research network. The Committee is responsible for reviewing all requests for access to raw data, digital variables or any other information used in this study. Conditions on re-use include acknowledgment of the data source and adherence to ethical standards outlined in the agreement. If a contributor or Indigenous or traditional representative chooses not to share their data for a particular study, those specific data will be removed without impeding the overall research. Contributors retain full autonomy over using their own data and can withdraw from the agreement at any time without prior notice. Requests should be sent to aapardalis@gmail.com.

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

**Acknowledgements** We are especially grateful to the rural Amazonian hunters and their families for their enormous collaboration during our studies and work. We thank R. Fewster for her suggestions on statistical analysis; J. Shimbo and M. Butti for providing detailed information on spatial deforestation data in Amazonia from MapBiomas; P. Fearnside for sharing insights on carbon emissions resulting from deforestation in the region. A.P.A. thanks the Conselho Nacional de Desenvolvimento Científico e Tecnológico (CNPq), Center for International Forestry Research and World Agroforestry (CIFOR-ICRAF), Comunidad de Manejo de Fauna Silvestre en América Latina (COMFAUNA), The National Geographic Society, Fundação Nacional dos Povos Indígenas (FUNAI), Federação das Organizações e Comunidades Indígenas do Médio Purus (FOCIMP) and Federação das Organizações Indígenas do Rio Negro (FOIRN). J.E.F., H.R.E.B. and L.C. thank the UK Research and Innovation's Global Challenges Research Fund (UKRI GCRF) through the Trade, Development and the Environment Hub project (ES/S008160/1), and the United States Agency for International Development (USAID). M.C.M.J. thanks the CNPq (proc. 402334/2021-3 and 306755/2021-1). E.A.R.C. thanks the Programa Monitora ICMBio, Programa ARPA. M.A.S.J. thanks the Gordon and Betty Moore Foundation. A.V.N. thanks the Conselho Nacional de Desenvolvimento Científico e Tecnológico (CNPq) process no. 160547/2023-7 and Center for International Forestry Research and World Agroforestry (CIFOR-ICRAF). J.P.B. thanks the UK Research and Innovation's (NERC) NE/T000341/1. PEP thanks the Instituto del Bien Común (IBC).

**Author contributions** A.P.A., P.d.A.L.C., D.P.M., H.R.E.B. and A.V.N. built and organized the Marupiara dataset. R.E.B., P.M., J.V., T.Q.M., C.R.-H., P.d.A.L.C., A.P.A., J.C.B.P., E.A.R.C., J.P.B., E.M.v.M., A.V.N., M.A.O., M.J.d.P., C.K.M.L., N.C.P., J.L.K., J.V.C.-S., M.A., G.H.R., M.A.S.J. and H.R.E.B. provided primary hunting data. P.d.A.L.C., A.P.A. and H.R.E.B. tabulated secondary data. M.A.C., P.E.P., P.E.P.-P., T.Q.M., T.G.F., M.A.R.d.M.V., R.M.A., L.M., G.L., L.M.G.S., E.C.S.P., A.V.C.P., J.L.V.S.P., G.C.P., A.P.R.L.K., D.B., V.S.M.B., W.S.L.B., A.O.F.B., Y.W., J.S.A., V.S.S.A., J.O.G.T., J.T.P.K., A.B.B., I.J.S.B. and E.P.A.L.T. provided invaluable assistance in the field during data collection. M.A.C., P.E.P., P.E.P.-P., T.Q.M., T.G.F., M.A.R.d.M.V., R.M.A., L.M., G.L., C.K.M.L., L.M.G.S., E.C.S.P., A.V.C.P., J.L.V.S.P., G.C.P., A.P.R.L.K., D.B., V.S.M.B., W.S.L.B., A.O.B., Y.W., J.S.A., V.S.S.A., J.O.G.T., E.P.A.L.T., J.L.K., J.T.P.K., A.B.B., I.J.S.B., D.T., V.S.F.A. and K.B.K. collected primary data or provided information on hunting practices in their territories. J.S.A. made the drawings for the figures. A.P.A., H.R.E.B., P.d.A.L.C., J.E.F., B.W.N., M.C.M.J. and D.P.M. analysed the data. A.P.A. and B.W.N. performed spatial analyses. A.P.A., M.C.M.J. and A.I. performed nutritional analyses. A.P.A., H.R.E.B. and J.E.F. wrote the manuscript. P.d.A.L.C., M.F.C., T.Q.M., B.W.N., M.C.M.J., A.I., P.M., L.C., E.A.R.C., J.C.B.P., R.E.B., D.P.M., J.V.C.-S. and C.R.-H. discussed and contributed to the manuscript.

**Competing interests** The authors declare no competing interests.

**Additional information**
**Correspondence and requests for materials** should be addressed to André Pinassi Antunes or Thais Q. Morcatty.

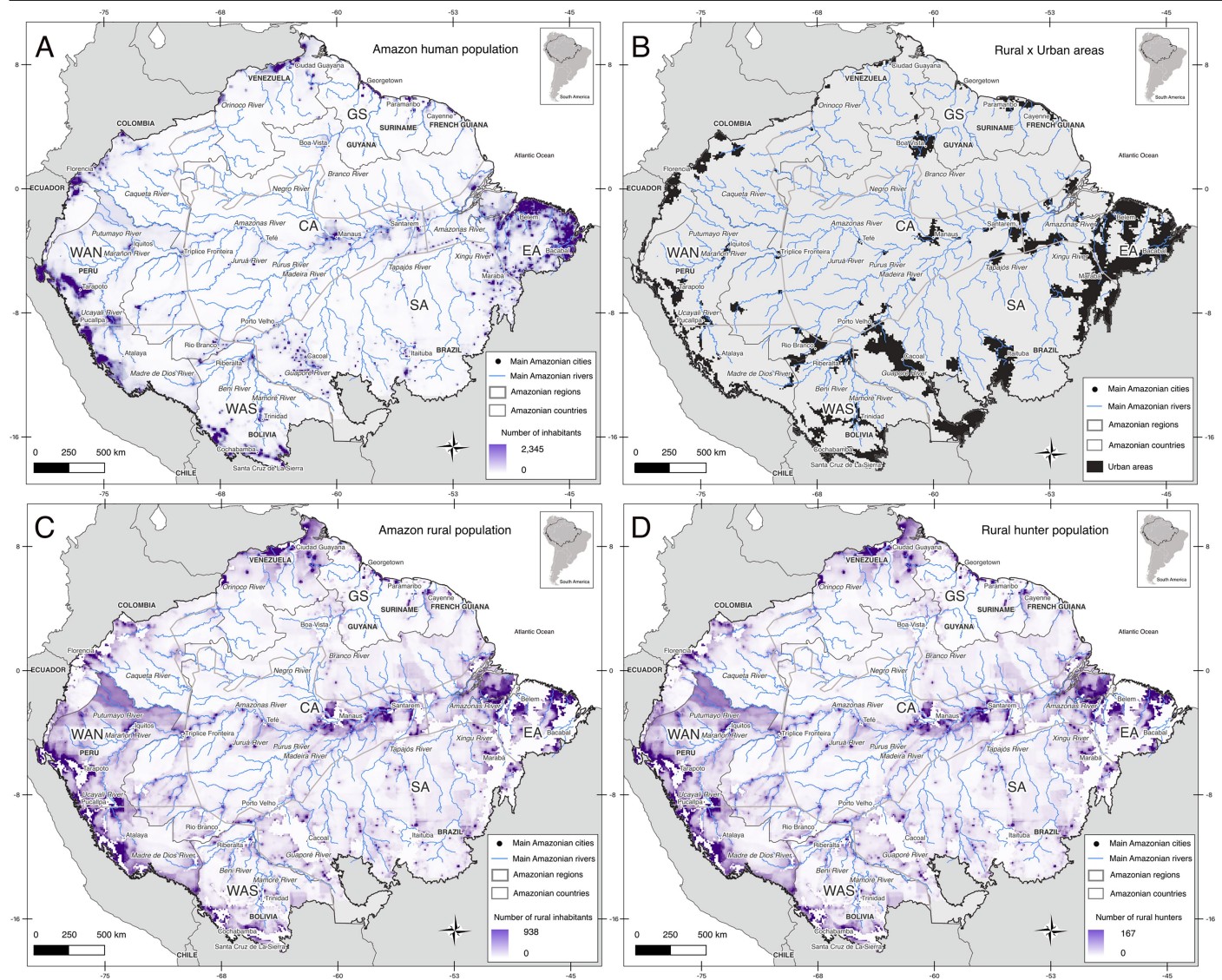

**Extended Data Fig. 1 | Heatmaps of Amazonian peoples and rural hunters.**
Amazonian peoples were defined as all inhabitants living in rural Amazonia, frequently self-declared as Indigenous, traditional or small-scale farming peoples, and were here estimated at 10,870,022 individuals (Amazonian peoples population size), corresponding to 34.2% of the 31,783,941 individuals living (total human population size) in Amazonia. We developed a raster of the total human population size within Amazonia boundaries (A) and removed all the urban areas (B) as defined in the urban-rural catchment area raster[57], to obtain the Amazonian peoples population size (APPS) raster (C). To obtain the rural hunters population size (RHPS) raster (D) we multiplied the APPS raster by 0.178 (median) and by 0.168, 0.173, 0.183 and 0.187 (the 0.10, 0.25, 0.75, and 0.90 quantiles), which represent the estimated proportion of hunters to consumers based on data from 72 localities of the Marupiara Dataset with both information of number of hunters and number of consumers. Based on these calculations, the total number of rural hunters in Amazonia was estimated at 1.93 ± 0.08 (1.83–2.04) M.

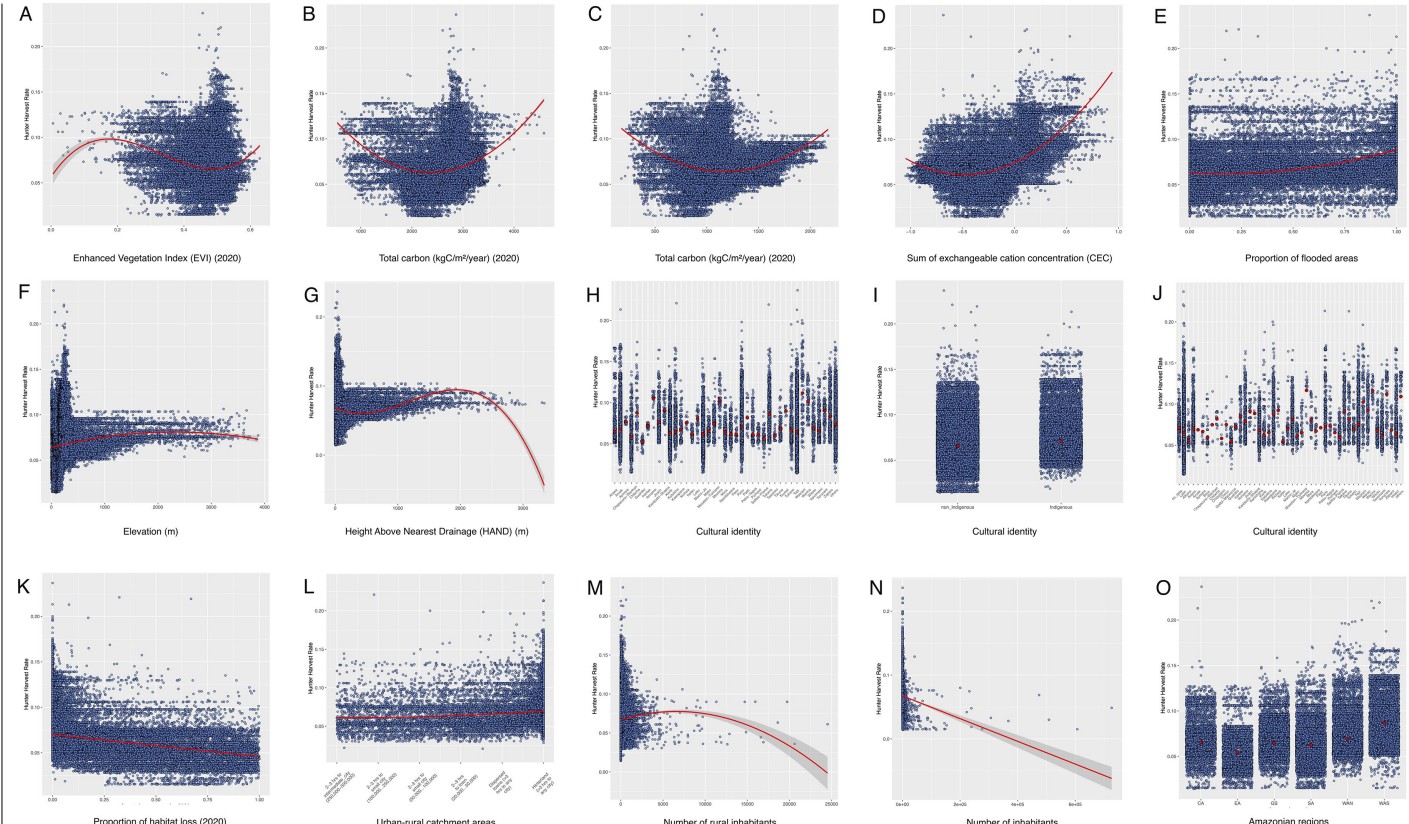

**Extended Data Fig. 2 | Relationships between Hunter Harvest Rate (HHR) and spatial variables.** A-L represent the environmental and anthropogenic variables used to model HHR.

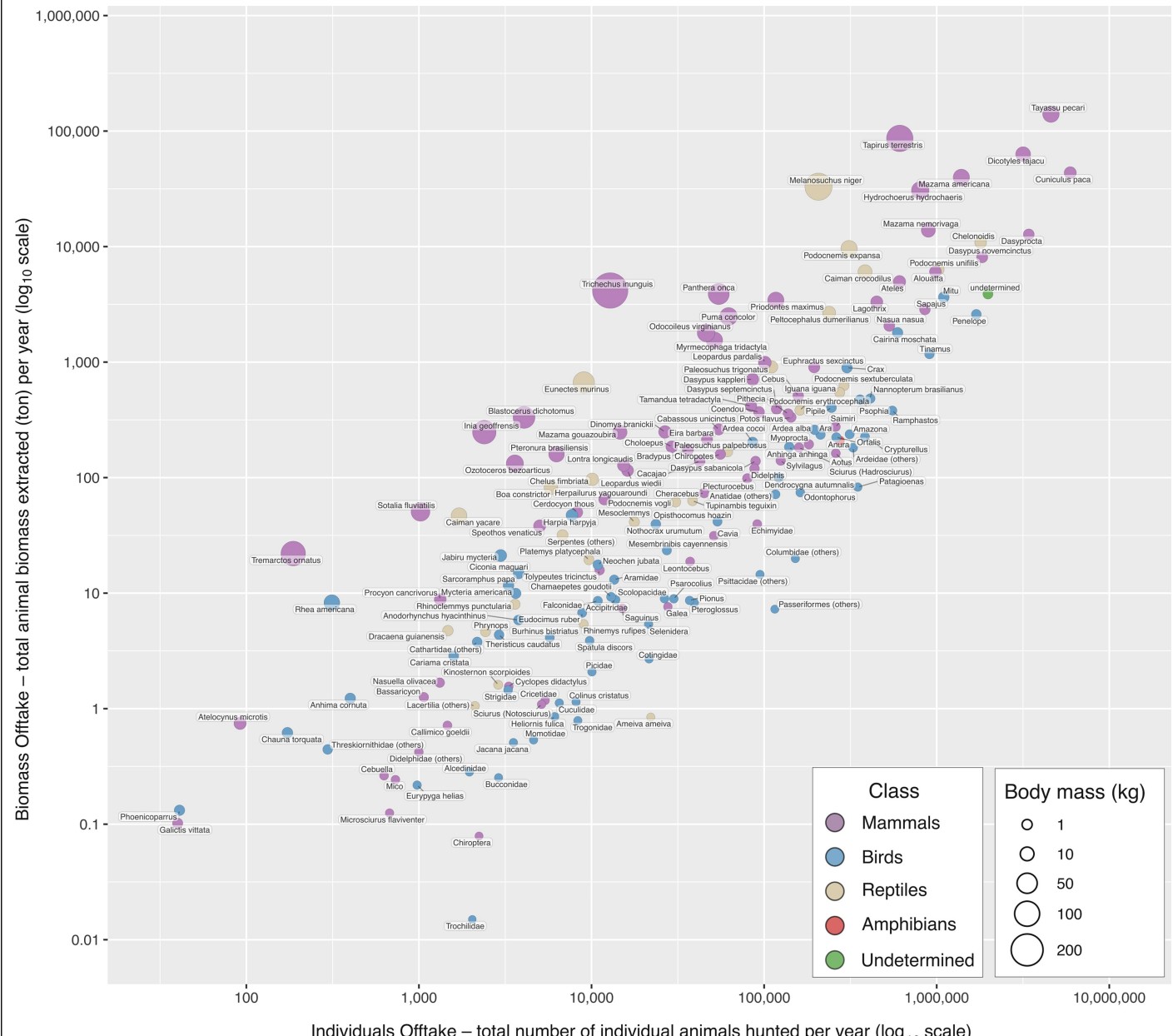

**Extended Data Fig. 3 | Relationship between the annual offtake of number of individuals hunted (on a log₁₀ scale) and biomass extracted (on a log₁₀ scale) for 174 animal taxa recorded in Amazonia.** Size of dots is proportional to the average taxon-specific body mass and colours represent animal classes.

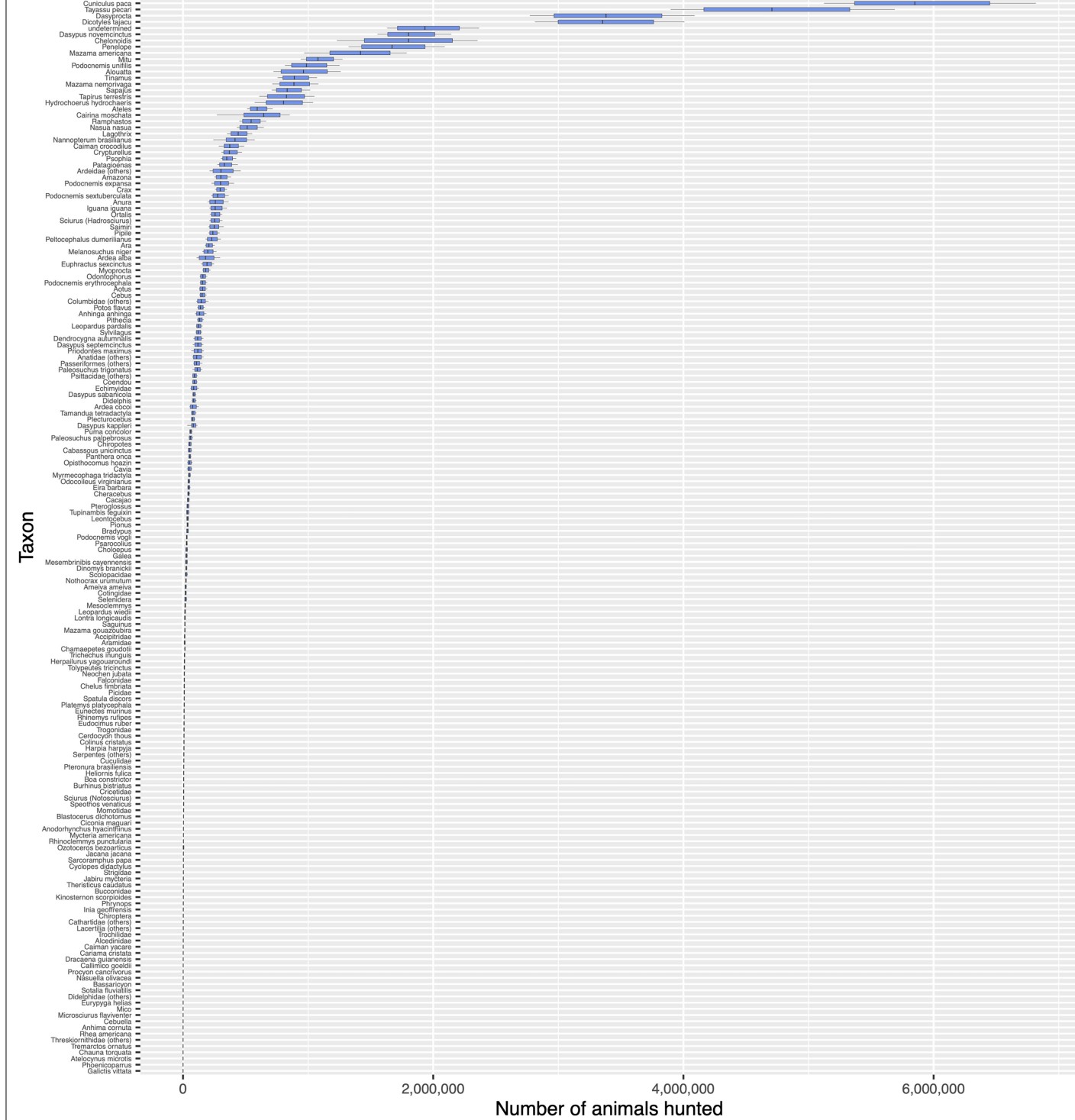

**Extended Data Fig. 4 | Estimated number of animals hunted per taxon per year in Amazonia (individual animals offtake).**

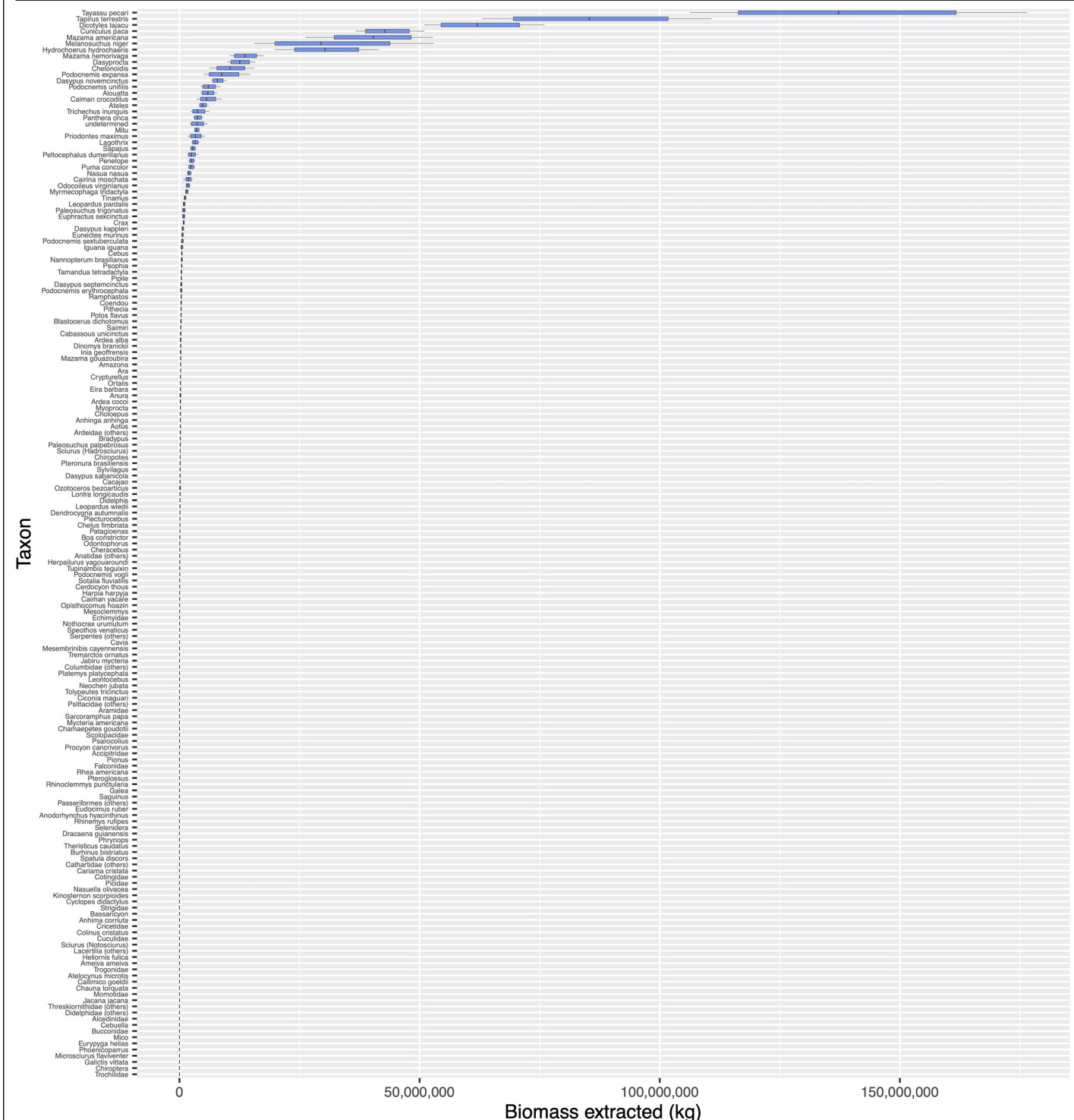

**Extended Data Fig. 5 | Estimated biomass extracted per taxon per year in Amazonia (animal biomass offtake).**

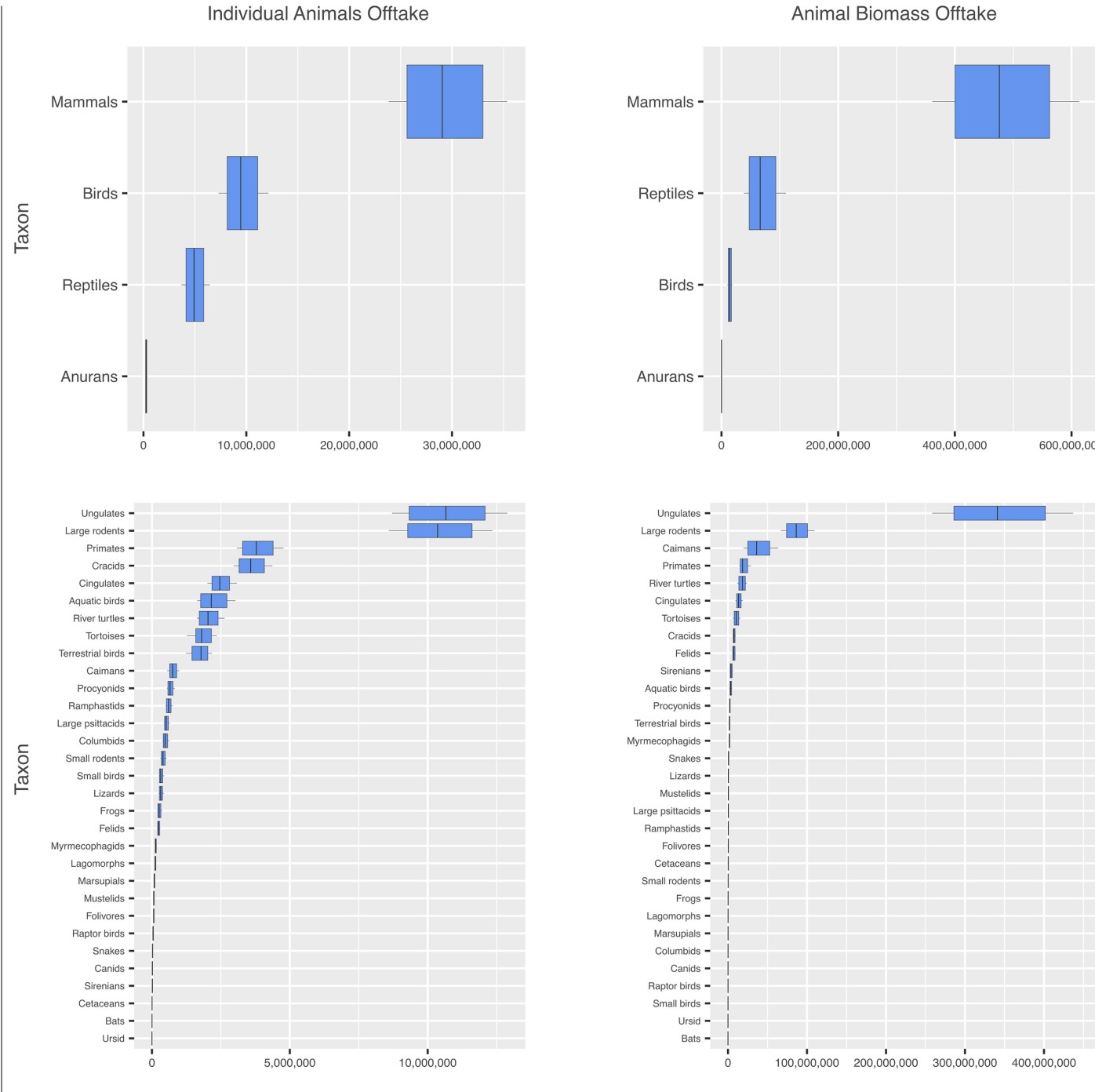

**Extended Data Fig. 6 | Estimated number of individuals hunted (individual animals offtake) and biomass extracted (animal biomass offtake) per animal class and animal groups per year in Amazonia.**

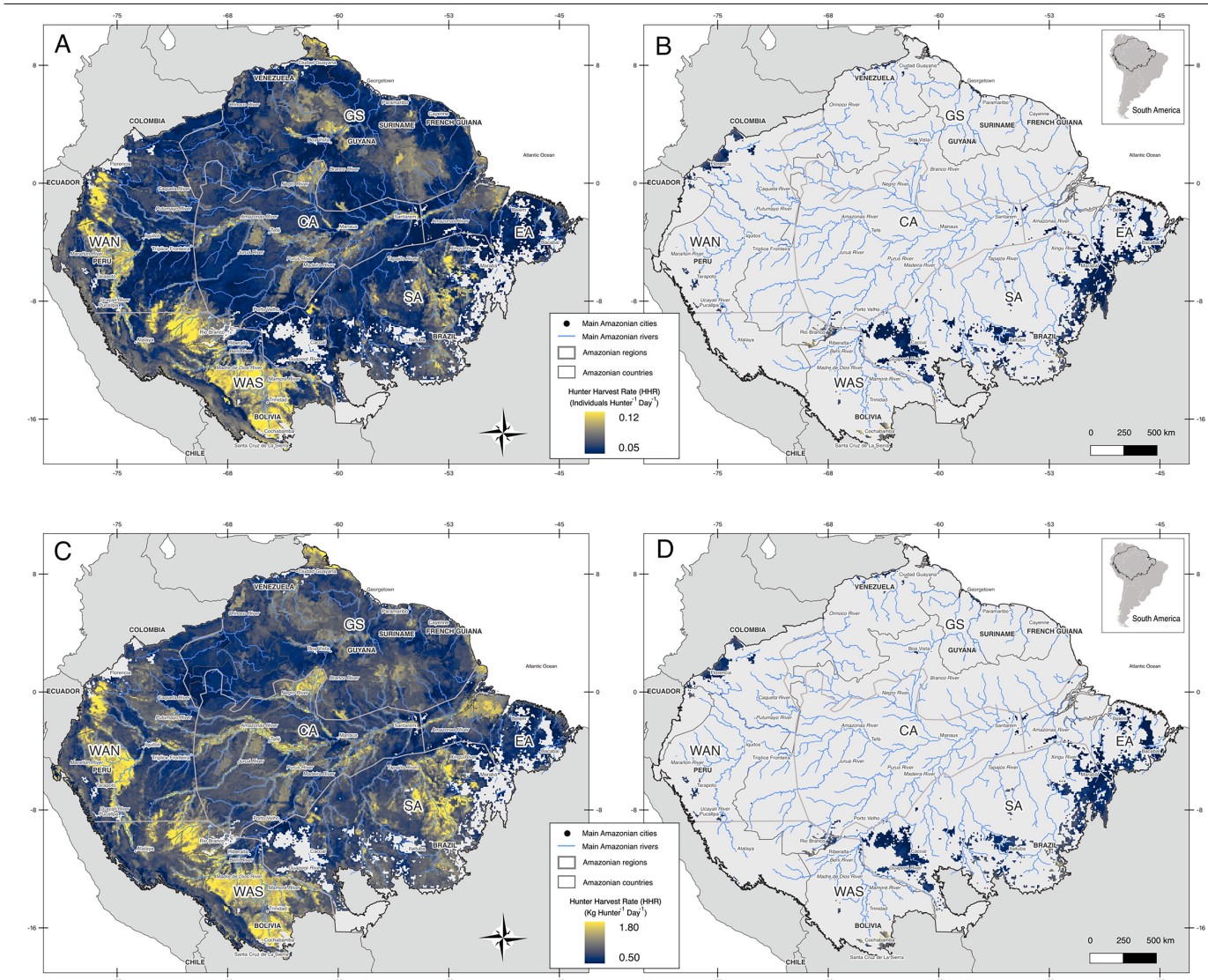

**Extended Data Fig. 7 | Spatial predictions of individual animals hunter harvest rate (HHR) and animal biomass hunter harvest rate (HHR) in areas with <70% of habitat loss and areas with >70% of habitat loss in Amazonia.** (A) Individual animals HHR in areas with <70% of habitat loss; (B) Individual animals HHR in areas with >70% of habitat loss; (C) Animal biomass HHR in areas with <70% of habitat loss; (D) Animal biomass HHR in areas with >70% of habitat loss.

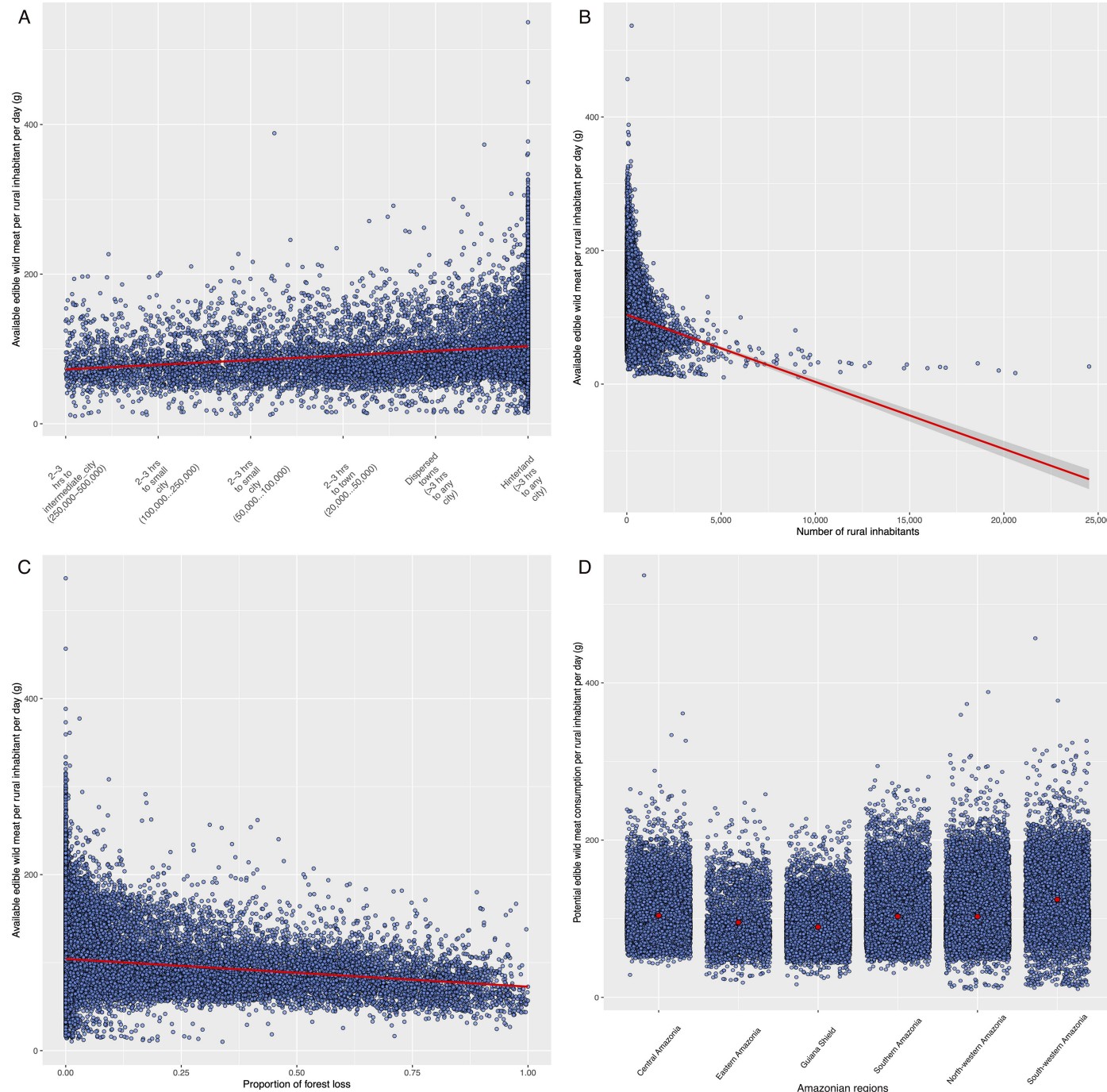

**Extended Data Fig. 8 | Relationships between wild meat per rural inhabitant and urban catchment area, number of inhabitants, forest loss and Amazonian regions. a–d**, Relationships between the spatial prediction of the average available edible wild meat per rural inhabitant and the level of the urban-rural catchment areas (**a**), the number of rural inhabitants (**b**), the proportion of forest loss (**c**), and Amazonian regions (**d**).

**Extended Data Table 1 | Estimated energy, macro- and micronutrient contents (mean and standard deviation) in 100 g of Amazonian wild meat**

| | Calories (Kcal) | Protein (g) | Total fat (g) | Iron (mg) | Zinc (mg) | Selenium (μg) | Vitamin B1 (mg) | Vitamin B2 (mg) | Vitamin B3 (mg) | Vitamin B12 (μg) |
|---|---|---|---|---|---|---|---|---|---|---|
| Amazonian wild meat | 129.8 ± 7.26 | 21.9 ± 0.51 | 3.86 ± 0.66 | 2.69 ± 0.47 | 1.67 ± 0.44 | 12.65 ± 0.25 | 0.16 ± 0.03 | 0.27 ± 0.02 | 4.45 ± 0.70 | 2.61 ± 0.75 |

Estimation was based on 265 observations for protein (59), Total fat (58), Energy (44), Iron (28), Zinc (7), Selenium (2), Vitamin B1 (20), Vitamin B2 (20), Vitamin B3 (20) and Vitamin B12 (7) from 26 Amazonian wild meat species. We ran each nutritional component 1000 times, drawing each estimated quantity from a Normal statistical distribution defined by its corresponding mean and standard error. After obtaining 1000 values for each metric, we took the mean and standard deviation.

# Reporting Summary

## Statistics

For all statistical analyses, confirm that the following items are present in the figure legend, table legend, main text, or Methods section.

| n/a | Confirmed | |
|---|---|---|
| ☒ | ☐ | The exact sample size (*n*) for each experimental group/condition, given as a discrete number and unit of measurement |
| ☒ | ☐ | A statement on whether measurements were taken from distinct samples or whether the same sample was measured repeatedly |
| ☐ | ☒ | The statistical test(s) used AND whether they are one- or two-sided<br>*Only common tests should be described solely by name; describe more complex techniques in the Methods section.* |
| ☐ | ☒ | A description of all covariates tested |
| ☐ | ☒ | A description of any assumptions or corrections, such as tests of normality and adjustment for multiple comparisons |
| ☐ | ☒ | A full description of the statistical parameters including central tendency (e.g. means) or other basic estimates (e.g. regression coefficient) AND variation (e.g. standard deviation) or associated estimates of uncertainty (e.g. confidence intervals) |
| ☒ | ☐ | For null hypothesis testing, the test statistic (e.g. *F*, *t*, *r*) with confidence intervals, effect sizes, degrees of freedom and *P* value noted<br>*Give P values as exact values whenever suitable.* |
| ☒ | ☐ | For Bayesian analysis, information on the choice of priors and Markov chain Monte Carlo settings |
| ☐ | ☒ | For hierarchical and complex designs, identification of the appropriate level for tests and full reporting of outcomes |
| ☒ | ☐ | Estimates of effect sizes (e.g. Cohen's *d*, Pearson's *r*), indicating how they were calculated |

*Our web collection on statistics for biologists contains articles on many of the points above.*

## Software and code

Policy information about availability of computer code

| Data collection | no software was used |
|---|---|
| Data analysis | The analyses were conducted using R software (version 4.1.0), employing several packages for statistical and spatial modelling, including terra (version 1.3-17), sf (version 1.0-8), raster (version 3.4-13), and randomForest (version 4.6-14). Geospatial analyses and map production were performed using QGIS (version 3.22.4), with final editing done in Inkscape (version 1.1) and GIMP (version 2.10.28). |

For manuscripts utilizing custom algorithms or software that are central to the research but not yet described in published literature, software must be made available to editors and reviewers. We strongly encourage code deposition in a community repository (e.g. GitHub). See the Nature Portfolio guidelines for submitting code & software for further information.

## Data

Policy information about availability of data

All manuscripts must include a data availability statement. This statement should provide the following information, where applicable:

- Accession codes, unique identifiers, or web links for publicly available datasets
- A description of any restrictions on data availability
- For clinical datasets or third party data, please ensure that the statement adheres to our policy

The dataset analyzed during the current study is not publicly available due to sensitivity and privacy concerns related to hunters and their communities. However,

the data can be made available from the corresponding author on reasonable request, contingent on the permission of both the authors and local representatives. Please note, under no circumstances, can the dataset be used in a manner that puts hunters and their communities at risk.

# Research involving human participants, their data, or biological material

Policy information about studies with human participants or human data. See also policy information about sex, gender (identity/presentation), and sexual orientation and race, ethnicity and racism.

| | |
|---|---|
| Reporting on sex and gender | The study does not involve direct research on human participants. Instead, we worked directly with, or utilized published data on, hunting practices by traditional and indigenous communities in Amazonia. Where data was gathered by the authors, hunting communities gave their consent to participate in sharing information on the animal species and numbers of individuals hunted. All communities involved were subjected to clear agreements, including Free, Prior, and Informed Consent (FPIC) procedures, ensuring ethical engagement and respect for their rights and autonomy. No specific information on gender or sex in these communities was necessary to be considered in our study, as the inclusive nature of the hunting practices, where wild meat is shared among families and communities, ensures that the data represents all members of the community. |
| Reporting on race, ethnicity, or other socially relevant groupings | Although each community where hunting data was collected (from both primary and secondary sources) was characterised according to its ethnic group, this information was not used in the primary analyses. However, we recognize that distinct ethnic groups may hunt different species due to cultural aspects and preferences. To investigate how different ethnic groups influence hunting patterns in Amazonia, we used a raster of the Amazonian language families as a covariate. This raster was constructed from maps provided by Loukotka (1967) and Eriksen (2011). |
| Population characteristics | The study focuses on wildlife hunting practices and does not involve collecting demographic data on human populations. |
| Recruitment | Most communities from which our hunting data were extracted were actively and formally involved in hunting monitoring schemes, some of which have been active for decades. Each of these initiatives followed an explicit process to encourage hunters or monitors to participate voluntarily. The selection procedures, along with details of the communities and individuals involved, are documented elsewhere. These reports provide comprehensive insights into the engagement processes and methodologies employed, ensuring accurate and reliable data collection. |
| Ethics oversight | This study created and analyzed the most comprehensive dataset to date on hunting and wild meat extraction in Amazonia, without involving human participants or human data. As a result, ethical approval specific to human subjects was not required. |

Note that full information on the approval of the study protocol must also be provided in the manuscript.

# Field-specific reporting

Please select the one below that is the best fit for your research. If you are not sure, read the appropriate sections before making your selection.

☐ Life sciences          ☐ Behavioural & social sciences          ☒ Ecological, evolutionary & environmental sciences

For a reference copy of the document with all sections, see nature.com/documents/nr-reporting-summary-flat.pdf

# Ecological, evolutionary & environmental sciences study design

All studies must disclose on these points even when the disclosure is negative.

| | |
|---|---|
| Study description | This study analyses the hunting practices of Amazonian peoples—including rural Indigenous, traditional, and farming communities—over nearly six decades, with a focus on the composition of hunted taxa and hunter harvest rates. Utilizing both primary and secondary data on wild terrestrial vertebrates, the study derives key metrics such as Taxon-Specific Offtake Proportion (TSOP) and Hunter Harvest Rate (HHR). Additionally, random forest models are employed to spatially predict these metrics in relation to environmental and anthropogenic variables. |
| Research sample | The research sample includes data on 447,438 individual animal kills recorded across 647 rural localities in Amazonia. The communities involved in this data collection consist of Indigenous, traditional, and farming peoples residing in rural areas of the region. These localities were selected due to the presence of existing hunting monitoring projects. Data sources encompass primary data from long-term monitoring efforts and shorter studies conducted by local monitors, as well as secondary data from published literature. This information details species, the number of animals, biomass hunted, and hunting effort. |
| Sampling strategy | The sampling strategy included both long-term monitoring initiatives and shorter studies conducted by the authors, supplemented by secondary data extracted from the literature. The selection of communities, hunters, or monitors for primary data collection was based on the specific objectives of each monitoring initiative, rather than being tailored specifically for this study. |
| Data collection | Secondary data were obtained from published sources, ensuring comprehensive coverage across different regions of Amazonia. |
| Timing and spatial scale | The study encompasses data collected from 1965 to 2023, covering a range of temporal scales. Spatially, it includes 647 localities across the entire Amazon basin, offering a comprehensive view of hunting practices on a large scale. |

| | |
|---|---|
| Data exclusions | Data on wild meat trade and urban consumption were excluded to maintain a focus solely on hunting practices within rural Amazonian communities. Additionally, any records with missing or unclear information were omitted from the analysis. |
| Reproducibility | Measures to ensure reproducibility included the use of established methodologies for data collection and analysis, such as standardized monitoring protocols and widely recognized statistical and spatial analysis tools. Detailed descriptions of data sources, sampling strategies, and analytical procedures are provided in the Methods section to facilitate replication. Furthermore, the R scripts used for the analysis are available from the corresponding author upon reasonable request. |
| Randomization | Randomization was not applicable in this observational study, as the data reflect natural hunting practices recorded over time. All possible available data were included to ensure comprehensive coverage and representation. |
| Blinding | Blinding was not applicable as this study involved the analysis of preexisting data on wildlife hunting. The data were collected through observational monitoring of natural hunting practices by local communities, where blinding is not feasible or relevant. |

Did the study involve field work? ☒ Yes ☐ No

## Field work, collection and transport

| | |
|---|---|
| Field conditions | Field data collection at the various sites reported in our research was conducted under a range of environmental conditions typical of the Amazon region, including dense forests, rivers, and floodplains. |
| Location | Data were collected from 647 rural localities across the Amazon basin, representing a variety of ecological zones. More detailed information about these localities can be found in the Methods section. |
| Access & import/export | Access to field sites was coordinated with local communities and authorities. Since the study focused on data collection within the Amazon region, there was no need for import or export of materials. |
| Disturbance | In all reported study sites, minimal disturbance to wildlife and habitats was prioritized, with data collection methods designed to be non-intrusive and respectful of local customs and regulations. All research was conducted in areas where a prior trust relationship had been established with local communities. |

# Reporting for specific materials, systems and methods

We require information from authors about some types of materials, experimental systems and methods used in many studies. Here, indicate whether each material, system or method listed is relevant to your study. If you are not sure if a list item applies to your research, read the appropriate section before selecting a response.

## Materials & experimental systems

| n/a | Involved in the study |
|---|---|
| ☒ | ☐ Antibodies |
| ☒ | ☐ Eukaryotic cell lines |
| ☒ | ☐ Palaeontology and archaeology |
| ☒ | ☐ Animals and other organisms |
| ☒ | ☐ Clinical data |
| ☒ | ☐ Dual use research of concern |
| ☒ | ☐ Plants |

## Methods

| n/a | Involved in the study |
|---|---|
| ☒ | ☐ ChIP-seq |
| ☒ | ☐ Flow cytometry |
| ☒ | ☐ MRI-based neuroimaging |

## Plants

| | |
|---|---|
| Seed stocks | N/A |
| Novel plant genotypes | N/A |
| Authentication | N/A |

