## [Peer Review File · Nature]

Healthy forests safeguard traditional wild meat food systems in Amazonia

Corresponding Author: Dr Andre Antunes

Version 0:

Reviewer comments:

Referee #1

(Remarks to the Author)

This study reports on an immense data set covering the entire Amazon basin, through a combination of primary and secondary sources.

Questions:

How does your manuscript address ecosystem health?

Lines 177-184: can carnivores, primates, tapir, white-lipped peccary be hunted sustainably? Where are offtakes sustainable for these species? Provide examples of hunting / wildlife management systems that achieve sustainability. What should people hunt? How can management systems ensure sustainability?

Lines 200-204: again, can large primates and manatee be hunted sustainably?

Lines 216-220: these examples seem to have very little impact overall. Do they contribute to wildlife conservation?

Lines 231-235: This section is about nutritional value of wild meat, so move the information on economic value/trade/income to another section. Is game meat sold for the same price as beef in local markets?

Lines 292-293: do you have information that chicken provides less micro-nutrients than wild animals?

Lines 294-295: beef is exported, is it not?

Lines 298-299: is 18,890-42,971 km² a lot? Less than 1% of the biome?

Line 315: you mention food health and pathogen surveillance here, but do not discuss these topics anywhere in the manuscript.

Lines 314-315: do you have examples to cite here of successful policies and effective wildlife management developed from taxon-specific extraction maps?

Lines 326-329: do you have examples to cite here?

(Remarks on code availability)

Referee #2

(Remarks to the Author)

MAJOR COMMENTS

This paper estimates overall harvests of terrestrial wild vertebrates in the Amazonian rainforest, employing meta-analysis based on an extensive literature set. The authors evaluated overall and species-specific harvest rates and total offtakes, listing 20 species predominantly hunted. The analysis is original and impressive, the statistical methods for estimating hunting pressure are rigorous, and the results shown are appropriate and informative. Introduction and Discussion are well-served, and the reference list is comprehensive. I only have two concerns, however.

First, the analysis is explicit spatially but not temporally. The dates of data used for this analysis range from 1965 to 2024. Hunting patterns are assumed to have dramatically changed during the 60 years for a variety of reasons, including hunting techniques modernization and economic changes in rural areas. Why didn't the analysis involve this potential temporal variation? I think it is necessary to include this in the analysis by incorporating the data from long-term studies and, more indirectly, by associating the data with environmental and anthropogenic variables that correspond to the period of each survey.

Second, "potential consumption of wild meat" (LL248–269, Figure 4), is misleading. If I understand the analysis well, it assumes that all the harvest in a given area (pixel?) is consumed within the same area, so it does not consider the trade and outside consumers (like tourists). This is a major limitation of this analysis, just as the authors also acknowledged (LL265–266), so I suggest that the authors replace the term "consumption" with "availability" and discuss the consumption with more explicit explanations of this limitation.

MINOR COMMENTS

L48, indigenous: Capitalise "I".

L131, human pressure (HP): The abbreviation HP is already used to refer to "Harvest Productivity" (L111), and I did not find the need for "human pressure" to have an abbreviation. Omit it?

Figure 1: I wanted to see the map with higher resolution (a larger map) to check the survey locations used in this study's analysis. Can you include a more extensive map as a Supplementary Figure to show the locations with national borders?

L173, (Fig. S2): A mistake to be "Fig. 2"?

L231, "US\$ 2.1 ... billion, based on the market value of beef in 2024": Isn't it overestimated? With my limited experience in the Colombian Amazon, wild meat is much cheaper than beef.

L436, Correct "Table X".

L443, "geofenced": typo

(Remarks on code availability)

The code is not provided in the reviewer dataset.

Referee #3

(Remarks to the Author)

KEY RESULTS

This article summarizes the geographic extent, volume, and nutritional value of wild meat harvested across the nine countries of the Amazon. Its most notable feature is an extensive dataset covering 560 rural locations, including 341 with primary data, and incorporating spatial analysis of consumption patterns. The article also evaluates the nutritional significance of wild meat for rural inhabitants and provides detailed insights into the consumption patterns of the twenty most consumed species. By combining this dataset with modeling, the article offers a large-scale, regional perspective on wild meat consumption in the Amazon, which is unparalleled given that most previous studies have been localized. This analysis helps identify the challenges involved in conserving hunted species in the region.

ORIGINALITY AND SIGNIFICANCE

This study is unique in its geographical scope and dataset size, enabling a comprehensive assessment of the importance of wild meat consumption for the nutrition of the rural Amazon population. By drawing from a broad sample of locations, it also highlights regional differences in wild meat consumption and explores the potential impacts on hunted fauna. No comparable studies exist for this region, and perhaps not globally.

DATA AND METHODOLOGY

The dataset consists of primary and secondary data from articles, theses, and various reports, spanning nearly six decades. As a result, the data vary in characteristics and potential accuracy, which is typical for evaluations of this scale. I believe the authors should more clearly emphasize this variation, as some methodological explanations are currently presented as if they were derived from primary sources (see, for example, lines 447-463).

In my assessment, the article has no major flaws that would prevent its publication. However, one significant limitation is the exclusion of urban wild meat consumption, which presents a partial view of overall consumption in the Amazon. That said, the article appropriately focuses on the population subset for whom wild meat consumption is most important. Moreover, it is important to note that the absence of urban consumption data is a common issue for studies in the region, largely due to the illegality of the practice in Brazil. While some studies address urban consumption, they are few in number and may be

biased, potentially impacting the article's calculation of total consumption.

Another methodological concern relates to the visualization of findings in the maps, which are a central feature of the article. Many maps are difficult to read, even with a magnifying lens, and in some cases remain unclear despite magnification. I thus recommend improving the figures and the text within them, as these maps are likely the most critical aspect of the article.

APPROPRIATE USE OF STATISTICS AND TREATMENT OF UNCERTAINTIES

The methods are sufficiently detailed and, in general, well-described, but I believe two aspects could be improved.

First, the article utilizes both primary and secondary data from various sources, which likely differ in quality. While some brief notes on limitations are provided, I found them insufficient. I recommend including a section that discusses the limitations of the dataset more thoroughly, particularly in terms of data quality, and spatial and temporal variation. This could be incorporated into the main text or, alternatively, in the appendix. Additionally, a more in-depth discussion of the potential limitations arising from the exclusion of urban data would be valuable.

Second, in the methods section, it is not entirely clear whether all procedures were applied to both the primary and secondary data (e.g., but not limited to, lines 563-576). Clarifying the distinction between procedures applied to each data source would be beneficial.

Third, I suggest the authors consider providing the data analysis coding to enhance the reproducibility of the results.

A final note: I am not familiar with random forest models, so I am unable to evaluate this aspect properly.

CONCLUSIONS

The results describe wild meat harvesting, the most commonly exploited species, and the nutritional contribution of macro- and micronutrients in the Amazon. While the conclusions that deforestation and population growth reduce access to wild meat are obviously not new, the overall assessment of the importance of wild meat in the region offers a novel perspective. The results also provide a clearer understanding of the immense challenges involved in controlling wild meat consumption in the region.

SUGGESTED IMPROVEMENTS

The authors have compiled an impressive spatial dataset that is both intriguing and likely to inspire further research, and they should be commended for this achievement. However, I have three main concerns, besides some more specific suggestions.

First, the unique contributions of this dataset need to be more clearly highlighted in the text. While the importance of wild animal hunting for local communities is emphasized, the article does not sufficiently explain how this dataset advances our understanding of Amazonian wild meat consumption or provide novel insights beyond existing local studies. I believe it is crucial to clarify how this dataset enhances our understanding of issues related to wild meat consumption and nutritional effects that were previously unclear with local-scale data.

Second, although the discussion focuses on the role of wild meat in achieving Sustainable Development Goals, the most intriguing aspect of this dataset is its ability to reveal regional variability and the factors that explain it. This aspect is only minimally explored in the article. The numerous maps presented are only briefly interpreted, and the text does not fully reflect the extensive work undertaken. The authors could consider briefly summarizing the relevance to the Sustainable Development Goals in a short paragraph, and use the remaining space to provide a more in-depth discussion of the spatial variation.

Finally, the authors should more thoroughly acknowledge the limitations of their results and discuss how these limitations may affect the interpretation of their findings. Specifically, they could include remarks in the discussion or conclusions on the implications of missing urban consumption data. While I understand the challenges of obtaining accurate data due to the illegality or regulation of hunting in much of the region, it would be valuable to highlight how this limitation might lead to an underestimation of the data.

If space permits, another important aspect to highlight is the new research directions or questions suggested by the article's conclusions.

SPECIFIC COMMENTS

Title: The title suggests that the article's primary focus is on food security and ecosystem health. However, the article primarily discusses estimates of consumption and the contribution of macro- and micronutrients. The connection between these results and ecosystem health is not clearly established in the text and does not seem to be a central aspect of the article. The text should either clarify this connection or the title should more accurately reflect the article's content.

Additionally, one of the article's main contributions is the spatialization of the data, which is not adequately reflected in the title.

Page 3, Lines 93-101: Could you also highlight how the study contributes to our scientific understanding or current debates? Specifically, how does this spatial approach advance our existing knowledge?

Page 3, line 120: Does this number represent the total across all decades combined, or is it the current figure?

Page 3, line 126-129: this part is not very clear.

Page 3, Line 131: Is HP the acronym for both hunter productivity (see line 111) and higher predicted human pressure?

Page 4, Line 146: Are these estimates derived from literature specifically focused on the Amazon?

Page 4 lines 147-148: What specific aspects are affected by this limitation? It is unclear to mention several problematic factors without explaining how they may impact the results.

Page 4, line 150: ...in according or ..., according?

Figures (all): I believe the font size in most of the figures in the main text and supplementary material should be larger. I had

to use a magnifying glass to read them, and for some graphs in the appendix, even the magnifying glass was not sufficient. Page 7, Lines 209-224: This section on the historical reliance of indigenous people seems out of context and does not provide sufficiently relevant information. Do you have current data to assess whether indigenous hunting patterns vary by cultural group or in comparison to non-indigenous people? Additionally, this text appears under the section titled 'Geographic Variation in Hunted Species,' but its relevance to the topic is unclear. For instance, it is not evident whether the authors intend to argue that this variation is culturally determined. While there is a map and data related to this issue, the article does not adequately explore them.

This section relies on the literature, but its connection to the dataset or study results is unclear. From what I understand, there is a spatial dataset indicating indigenous language groups (as shown in Extended Data Figure 3). However, it is not evident how this dataset was utilized or how it contributed to the findings. I would expect to see actual results regarding cultural diversity.

Page 9, Figure 4: unclear the source of the requirements used to calculate this.

Page 9, Figure 4: This map shows the exact shape of certain indigenous territories in the Eastern Amazon. I am curious about its relevance to the results. This is just one example of spatial differences that the article's findings do not explore.

Page 10, Line 248-269: How do the results compare to previous studies and other regions worldwide? If the authors could summarize these references in relation to the Sustainable Development Goals, it would allow for a more direct comparison of their findings with existing literature—something that is currently lacking.

Page 10, Line 262: SDGs instead of Sustainable Development Goals?

Page 10, Lines 267-269: I believe that discussing these regional differences is more compelling and important than referencing the various Sustainable Development Goals.

Page 11, Line 285: Are you referring to "biodiversity" or "diversity" indicator"? Additionally, this phrase is quite convoluted.

Page 11, Lines 298-299: How does this translate into the territorial percentage of additional deforestation, for example?

Page 11, Line 303: Instead of "parks," it would probably be better to use "protected areas, including extractive reserves".

Page 11, Line 312: Is there a comma missing after "densities"?

Page 11, Line 314-315: This point sounds important, but it would be more impactful and helpful if you could specify how it contributes.

Page 11, Line 316-329: How does this relate to the findings? It might be beneficial for the authors to concentrate more on exploring regional differences, for instance, as this could provide more insight than including statements that seem unrelated to the findings or lack specific evidence.

Page 14, Line 431: Why is the term "traditional hunting dataset" used? Typically, "traditional" refers to specific small-scale societies with a long history in the same location and distinct cultural practices. Since the data also include agriculturalists, who are often more recent inhabitants, this term seems misleading. Also, earlier in the article, the authors refer to "Amazonian peoples" and not traditional people (see line 88). Additionally, they should clarify whether they are specifically referring to rural small-scale communities and explain which types of rural communities are excluded from/included in the sample.

Page 14, Line 436: Which is Table X or where is it? Moreover, there are 199 articles (please check the mistake in references 66 and 67, as they appear to be the same).

Page 14, Line 446: Could you briefly explain why data on wild meat trade and urban consumption were excluded?

Page 15, Line 455: It is probably better to refer to it as face-to-face interviews or systematic interviews. Some authors use the term "questionnaire" exclusively for self-administered protocols.

Page 15, Line 449-450: How do you ensure accuracy with data dating back to 1965? Additionally, how do you guarantee that the data are both culturally appropriate and scientifically rigorous, particularly when using secondary sources? It would be helpful to clearly distinguish between the primary and secondary data, as combining them without explanation may lead to confusion. Additionally, you might consider avoiding adjectives unless you provide an explanation for them.

Page 15, Line 462: Could you clarify if this refers only to primary data? If not, how was consistency ensured? Were the same protocols applied to all data, and were trained surveyors involved? It would be helpful to provide more detail on the methods used, as simply stating that consistency and accuracy were ensured may not be sufficient.

Page 16, Lines 515-546: In some instances, the titles in Extended Data Figures 2 and 3 do not match exactly those in the main text of the article. For example: (i) "Annual Net Primary Productivity" – the word "Annual" is missing in the figure; (ii) "Proportion of Flooded Forest" – the figure shows "Proportion of Flooded Area" (please verify all discrepancies). Additionally, the order of appearance is inconsistent between documents.

Page 17, Lines 539-540. The explanation in the methods section is clear; however, as I read through the article, it is not entirely clear how the data on indigenous language zones was applied, as the relevant results are difficult to identify. It would be helpful if the authors provided more detailed information on the variations in human populations associated with hunting. In several sections on this topic, the information seems to rely more on literature than on the authors' own data.

Page 17, Lines 557-562: I understand that you aimed to minimize the issue of temporal variation. However, I believe it would be important to highlight how this limitation may have impacted your results. While this is perhaps the most apparent limitation, there is no information provided on how it may have introduced bias into the findings.

Page 19, Line 642: Do you mean the average per species or the average across all species? This distinction becomes clear only later in the article.

Page 19, Line 644: A comma is missing after "Protein".

Page 20, Line 674: The text lacks a smooth transition between the two paragraphs and could benefit from a connecting phrase to create a more cohesive flow.

COMMENTS TO OTHER DOCUMENTS

LIST OF LITERATURE CONTAINING HUNTING DATA IN RURAL ETC.

Check references 66 and 67: there is only one reference.

Correct reference 100: instead of Luz C. the correct is Luz AC. (sorry, I know her... ;)

EXTENDED DATA - ALL: I found it difficult to read the text on most maps with the naked eye, for instance in Figures 5, 6, 7, and 9. Could you increase the caption size?

EXTENDED DATA FIGURE 3: It is unclear what purpose this was intended to serve or how it fits into the analysis.

EXTENDED DATA- TABLE 1: The table (first column) is not in alphabetical order. Could you please check that?

Extended Data Table 9: In Table caption is written: "Estimation was based on 256 observation" . There is an S missing in "observation".

EXTENDED DATA TABLE 11: The title is "Daily values of Estimated Energy Requirements "for energy". Is this correct?

REFERENCES

The previous literature is properly referenced.

CLARITY AND CONTEXT

The abstract is clear, well-written, and provides an appropriate summary of the article. However, if possible, the authors might consider emphasizing the importance of understanding urban wild meat consumption as well.

Best wishes,
Carla Morsello

(Remarks on code availability)

The code was not provided.

Referee #4

(Remarks to the Author)

A: Summary of the Key Results

The paper presents a comprehensive study on sustainable wild meat harvesting in Amazonia, emphasizing its significance for nutritional security and ecosystem health. It quantifies the geographic extent, volumes, and nutritional value of wild meat harvested in Amazonia, demonstrating its critical role in meeting the dietary needs of remote rural populations. The study also highlights the negative impacts of deforestation on wild meat productivity and proposes the urgent need to preserve traditional food systems to achieve the Sustainable Development Goals (SDGs). The results indicate that wild meat plays a critical role in the nutritional security of rural and Indigenous communities in the Amazon, contributing significantly to their dietary intake of proteins and micronutrients like iron and B vitamins. Deforestation and proximity to urban areas, however, are linked to declining wild meat productivity, emphasizing the need for sustainable practices.

B: Originality and Significance

The paper addresses a crucial and under-explored aspect of environmental conservation and food security, specifically in the Amazon region. While the study's scale and methodology are significant contributions, the topic itself is not entirely novel as similar studies have addressed wild meat consumption and its implications in other contexts. However, the large-scale dataset and the detailed spatial analysis make this study particularly valuable. The work significantly contributes to the discourse on sustainable resource management and conservation in Amazonia. While the subject of wild meat has been explored, the integration of sustainability and SDG impacts in the Amazon context is original.

C: Data & Methodology

The methodology is robust, with a large dataset spanning several decades and a substantial geographic area. It comprises data on over 441,000 animal kills across 560 rural localities. The methodology, using Random Forest models to spatially predict hunting metrics, appears sound. However, details about some data sources (i.e., specific geographic variability) could be expanded. The use of both primary and secondary data enhances the study's validity, though more granularity in the source of secondary data (e.g., author verification of older records) could improve reliability. The use of Random Forest models to predict hunting metrics and the comprehensive analysis of environmental and anthropogenic variables is appropriate and well-executed. The data is of high quality, and the presentation is clear, with thorough explanations of the methods used. However, more detailed information on how data from different sources were standardized and integrated would enhance the transparency of the approach.

D: Appropriate Use of Statistics and Treatment of Uncertainties

The statistical methods used in the study are appropriate for the data and research questions. The treatment of uncertainties is well-managed, particularly through the use of confidence intervals and quantile predictions. The authors acknowledge the limitations of their spatial models and provide a balanced discussion of the uncertainties inherent in their estimates.

E;

E: Conclusions: Robustness, Validity, Reliability

The conclusions drawn are robust and supported by the data. The study effectively links wild meat harvesting to broader issues of nutritional security and ecosystem health. The discussion is well-grounded in the results, and the implications for policy and conservation are clearly articulated. However, the paper could benefit from a more explicit discussion of the potential for generalization beyond the Amazon region. Additionally, expanding on alternative conservation measures or sustainable practices could make the study more actionable.

References

The study cites a wide range of relevant literature, including both recent and seminal works on biodiversity, ecosystem services, and human nutrition. However, further integration of Indigenous knowledge sources or community-led research could strengthen the contextual relevance.

Suggested Improvements

1. Standardization of Data Sources: Provide more details on how data from various studies and regions were standardized for integration into the analysis.
2. Spatial Resolution: Consider discussing the implications of the 10×10 km spatial resolution for local-scale conservation efforts.
3. Policy Implications: Expand on the policy recommendations, particularly on how to implement sustainable wild meat harvesting practices across different regions with varying ecological and socio-economic contexts.
4. Impact of Climate Change: Include a more detailed analysis of how climate change might further impact wild meat availability and ecosystem health in Amazonia.
5. Explicit mention of obtaining Free, Prior, and Informed Consent (FPIC) from Indigenous communities.
6. Clarification is needed on data collection protocols in areas with significant geographic or cultural variability.
7. More discussion on the limitations of model-based predictions is necessary, especially in heterogenous environments like the Amazon.

Clarity and Context

The paper is generally clear and well-structured. The abstract provides a succinct summary of the research, and the introduction effectively sets out the context. However, some sections, particularly those detailing the methodology, could be streamlined for better readability. In addition, the conclusion could more clearly reiterate the main findings and their implications for policy and future research.

Overall, this paper is a significant contribution to the field of environmental conservation and sustainable resource management in Amazonia. With some refinements, it could be an even more powerful document for influencing policy and conservation practices.

Ethics

The study demonstrates a solid effort in ethically working with Indigenous communities across the Amazon region. The inclusion of Indigenous collaborators, such as hunters from multiple Indigenous groups, suggests an approach aligned with respectful and collaborative research. Data collection methods appear to prioritize community engagement, involving local researchers and community members. The monitoring schemes seem to have been designed with cultural sensitivity, which is crucial for maintaining ethical standards. However, the paper would benefit from explicitly stating whether free, prior, and informed consent (FPIC) was obtained and if there were specific protocols to ensure how Indigenous knowledge was respected and protected. Including such details would strengthen its ethical grounding.

The paper would benefit from a more explicit discussion of the ethical standards and guidelines followed during the study. For instance, it should clarify how informed consent was obtained from participants and whether any specific ethical approvals were sought from local or national bodies overseeing research involving Indigenous communities. In addition, the paper could address how the data will be used and shared in a way that benefits the communities involved, ensuring that they are not merely subjects of study but active participants in the research process.

(Remarks on code availability)

Referee #5

(Remarks to the Author)

Sustainable Wild Meat Harvesting in Amazonia: A Keystone for Nutritional Security and Ecosystem Health

The Amazonia is by far one of the most unique tropical forest regions on Earth, harboring high species and rich indigenous cultures. For thousands of years, documented interactions revealed that rural indigenous people have been known to depend on meat from animals in the forests as a key source of food, and most have done so in a sustainable manner. Yet, this relationship, in the form of traditional food systems, may be disrupted in recent times due to globalization, urbanization, and climate change impacts. In particular, deforestation may affect the supply and availability of wild meat to the rural people who are dependent on them for essential nutrients.

The authors are right to call for attention to human and forest ecosystem health, and hunger issues (all key Sustainable Development Goals) all highly relevant to the Amazonian peoples. Drawing on an impressive dataset collected over 40 years making up of over 400,000 individual animals harvested across at least 500 rural localities contributed from an extensive collaboration network of researchers and managers, the authors have estimated, through biological and spatially explicit modeling, an annual extraction of 0.33 Mt of edible wild meat, largely from 20 hyperdominant taxa. While this amount of wild meat is sufficient to meet some of the key nutritional requirements (both macro and micro) for rural people, high extent

of deforestation in Amazonia may lead to a 60% decrease in wild meat productivity, jeopardizing both the wildlife and the traditional food systems known to preserve the Amazonian way of life for millennia.

Yet, this potentially ground-breaking and profound piece of work (and it is well-written!) hinged on the assumptions that the underlying data and models, for which the key results were estimated, are reasonable. I am outlining some key issues below that I would like the authors to address so the results could be more convincing, but it might take some effort to sort them out.

Being a large-scale spatially explicit assessment, there is a need to reasonably estimate the wild animal harvest rates and offtake levels. With this forming the basis of how wild meat contributes to food security, it becomes extremely crucial to get the initial modeling right. My biggest issue is with how the metrics (TSOP and HHR) were estimated. The authors used Random Forest models to spatially predict the metrics with twelve environmental and anthropogenic variables but these variables may be independent from the time the surveys were undertaken. The forest cover data is a good case in point. Say for example that if harvested data from the surveys stretched over two decades, and studies have shown that Amazonian forest cover has been changing rapidly. How is it possible that the harvested animal rate is estimated without accounting for the temporal changes to the numerous environmental and anthropogenic variables?

The above issue is also directly related to the key finding that higher wild meat offtake is predicted mostly around major Amazonian cities. All other spatially and demographically related findings are conditional on the assumptions made for the spatial models. While the authors acknowledge the limitations inherent to the study, particularly regarding the spatial coverage of their data (lines 147-148), they also need to consider the temporal coverage of their digital spatial variables and models. Can the authors address these issues (including having sensitivity analysis and simulations) to improve their models and outcomes? If not, the authors would need to provide an acceptable argument to defend their current approach. In my opinion, this is the most defining issue for this manuscript that would need to be resolved.

What is so interesting about this region is the presence of diverse indigenous cultures and their hunting practices, and target and preferred species. The authors did a good job in highlighting how the Amazonian Peoples have a deep understanding of the Amazonia ecological landscape. But how deforestation may pit tribes (and cultures) against each other in order meet food security, which may indirectly increase conflicts among neighboring areas/tribes in the region? What are the risks of such conflicts under different scenarios? It is plausible that the diminishing supply of traditional food may cause shifts in diet which may increase competition and conflicts for shared species (even hyperdominance ones), hence leading to unsustainable offtakes and food system compared to the past. As stated in lines 222-224, the "dynamic interplay has led Amazonian Peoples to develop diverse and sophisticated systems of hunting rules, species preferences, and taboos, resulting in varied hunted assemblages across hundreds of cultures", but this may be potentially disrupted by deforestation and climate change issues with time. While I understand that modeling under different deforestation and climate change scenario may be outside of the scope of the manuscript, some key discussion, at the minimum, on such potential conflicts may be warranted.

Other comments include the following:

Line 126: Is HP the same as wild meat offtake?

Line 130: Why not use per capita harvest?

Figure 1. Perhaps use a different color legend for each panel since the ranges are quite different.

Figure 2. Appreciate the animal drawing from an indigenous artist and author!

Figure 3. For better visual effect on relative HP, I would suggest using a standardized color scale across all species. The point here is to offer cross-taxon comparison. While the present illustration reflects well on the spatial distribution of HP for individual species, it is not ideal for cross-species comparisons.

Lines 266-267. Is there any change in terms of amounts eaten per person observed over the course of 60 years or a lifespan and gender?

Lines 282-284. Is the point that even hyperdominant species are not spared from the climate change impact a major issue? Or is this quite unlikely considering that they are hyperdominant because they are adaptable to environmental changes, including deforestation and climate change?

Lines 290-291. The bit on cultural erosion impact is not that obvious here.

Lines 294-299. This is such an important point, and a huge paradox. Obviously, the agricultural and ranching impacts, if modeled spatially, could be really useful for future management and planning, and impact assessment.

Lines 308-311. Are these from Goal 15 the only targets? How about other goals such as health, equity etc?

Line 436. Table X?

Lines 438-444. Re: ATH dataset. How do we know the hunted taxa are correctly identified? Surely, some primary/secondary data are more accurate than others. Do the authors take data quality and accuracy into account during the modeling?

Line 448. Where can we find the documentation of the hunting monitoring schemes? Can we use this to evaluate the quality of the data?

Line 468-469. Is the estimation dependent on the time of the survey? This is similar to the key issue highlighted above.

Line 513. Again, are these time dependent or independent to the survey? They do not seem so but perhaps the authors can explain.

Lines 557-562. I am unsure if the use of data from 2005, which is the averaged year of ATH dataset, is the most rigorous approach. The variation from the time dependent survey is an important issue, which have not been addressed directly.

(Remarks on code availability)

Version 1:

Reviewer comments:

Referee #1

(Remarks to the Author)

Thank you for carefully addressing, in great detail, each of my comments and questions. I do not have any additional questions.

(Remarks on code availability)

Referee #2

(Remarks to the Author)

Thank you for seriously considering my comments on the previous version of the manuscript; I think it is now largely improved by rigorously addressing the temporal variation of the data. I have one question and one suggestion remaining, however.

First, the authors defined the HHR (hunter harvest rates) as “the average number of animals hunted per hunter per day in each locality”. However, I would like to confirm whether HHR from your primary and secondary datasets are calculated to include days when hunters did not engage in hunting in the denominator. If I understand well, HHR of a single locality should be calculated as: $(\text{The Total Number (or Biomass) of Animals Observed to be Harvested}) / [(\text{Number of Hunters Observed}) * (\text{Number of Days Surveyed})]$. To accurately estimate the annual harvest rate and available wild meat, the Number of Days Surveyed must include not only the days hunters entered the forest for hunting but also the days when they did not engage in hunting (e.g., days hunters stayed home or engaged in other activities) because hunters are unlikely to go hunting everyday. Otherwise, the estimates would be overestimated. I did not find clear explanations to address this question. So, could you add a brief sentence to explain how the authors calculated HHR explicitly? Similarly, I imagine that most secondary data employed ad libitum sampling in terms of the choice of hunters. Consequently, the researchers of the secondary data may have tended to choose “frequent hunters” who engage in hunting relatively frequently. If this is the case, HHR would be overestimated even if it is correctly calculated due to the lack of data on non-frequent hunters. This point may be discussed as a limitation of the study and a future recommendation for producing hunting data.

Second, I feel “The fate of wild meat food systems in Amazonia” is too long and speculative. This section mostly discusses the sustainability of the current wild meat hunting system in Amazonia. However, this study estimated only the current state of local hunting but did not examine trade, consumption, or animal abundance. So, this study has limited resources for discussing sustainability. I understand that this revision is based on the response to Reviewer #1’s comment (rebuttal file, pp. 7-8), but personally, I am opposed to this addition. I believe that this addition weakens the arguments of the manuscript. I suggest reducing the section and creating a supplementary discussion by merging some of the discussions in this section with the responses on pp. 7-8.

HONGO Shun
Research Institute for Humanity and Nature

(Remarks on code availability)

Referee #3

(Remarks to the Author)

This article presents a large-scale analysis of wild meat consumption across the Amazon, drawing on data from 560 rural

locations—341 of which are based on primary data. It includes spatial analysis, models consumption patterns, and assesses the nutritional importance of wild meat for rural populations. By combining this dataset with modelling, the article offers a regional perspective on wild meat consumption in the Amazon that is unparalleled, given that most previous studies have been localized. This analysis also helps identify the challenges involved in conserving hunted species in the region. The authors have appropriately addressed my comments and have notably: highlighted the novelty of the study; improved the description of spatial patterns; enhanced the figures; and clarified the procedures applied to both primary and secondary data. I am therefore satisfied and would like to compliment the authors on this impressive paper. There are a few minor typos I have found below.

Minor points

Line 90: Delete the “the” here: “the” Sustainable Development Goals (SDGs)

Line 189: “targeted” instead of “target”

Line 195-197: Review this phrase: “In contrast, those living in upland terra firme forests, which cover 86% of Amazonia, yield more tinamous, trumpeters, wood quails, guans, and armadillos are more commonly hunted”

Line 583: Unclear: “different local contexts local-scale variations”?

Lines 621-622: I did not understand this phrase: “Although the spatial process of excluding urban areas probably excludes rural or peri-urban inhabitants”. How does excluding urban areas lead to the exclusion of rural inhabitants?

Carla Morsello

(Remarks on code availability)

Referee #4

(Remarks to the Author)

I appreciate the thorough and detailed reply from the authors regarding the ethical considerations raised in my initial review. It is evident that significant thought and effort have been dedicated to addressing these issues.

The response provides extensive clarification on the participatory nature of data collection, the longstanding collaborations with Indigenous and Traditional Peoples, and the ways in which community members have been involved in the design, implementation, and feedback processes of the monitoring initiatives. I particularly welcome the authors’ acknowledgment that formal institutional ethics approvals, while important, may not be sufficient in isolation, and that they have sought to ensure ethical alignment through community involvement and leadership.

I also appreciate the authors’ positive response to the concern regarding data-sharing governance. The establishment of a formal committee including representatives from COIAB, CNS, and RedeFauna to oversee data access is a welcome and important step toward ensuring Indigenous data sovereignty and shared decision-making.

Finally, the proactive submission of the manuscript to Indigenous representative organizations for independent evaluation, along with the supporting letter from CNS and the pending feedback from COIAB and its regional partners, further demonstrates a strong commitment to ethical and culturally sensitive research practices.

In conclusion, I believe the authors have made a sincere and meaningful effort to respond to the concerns raised.

(Remarks on code availability)

Referee #5

(Remarks to the Author)

I am one of the original reviewers for the manuscript and I am happy to provide further inputs to the revision.

By and large, I commend the authors for providing high quality responses and edits to the issues that I have raised. They have made a serious attempt to improve the manuscript.

However, there remains two issues that I feel need to be directly addressed in the manuscript. They are regarding the following.

1) "What is so interesting about this region is the presence of diverse indigenous cultures and their hunting practices, and target and preferred species. The authors did a good job in highlighting how the Amazonian Peoples have a deep understanding of the Amazonia ecological landscape. But how deforestation may pit tribes (and cultures) against each other in order meet food security, which may indirectly increase conflicts among neighboring areas/tribes in the region? What are the risks of such conflicts under different scenarios?"

The authors responded appropriately to my comment. However, it would benefit the readers if the authors could include their responses into the main text.

2) "It is plausible that the diminishing supply of traditional food may cause shifts in diet which may increase competition and conflicts for shared species (even hyperdominance ones), hence leading to unsustainable offtakes and food system compared to the past. As stated in lines 222-224, the "dynamic interplay has led Amazonian Peoples to develop diverse and sophisticated systems of hunting rules, species preferences, and taboos, resulting in varied hunted assemblages across hundreds of cultures", but this may be potentially disrupted by deforestation and climate change issues with time."

Similarly, I would really appreciate it if the authors could include their responses into the main text. Compared to the responses in the above comment, this is much less developed. If the authors could work on a better response to my comment and include it in the main text, that would really benefit the readers.

Otherwise, this has been a very high quality manuscript and an important topic to work on.

(Remarks on code availability)

Responses to comments from Referees of the Antunes *et al.* article

Editor comments

Importantly, we asked for advice for an expert on the ethics for dealing with such datasets (R6). Though this referee did not provide formal comments, we have synthesized their advice in a list of requests. It is important that these are addressed, as we will not be able to proceed to publication without the relevant information and consent.

Additionally, you will notice that both R2 and R5 ask that you explore the temporal variation in your data and note different factors that may influence the results (e.g. trade and outside consumers, R2, conflict, R5). Similarly, R3 notes that you do not consider urban wild meat consumption. We are not sure whether it is possible to address this latter concern, but do ask that you expand your analyses to address these points and revise your manuscript so that it clearly reflects the limitations of the dataset.

Response

In this new version of the manuscript, we have expanded the traditional hunting database, the digital spatial co-variables to improve the models. These made our claims and discussions even stronger. To do so, we had to include additional co-authors who contributed significantly at key stages of this new version (including obtaining the data and spatial variables). Below, we provide brief responses to the main points raised by the reviewers and summarised by the Editor.

1. Collaborative research and FPIC compliance

We offer a comprehensive justification for the collaborative nature of our hunting monitoring initiatives in Indigenous and traditional Amazonian communities, where we have worked for many years, often decades. As noted in our response to Reviewer 6, these initiatives extend beyond purely academic objectives. They are developed within the territories with broad local participation and aim to generate data for the sustainable management and guarantee rights over natural resources to local people. The use of these data for research is agreed upon between researchers and community leaders/representatives, several of whom are co-authors of this paper.

In response to Reviewer 6's concerns, we provide clear explanations of the ethical procedures adopted by us, and completed even further the supplementary material with information on ethical boards' approval or legislation followed (when government-led) in our initiatives.

In addition, as this is the first article we are aiming to publish using the data, and we envision this to be the basis for subsequent ones, we are committed to organise an Evaluation Committee that will oversee and evaluate the ethical issues of future publications with the dataset

We have already submitted the manuscript to organizations with broad Indigenous and traditional community representation in the Amazon to assess whether the article's scope and methods ensured that the studied communities had full autonomy to provide or withhold Free, Prior, and Informed Consent (FPIC) regarding research affecting their territories,

rights, and ways of life—while respecting their culture, self-determination, and equity. So far, we have received a letter of support from the National Council of Extractivist Populations (CNS) (attached in this submission) representing traditional non-Indigenous peoples of the Brazilian Amazon (which covers around 60% of the Amazon). We are awaiting feedback from the Coordination of Indigenous Organizations of the Brazilian Amazon (COIAB), which represents Indigenous peoples in the region. COIAB has also agreed to consult with AIDSESEP (Peru) and FOAG (Guyana) to evaluate the manuscript's methods and potentially join the Evaluation Committee.

2. Improved analytical approach regarding spatiotemporal variations

To address the spatiotemporal limitations noted by three of the Reviewers, we applied a new analytical approach. Although we had already considered this issue in the first version, the current revision tackles it more rigorously. We re-ran all analyses using a more robust set of spatial covariates, incorporating shorter time windows as predictors where available. This did not change the results substantially, but rather greatly enhanced our claims with even stronger patterns identified.

3. Enhanced cultural variables

Following Reviewer 4's suggestion, we refined and expanded the cultural spatial variables. In addition to the Historical distribution of Indigenous languages, we now include *Current distribution of Indigenous and non-Indigenous peoples* and *Current distribution of language families as variables*. The latter variable proved particularly important in the Harvest Productivity random forest models. We also expanded the discussion on how cultural factors influence traditional hunting patterns in the Amazon, as suggested by the Reviewer.

4. Urban wild meat consumption: justification for exclusion

We maintained our decision not to include urban wild meat consumption data, as it falls outside the article's focus on traditional food systems. However, we added a justification in the Methods section and emphasised our findings on urbanization's effects, including peri-urban consumption.

5. Code Availability

Due to technical challenges during the first review round, including limited experience with the Ocean Code platform and its computational constraints (30 hours/month processing limit and 20 GB storage), we could not fully deploy the original code. Moreover, we were not able to use some important spatial analysis packages and had to change part of the original code. Given the spatial nature of our analyses, which often require several hours or even days to complete, we were unable to run and debug all the code within the platform's limitations during the first review stage. For this revision, we are working on the Ocean Code to make available the core *Random Forest spatial models* for HHR (Hunter Harvest Rate) and TSOP (Taxon-Specific Offtake Proportion). However, we are also prepared to share the entire code and data directly with reviewers if required.

Referee #1

General Comments

This study reports on an immense data set covering the entire Amazon basin, through a combination of primary and secondary sources.

We thank the reviewer for all the insights, which greatly improved our work.

Specific Comments

How does your manuscript address ecosystem health?

In this new version, we replaced the title with “Healthy forests safeguard traditional wild meat food systems in Amazonia”. Ecosystem health, environmental sustainability, and food security are closely interconnected. Considering that ecosystem health refers to the conditions in which ecological processes function effectively to sustain biodiversity, productivity, and resilience over time, we show how deforestation and urbanization negatively affect harvest productivity and the availability of wild meat. Food production increasingly relies on large-scale conversion of natural ecosystems to intensive monocultures or pastures, but effective management of resources within natural ecosystems offers the potential for a more sustainable approach. In tropical developing countries, the first scenario often leads to extensive tracts of intensive farmland devoid of biodiversity value and with economic benefits only for a wealthy minority. Equalizing development opportunities and eliminating poverty may instead be more effectively achieved through the considerate exploitation of natural resources at a sustainable level, while also reducing tropical deforestation and bringing nutritional health benefits to rural and urban consumers. Agriculture in the Brazilian Amazon, for example, relies on converting natural forest into cattle pastures and soya cropland, the latter exported for livestock feed in China, North America, and Europe. These intensive agro-pastoral land uses may boost global food supplies but erode biodiversity and forest ecosystem services, fueling greenhouse gas (GHG) emissions and concentrating wealth into large-holders to the detriment of most rural poor. Global and regional demand for food can impact the environment, causing biodiversity loss and threatening Indigenous Peoples and Traditional Communities. In contrast, local food systems rely on biodiversity conservation, ecosystem health, and environmental sustainability. Hunting undoubtedly affects animal populations. In particular, hunting for trade, linked to market dynamics, is more prone to incur more severe depletion than the traditional hunting for the subsistence of Indigenous people and local communities in Amazonia. In addition to its direct effects on animal populations, hunting can also lead to the dismantling of the ecosystem by disrupting the ecological services that animals perform, including the consumption and dispersal of seeds and interspecific interactions between animals, such as predation, which can lead to changes in the structure of the plant and animal communities. Consequently, unhealthy animal populations and ecosystems can weaken the human populations that rely on these resources. Hunting in tropical forests inherently depends on forest conservation. In this context, managing wild protein potentially provides a promising opportunity to develop modes of food production that, in contrast to commercial beef, support local livelihoods, improve diet diversity and health, reduce income and social inequality, and relieve pressure on ecosystems threatened by agricultural

expansion and intensification. Our findings highlight the importance of protein sources from wildlife, which is strongly connected with ecosystem health.

We explain this relationship in new texts added to the paper as follows:

New text (Lines 84-92):

This paper examines the links between wildlife, human nutrition, and traditional food systems in Amazonia. We assess key indicators of ecosystem integrity—including species diversity, biomass availability, and environment condition. Our findings show that deforestation and urbanisation reduce wild meat productivity and simplify the composition of hunted taxa, ultimately threatening traditional food systems and the nutritional well-being of Amazonian Peoples. Protecting Amazonia is therefore essential not only for conserving biodiversity but also for sustaining ecosystem functions¹¹ and achieving multiple global the Sustainable Development Goals (SDGs) under the United Nations Convention on Biological Diversity, including those related to ecosystem and human health.

New text (Lines 240-266):

Threats to wild meat food systems

Our spatial analyses revealed a negative correlation between habitat loss and HP in Amazonia. In regions with over 70% deforestation, covering 0.80 M km², we observed a 74.7% decline in the number of individuals harvested per hunter and a 67.3% reduction in harvested biomass per hunter (Extended Data Fig. 12).

*Available wild meat per rural inhabitant was significantly lower in communities with (1) higher numbers of rural inhabitants, (2) closer proximity to urban centres, and (3) greater deforestation levels (Extended Data Fig. 13). In these more degraded areas, our TSOP spatial estimates indicate a shift in hunting patterns, with ecological generalists such as the nine-banded armadillo (*Dasypus novemcinctus*), capybara (*Hydrochoerus hydrochaeris*), guans (*Cracidae*, *Penelope spp.*), and pigeons (*Columbidae*) becoming proportionally more hunted than in better-conserved forests (Extended Data Table 8). In contrast, large atelid primates such as the woolly, spider and howler monkeys, which are quite vulnerable to the synergistic effects of deforestation and overhunting, are much less hunted in degraded areas.*

Large-scale agriculture, land grabbing, logging, mining, and infrastructure development have led to deforestation³³, increasingly undermining Amazonian People's reliance on biodiversity. The combined effects of deforestation and wildlife overharvesting have produced simplified animal assemblages, with lower species richness and fewer large-bodied species³⁴. These pressures are further compounded by recent climatic changes, including more frequent floods, droughts, and large-scale fires³³, all of which threaten both habitats and many key hunted taxa³⁵. We recommend that further studies should investigate the future impacts of deforestation, wildfire, and climate change on animal populations and on the supply of wild meat.

We also predicted higher offtake and lower HP near urban centres, raising concerns about the sustainability of hunting in these areas. High meat demand in densely populated peri-urban areas, coupled with declining wildlife populations³⁶, may shift rural diets toward cheaper domestic meats like chicken³⁷, which generally contains four times less iron, twice less B vitamins, and lower levels of protein and zinc than wild meat³⁸.

Due to its high cost and the significant logistical challenges of producing, transporting, and storing in remote areas, beef is rarely consumed across most of Amazonia³⁷. Paradoxically, cattle ranching remains the leading driver of deforestation in the region, contributing to the loss of approximately 0.63M km² of forest since 1978, primarily to supply domestic meat markets³⁹. Replacing the estimated edible wild meat production of 0.37 M t with beef, based on current cattle ranching yields in Amazonian traditional pastures (0.2–0.8 kg ha⁻¹ day⁻¹)⁴⁰, would require converting 7,602–63,798 km² of forest into pasture. This conversion would emit 0.3×10⁹ – 3.5×10⁹ tonnes of CO₂, equivalent to up to 10% of global annual emissions. Despite this significant environmental cost, domestic meat production still fails to ensure equitable access for rural peoples in remote areas.

Lines 177-184: can carnivores, primates, tapir, white-lipped peccary be hunted sustainably? Where are offtakes sustainable for these species? Provide examples of hunting / wildlife management systems that achieve sustainability. What should people hunt? How can management systems ensure sustainability?

177 Of the total number of taxa hunted and consumed only 20 surpassed the threshold of more than 500,000

178 animals hunted annually. These included 14 mammals (1 cingulate, 1 carnivore, 3 rodents, 4 primates,

179 and 5 ungulates), 4 birds and 2 reptiles (Fig. 2). Patterns of exploitation for each of these taxa are

180 shown in Fig. 3. Collectively, these taxa contribute 70.0% of all individual animals hunted and 85.2%

*181 of the overall biomass extracted, with the white-lipped peccary (*Tayassu pecari*) and the tapir (*Tapirus**

182 terrestris) together accounting for almost half of the total biomass extracted. Accordingly, only a

183 relatively small subset of wildlife contributes to the overall numbers of animals harvested and biomass

184 extracted in Amazonia.

The sustainability of wildlife use, as framed within the Sustainable Development Goals (SDGs), does not focus solely on the sustainability of target species populations. Instead, our analysis considers sustainability in the broader context of SDGs, addressing food security, livelihoods, and governance dimensions. Furthermore, evaluations of hunting sustainability are “context-specific.” For this our article underscores:

New text (Lines 276-326):

The fate of wild meat food systems in Amazonia

Our study highlights the essential role of traditional hunting and wild meat access in advancing several Sustainable Development Goals (SDGs) of The United Nations Convention on Biological Diversity, supporting nutritional security and health, helping reduce malnutrition, strengthening traditional food systems, and promoting sustainable wildlife use and ecosystem conservation across Amazonia (Extended Data Fig. 14). Although the sustainability of hunting in tropical forests and the risks of zoonotic diseases

potentially linked to hunting have been intensively debated, we focus on understanding traditional hunting practices and wild meat access within the broader context of achieving the SDGs in Amazonia. We demonstrated that the health of the Amazon Forest is vital to sustaining traditional wild meat food systems and the nutrition of Amazonian Peoples. From this evidence, we contend that illegitimate proposals to ban, restrict, or replace wild meat without acknowledging its cultural and nutritional significance reflect a colonial mindset that undermines the autonomy and traditional food systems of Amazonian Peoples.

We provide new insights into the complex interplay of environmental, cultural and threatening factors shaping wildlife harvest patterns throughout Amazonia. The finer details of these interactions merit further locally focused research and avoid determinisms. While the Sustainable Development Goals and IUCN Red List of Threatened Species frameworks emphasise global policies and actions, we stress that wildlife management initiatives must be tailored to local ecological, cultural, socioeconomic, and vulnerability nuances, ideally focused on local and regional key hunted taxa and led by Amazonian Peoples. Wildlife management initiatives shaped by Amazonian Peoples' demands and cultural practices are more legitimate and likely to remain viable long-term. Indigenous Lands and Extractive Reserves usually maintain healthy populations of key hunted species⁴¹, even those once commercially over-harvested⁴². Indigenous and traditional knowledges are critical for assessing the conservation status of animal populations⁴³ and understanding key ecological parameters⁴⁴, including their varying density⁴⁵, reproductive rates⁴⁶, and population dynamics²⁶. These factors, along with deforestation³⁴ and commercial hunting pressures⁴⁷, and climate change impacts³⁵, strongly influence sustainable harvest potentials. Our predictive heatmaps offer valuable spatial insights into harvest productivity, human-wildlife interactions, and human nutrition in Amazonia, supporting more effective policies integrating conservation and public health.

Scientific evidence demonstrates that traditional hunting in Amazonia has historically relied on sparse human populations and the limited spread of hunters across vast, conserved forests⁴⁷. This mechanism guarantees the preservation of large spatial refuges for terrestrial species, in contrast to aquatic species⁴⁷, which often require spatial zoning and local agreements to maintain healthy populations⁴⁸. While well-preserved forests combined with strong local governance can support sustained harvests through source-sink hunting dynamics, our findings show that forest degradation is increasingly threatening wild meat production, undermining unique traditional food systems. In these contexts, building agreements for hunting management may be critical to ensure Amazonian Peoples' well-being and the sustainability of the wildlife they rely on.

Amazonian systems of knowledges and practices are grounded in ontologies attributing agency, personhood, and humanity to various beings, including animals⁴⁹, which must be taken seriously in ecological assessments of traditional hunting and management systems⁵⁰. For Amazonian Peoples, relationships with wildlife are not based on resource extraction and nutrient provisioning but rather on social reciprocity governed by norms and ethical obligations⁵¹. Dietary restrictions and spatial avoidance function as wildlife management and conservation systems akin to species-specific protection, spatial zoning rules, and hunting bans⁵². These cultural practices are effective tools in regulating wildlife harvesting and the most legitimate management and conservation strategies in Amazonia⁵³. Wild meat is a critical food source and a social cornerstone that motivates Amazonian Peoples to safeguard their territories. The health of Amazonian ecosystems and wildlife is inextricably linked to the well-being of Amazonian Peoples, underscoring the importance of recognizing

their land rights and supporting policies that enhance their autonomy and governance over their territories and biodiversity.

Below, we provide further explanations to the reviewer that support this broader sustainability perspective and elaborate on the complexities of hunting sustainability beyond target species populations. While a comprehensive discussion of this topic would require significantly more space than is available for our manuscript, we aim to highlight key insights that address the reviewer's concerns.

Like any extractive activity, traditional hunting can impact the demographics of hunted species populations. However, a broader perspective is necessary to understand its sustainability. If traditional hunting were inherently unsustainable, at a small or a large scale, it would be difficult to explain how animal populations have persisted in Amazonia despite millennia of Indigenous hunting, especially considering the numbers presented in this large-scale study and many others at a local scale (Antunes et al. 2016; Peres 2000; Robinson and Redford 1991). Over the last few decades, research has examined this question, providing insights into the conditions under which hunting can be sustainable. Studies from the 1990s and 2000s consistently found that subsistence hunting locally reduces the density, abundance, and biomass of medium and large-bodied species based on animal observation in transect surveys (Robinson & Redford 1991; Robinson & Bennett, 2000; Peres, 2000), with greater impacts on species with low reproductive rates, such as tapirs and large primates (Bodmer et al., 1997). The sustainability index developed by Robinson and Redford (1991), which compares extraction rates with reproductive rates, often predicts the local extinction of low-reproductive species. This led to the concept of the "empty forest" hypothesis, which suggests that defaunation from hunting can disrupt ecological processes, jeopardizing forest viability (Redford, 1992), which indeed can occur on local scales (Terborgh et al. 2008)

More recent studies have critically reassessed these conclusions. The primary method for obtaining population parameters in the mentioned articles - line transect surveys - tends to underestimate population sizes in areas with intensive hunting, as animals become more elusive (Fragoso et al., 2016). See these articles for similar discussion: Zwicker (2023), Sánchez-Mercado et al. (2022), Rovero et al. (2023), Carvalho et al. (2021), and Torres et al. (2021).

Furthermore, some studies have demonstrated that offtake rates for species predicted to be locally extinct remain stable over time, suggesting the existence of replenishment mechanisms (Ohl-Schacherer et al., 2007; Constantino 2016). The Robinson and Redford (1991) model does not account for migration from nearby refuge areas, which play a crucial role in sustaining hunted populations (Joshi & Gadgil, 1991). The source-sink hunting dynamics, where hunted areas are replenished by individuals from nearby non-hunted or lightly hunted refuges, are one of the key mechanisms for maintaining hunting sustainability in Amazonia (Antunes et al., 2016, Constantino et al. 2008). When not combined with deforestation, road expansion, or commercial hunting, on a landscape scale, subsistence hunting can be sustainable even for vulnerable species such as large primates and tapirs (Novaro et al., 2000; Levi et al., 2009; Constantino, 2016, de Paula et al. 2022).

The Amazon's vast expanse further limits the potential negative impacts of subsistence hunting. Most hunting trips occur within a 6 km radius of a hunter's settlement, meaning that hunted areas remain relatively small compared to the landscape (Constantino et al 2018, REFs). As a result, extensive refuges without hunting pressure help sustain wildlife

populations (Novaro et al., 2000; Levi et al., 2009). Approximately 70% of the lowest-fecundity and most highly prized hunted species, such as ateline primates (*Ateles* and *Lagothrix*), currently exist at carrying capacity at both landscape (Levi et al., 2009) and basin-wide scales (Peres et al., 2016). Wildlife populations can be maintained in areas without deforestation through mechanisms analogous to the source-sink dynamic (Pulliam, 1988). However, sustainability potential is diminished in regions where deforestation, habitat fragmentation, road networks, commercial exploitation, and emerging zoonotic diseases overlap (Peres, 2001; Constantino, 2016; Richard-Hansen et al. 2019).

In addition, Indigenous perspectives on hunting diverge from conventional ecological models. Many Indigenous groups conceptualize their relationship with wildlife as social reciprocity rather than simple resource extraction. Ethical considerations and spiritual beliefs shape hunting practices, reinforcing self-imposed restrictions and minimizing overexploitation. Cultural food restrictions, and spatial avoidance, widespread in Indigenous and traditional communities, serve as informal institutions regulating wildlife use (Colding & Folke, 2001). These community-based management strategies, often more enduring and effective than formal state regulations, resemble Western conservation strategies such as zoning, quotas, seasonal bans, and gear restrictions. However, cultural shifts, technological advances, market integration, and external threats to ecosystems can erode these traditional practices.

The establishment of protected areas has been a key conservation strategy. Indigenous territories, extractive reserves, and parks offer regulatory habitat protection and frameworks for sustainable resource use, fostering community participation in conservation and management efforts. Recognizing and integrating traditional governance systems into formal policies can enhance conservation outcomes, as these local rules function similarly to protected area zoning by delineating intensive and extensive use zones and establishing hunting restrictions (Constantino et al., 2018). The role of non-hunted refuges in maintaining wildlife populations has led to their proposed inclusion in conservation planning within protected areas (Constantino et al., 2018).

In summary, our approach in the manuscript was analysing wild meat sustainability within the broader socio-economic, cultural, and governance dimensions outlined in the SDGs. It is not simply a question of ecological viability of target species but of ensuring just, inclusive, and resilient systems that balance conservation goals with local needs. Sustainable hunting is not a binary metric but a context-dependent outcome influenced by governance structures, traditional knowledge, and conservation policies.

References:

- Antunes, A. P. et al. Empty forest or empty rivers? A century of commercial hunting in Amazonia. *Sci. Adv.* 2, e1600936 (2016).
- Bizri, H.R.E., Araújo, L.W.D.S., Araújo, W.D.S., Maranhão, L. and Valsecchi, J., 2016. Turning the game around for conservation: using traditional hunting knowledge to improve the capture efficiency of Amazon lowland pacas. *Wildlife Biology*, 22(1), pp.1-6.
- Bodmer R. E., J. F. Eisenberg, K. H. Redford, Hunting and the likelihood of extinction of Amazonian mammals. *Conserv. Biol.* 11, 460–466 (1997).
- Carvalho, W. D., et al. (2021). Effects of human disturbance on the behavior and ecology of Amazonian mammals. *Biotropica*, 53(4), 1123-1135.

- Colding, J. & Folke, C. Social taboos: “invisible” systems of local resource management and biological conservation. *Ecol. Applic.* 11, 584-600 (2001).
- Constantino PAL (2016). Deforestation and hunting effects on wildlife across Amazonian indigenous lands. *Ecology and Society*, 21(2).
- Constantino PAL, Benchimol M, Antunes AP (2018). Designing Indigenous Lands in Amazonia: Securing indigenous rights and wildlife conservation through hunting management. *Land Use Policy*, 77, 652-660.
- Fragoso J. M. V. et al. Line transect surveys underdetect terrestrial mammals: Implications for the sustainability of subsistence hunting. *PLOS One* 11, e0152659 (2016).
- Joshi N. V., M. Gadgil, On the role of refugia in promoting prudent use of biological resources. *Theor. Popul. Biol.* 40, 211–229 (1991).
- Levi T., G. H. Shepard Jr., J. Ohl-Schacherer, C. A. Peres, D. W. Yu, Modelling the long-term sustainability of indigenous hunting in Manu National Park, Peru: Landscape-scale management implications for Amazonia. *J. Appl. Ecol.* 46, 804–814 (2009).
- Novaro A. J. , K. H. Redford, R. E. Bodmer, Effect of hunting in source-sink systems in the Neotropics. *Conserv. Biol.* 14, 713–721 (2000).
- de Paula MJ, Carvalho EA, Lopes CKM, et al. Hunting sustainability within two eastern Amazon Extractive Reserves. *Environmental Conservation*. 2022;49(2):90-98. doi:10.1017/S0376892922000145
- Peres C. A., Effects of subsistence hunting on vertebrate community structure in Amazonian forests. *Conserv. Biol.* 14, 240–253 (2000).
- Peres, C.A., 2001. Synergistic effects of subsistence hunting and habitat fragmentation on Amazonian forest vertebrates. *Conservation biology*, 15(6), pp.1490-1505.
- Peres C. A., T. Emilio, J. Schietti, S. J. M. Desmoulière, T. Levi, Dispersal limitation induces long-term biomass collapse in overhunted Amazonian forests. *Proc. Natl. Acad. Sci. U.S.A.* 113, 892–897 (2016).
- Redford K. H. The empty forest. *BioScience* 42, 412–422 (1992).
- Richard-Hansen, C., Davy, D., Longin, G., Gaillard, L., Renoux, F., Grenand, P., & Rinaldo, F. (2019). Hunting in French Guiana Across Time, Space and Livelihoods. *Frontiers in Ecology and Evolution*, 7, 289. <https://doi.org/doi: 10.3389/fevo.2019.00289>
- Robinson J. G., K. H. Redford, Sustainable harvest of neotropical forest animals, in *Neotropical Wildlife Use and Conservation*, J. G. Robinson, K. H. Redford, Eds. (The University of Chicago Press, Chicago, 1991), chap. 27, pp. 415–429.
- Robinson J. G., E. L. Bennett, *Hunting for Sustainability in Tropical Forests* (Columbia Univ. Press, New York, 2000).
- Rovero, F., et al. (2023). Camera trapping reveals the impacts of hunting on Amazonian mammal communities. *Ecological Applications*, 33(1), e2745.

Sánchez-Mercado, A., et al. (2022). Hunting-induced changes in the behavior and ecology of Amazonian primates. *American Journal of Primatology*, 84(1), e23345

Terborgh, et al. Tree recruitment in an empty forest. *Ecology* 89, 1757–1768 (2008).

Torres, P. C., et al. (2021) Hunting pressure modulates the spatial behavior of Amazonian mammals. *Journal of Applied Ecology*, 58(6), 1234-1244.

Zwicker S. 2023. Elusive felids of lowland Amazonia: Assessing the effects of human disturbance across an unprotected landscape. Dissertation, University of Washington.

Lines 200-204: again, can large primates and manatee be hunted sustainably?

*200 caimans, howler monkeys, capybara, manatee, and waterfowl (ducks, cormorants, and egrets)
201 compared to regions dominated by upland terra firme forests. In the latter habitats, which
make up 86%*

*202 of Amazonia²⁶, tinamous, trumpeters, wood quails, guans, curassows and armadillos are
relatively more*

*203 hunted. Large primates, like woolly (Lagothrix spp.), spider (Ateles spp.), howler (Alouatta
spp.) and*

204 capuchin monkeys (Sapajus spp.), are hunted more in Western and Central Amazonia.

The same principle discussed in the previous question applies to species more vulnerable to hunting, such as carnivores and primates. Most carnivores, tapirs, peccaries, and monkeys are terrestrial and benefit from extensive refuge areas. This is largely due to the limited access of hunters within the vast and often non-accessed interior of the Amazon rainforest, as previously explained. For example, a study in the Peruvian Amazon (Levi et al. 2009) found that indigenous hunting is unlikely to deplete large-bodied spider monkeys and, by extension, other fauna despite rapid human population growth, mostly because of source-sink dynamics.

Conversely, manatees face a significantly higher risk of overhunting due to their unique ecological characteristics. In addition to their slow reproductive rates, they are aquatic, making them far more vulnerable to hunting (Antunes et al., 2016). Human settlements in the Amazon are predominantly located along rivers, where boats and canoes provide easy access to these animals. The relatively smaller size of flooded forests compared to upland forests—approximately 14% of the total area—further restricts potential refuge zones for manatees. This combination of factors has historically led to high hunting pressure on this species.

Manatees also have a distinct historical trajectory of human use in the Amazon. For centuries, they formed a staple part of the diet of Indigenous peoples and, later, of European colonizers in the Amazon River mainstem and its major tributaries. However, with the advent of commercial hunting, manatees were relentlessly exploited for their fat, meat, and hide. The introduction of fishing nets and high commercial demand led to a dramatic decline in their populations throughout the second half of the 20th century (Antunes et al., 2016).

Despite this decline, various conservation measures have contributed to population recovery in several areas. These include the prohibition of commercial hunting, including manatees on endangered species lists, establishing protected areas, and implementing local and national regulations governing subsistence hunting (Souza 2015). Similar trends have been

observed in other aquatic species that suffered intense commercial exploitation during the 20th century (Antunes et al. 2016), such as the giant otter (Lima et al. 2014) and black caiman (Silveira & Thorbjarnarson 1999). Due to conservation efforts, their populations have rebounded, and sustainable harvesting is now more feasible compared to historical overexploitation.

The manatee holds profound importance for the Paumari Indigenous people, inhabitants of the highly productive transition zones between flooded and upland forests along the Purus River and its tributaries. Beyond being a key food source, the species is deeply embedded in their cosmological and cultural traditions. The Paumari have hunted manatees for centuries, if not millennia, following specific seasonal and spatial restrictions that ensure sustainability within their territories (A. P. Antunes personal observation).

However, it is clear that hunting highly vulnerable species, such as large primates and manatees, is generally riskier due to their slow reproductive rates, low population densities, and susceptibility to habitat loss and overexploitation. While some Indigenous communities have historically incorporated these species into their economic strategies, ensuring sustainability can require more stringent management approaches. These may include hunting quotas, seasonal restrictions, habitat protection, and regulated harvest systems.

As stated earlier, our study does not focus on assessing the sustainability of specific species populations but rather on the broader socio-economic, cultural, and governance dimensions of wild meat use, as outlined in the SDGs. A comprehensive analysis of species-specific sustainability requires a different methodological approach and falls beyond the scope of our study.

References:

Antunes, A. P. et al. Empty forest or empty rivers? A century of commercial hunting in Amazonia. *Sci. Adv.* 2, e1600936 (2016).

Levi, T., Shepard Jr, G.H., Ohl-Schacherer, J., Peres, C.A. and Yu, D.W., 2009. Modelling the long-term sustainability of indigenous hunting in Manu National Park, Peru: landscape-scale management implications for Amazonia. *Journal of Applied Ecology*, 46(4), pp.804-814. <https://doi.org/10.1111/j.1365-2664.2009.01661.x>

Lima D. S., M. Marmontel, E. Bernard, Reoccupation of historical areas by the endangered giant river otter *Pteronura brasiliensis* (Carnivora: Mustelidae) in Central Amazonia, Brazil. *Mammalia* 78, 177–184 (2014).

Silveira R., J. B. Thorbjarnarson, Conservation implications of commercial hunting of black and spectacled caiman in the Mamirauá Sustainable Development Reserve, Brazil. *Biol. Conserv.* 88, 103–109 (1999).

Souza D. S., Peixe-boi da Amazônia (*Trichechus inunguis natterer* 1883): mortalidade e uso do habitat na reserva de desenvolvimento sustentável Piagaçu-Purus, Amazônia central, Brasil, thesis, Instituto Nacional de Pesquisas da Amazônia, Manaus, Amazonas (2015).

Lines 216-220: these examples seem to have very little impact overall. Do they contribute to wildlife conservation?

216 For instance, some hyperdominant hunted species, like

217 ungulates, are deliberately avoided by certain Aruak and Tupi Indigenous Peoples along the Xingu and

218 Madeira rivers (L. Suruí and Y. Waura, pers. obs.). Conversely, other Aruak Peoples in Northwestern

219 Amazonia highly value slender-legged tree frogs (Osteocephalus spp., D. Baniwa, pers obs), a
220 species often overlooked by many other Amazonian peoples.

We provided these local examples to show how crucial cultural factors can be in shaping the composition of hunted species throughout the Amazon. The reasons why the Indigenous peoples of the Amazon adopt restrictions and prohibitions on eating animals are a matter of debate. In many Indigenous societies, it is believed that wildlife use requires a balance of spiritual forces between the hunter and the animal spirit-owners. This prevents the exaggeration of hunters and often prohibits species and places. Therefore, although these prohibitions, precautions, and bans may not have a conservationist origin in practice, they can considerably reduce the negative demographic effects on hunted species, akin to conservation strategies like zoning, closed seasons, species protected by law, prohibition of harmful techniques, integral preservation of certain areas, among others. In this new version, we included more information about the role of cultural factors on wild meat use in Amazonia, based on some of our results, as follows:

New text (Lines 200-206):

Cultural identity strongly influences HP and TSOP across taxa. For example, most Aruak Peoples value hunting ungulates, yet those in the Xingu River deliberately avoid them. In contrast, some Indigenous peoples in the northwestern Amazon highly value the often overlooked slender-legged treefrogs (Osteocephalus spp.). These patterns reflect a complex interplay of cultural preferences, environmental conditions, and ecological factors shaping hunting dynamics. Understanding these systems requires moving beyond environmental or cultural determinism toward an interdisciplinary perspective.

The effectiveness of these cultural strategies on the demography of hunted species still needs to be studied in depth in the Amazon. Consider, for example, the Paiteer Suruí indigenous territory of the Tupi linguistic family in Rondonia state, Brazil, which has 248,000 hectares. In this entire area, the red brocket deer (*Mazama americana*) are not hunted because they are considered to have an evil spirit. In contrast, tapirs (*Tapirus terrestris*) are only eaten by the elders because they are the only ones with a consolidated and powerful spirit, ready to face the strong spirit of the tapir. There are similar examples throughout the entire Amazon. There is no doubt that these strategies combined across different Indigenous societies have tremendous practical effects on the demography and conservation of hunted species in large areas.

However, we agree with the reviewer that these positive effects are context-dependent and function at the local scale. These cultural systems alone can be insufficient for proper wildlife conservation in regions where hunting is not well-regulated. Effective wildlife conservation requires a multifaceted approach, integrating strong legal frameworks, habitat protection, community engagement and autonomy, cultural respect, feasible management strategies, and even alternative livelihoods in case people demand to address both immediate and long-term biodiversity threats.

To address these, we finished our article with the following statements:

New text (Lines 314-326):

Amazonian systems of knowledges and practices are grounded in ontologies attributing agency, personhood, and humanity to various beings, including animals⁴⁹, which must be taken seriously in ecological assessments of traditional hunting and management systems⁵⁰. For Amazonian Peoples, relationships with wildlife are not based on resource extraction and nutrient provisioning but rather on social reciprocity governed by norms and ethical obligations⁵¹. Dietary restrictions and spatial avoidance function as wildlife management and conservation systems akin to species-specific protection, spatial zoning rules, and hunting bans⁵². These cultural practices are effective tools in regulating wildlife harvesting and the most legitimate management and conservation strategies in Amazonia⁵³. Wild meat is a critical food source and a social cornerstone that motivates Amazonian Peoples to safeguard their territories. The health of Amazonian ecosystems and wildlife is inextricably linked to the well-being of Amazonian Peoples, underscoring the importance of recognizing their land rights and supporting policies that enhance their autonomy and governance over their territories and biodiversity.

In addition, we also considered

New text (Lines 207-213):

While our models adequately capture TSOP variation for key hunted taxa, they are less reliable for underrepresented species (Supplementary Data 2). In addition, hunting practices have evolved over the past 60 years, influenced by cultural and technological shifts. Increased rifle use, for example, has increased hunting efficiency, particularly for large species²⁴, while the growing use of flashlights has intensified the hunting pressure on nocturnal animals such as the paca²⁵, now the most hunted species in Amazonia. Our models also do not account for recently documented natural population cycles in white-lipped peccaries²⁶, which may influence hunting patterns over time.

Lines 231-235: This section is about nutritional value of wild meat, so move the information on economic value/trade/income to another section. Is game meat sold for the same price as beef in local markets?

230 The economic value of this

231 meat would be equivalent to US\$ 2.1±0.9 (1.3–3.1) billion, based on the market value of beef in 2024.

*232 The high economic value of wild meat suggests that sustainable harvesting can provide local
233 communities with affordable high-quality nutrition by reducing substantial food expenditure,
while*

*234 local trade can generate income. Combined, these factors contribute to achieving targets 1.1
and 1.4*

235 related to poverty alleviation (No Poverty, SDG 1).

Thanks for the suggestion. We have now moved the information to another section:

New text (Lines 214-224):

Overall wild meat production

We estimated that the annual edible wild meat production in Amazonia amounts to 0.34±0.09 (0.24–0.44) M t, representing 58.5% of the total undressed biomass (see Methods). As much as 86.4% of the wild meat produced in Amazonia—equivalent to 0.30±0.08 (0.22–0.38) M t—is derived from the key 20 taxa.

The annual monetary value of this wild meat production is approximately US\$ 2.2±0.6 (1.6–2.9) billion, based on 2024 beef market prices. Accurately assessing the economic value of wild meat remains challenging due to the often informal or illegal nature of the trade in most Amazonia. However, this hidden economic value of wild meat production suggests that traditional hunting is a significant ecosystem service for Amazonian Peoples, providing affordable, high-quality nutrition and reducing food expenditures.

Lines 292-293: do you have information that chicken provides less micro-nutrients than wild animals?

291 The diminished access to

292 wildlife near urban areas prompts hunger or dietary shifts to domestic meats, particularly chicken,

293 potentially lowering vital micronutrient intake compared to regular wild meat consumers³⁶

Yes, we do. Comparing our calculated averages of macro- and micro-nutrients in wild meat and a reference for chicken from the Brazilian Table of Food Composition, these are the results:

Protein (g) → 21.9 (higher than chicken: 16.4 g)

Iron (mg) → 2.69 (higher than chicken: 0.6 mg)

Zinc (mg) → 1.67 (higher than chicken: 1.1 mg)

Vitamin B1 (mg) → 0.16 (higher than chicken: 0.08 mg)

Vitamin B2 (mg) → 0.27 (higher than chicken: 0.03 mg)

References

Núcleo de Estudos e Pesquisas em Alimentação – NEPA. (2011). *Tabela Brasileira de Composição de Alimentos – TACO* (4^a ed. revisada e ampliada). Universidade Estadual de Campinas – UNICAMP. Available at: <http://www.unicamp.br/nepa>

We included the reference in the text:

New text (Lines 262-266):

High meat demand in densely populated peri-urban areas, coupled with declining wildlife populations³⁶, may shift rural diets toward cheaper domestic meats like chicken³⁷, which generally contains four times less iron, twice less B vitamins, and lower levels of protein and zinc than wild meat³⁸.

Lines 294-295: beef is exported, is it not?

294 Paradoxically, although beef is rarely consumed in most of Amazonia due to its high cost and logistical

295 constraints, it significantly contributes to forest loss.

Not really, around 1/5 of the beef produced is exported, the remaining consumed domestically (Zu Ermgassen et al., 2020).

References

Zu Ermgassen, E. K. H. J., Godar, J., Lathuillière, M. J., Löfgren, P., Gardner, T., Vasconcelos, A., & Meyfroidt, P. (2020). The origin, supply chain, and deforestation risk of Brazil's beef exports. *Proceedings of the National Academy of Sciences*, 117(50), 31770-31779. <https://doi.org/10.1073/pnas.2003270117>

Lines 298-299: is 18,890-42,971 km² a lot? Less than 1% of the biome?

297 our results show that substituting the

298 estimated edible wild meat figures (0.33 million t) with beef would necessitate converting 18,890–

299 42,971 km² of forest into traditional pastures.

The Amazon biome spans approximately 6.7-8.0 million km² depending on the criteria adopted (ecological, geopolitical, and hydrological). Thus, 8,121–69,008 km² represents 0.01% to 0.1% of the total area. While this is a small percentage, This figure corresponds to the size of Switzerland, the Netherlands or Ireland. This also represents an emission of $0.3 \times 10^9 - 3.5 \times 10^9$ tonnes of CO₂, corresponding to up to 10% of global annual emissions. The ecological, climatic, and social impacts of converting an equivalent area of forest to pasture should not be underestimated; they remain significant due to the values of even small areas in the rainforest. Most of the deforestation and land grabbing in Amazonia result in social conflicts that are frequently quite violent.

New text (Lines 267-275):

Due to its high cost and the significant logistical challenges of producing, transporting, and storing in remote areas, beef is rarely consumed across most of Amazonia³⁷. Paradoxically, cattle ranching remains the leading driver of deforestation in the region, contributing to the loss of approximately 0.63M km² of forest since 1978, primarily to supply domestic meat markets³⁹. Replacing the estimated edible wild meat production of 0.37 M t with beef, based on current cattle ranching yields in Amazonian traditional pastures (0.2–0.8 kg ha⁻¹ day⁻¹)⁴⁰, would require converting 7,602–63,798 km² of forest into pasture. This conversion would emit $0.3 \times 10^9 - 3.5 \times 10^9$ tonnes of CO₂, equivalent to up to 10% of global annual emissions. Despite this significant environmental cost, domestic meat production still fails to ensure equitable access for rural peoples in remote areas.

Line 315: you mention food health and pathogen surveillance here, but do not discuss these topics anywhere in the manuscript.

314 harvesting. Our taxon-specific extraction maps can guide policies on effective wildlife management,

315 food health and pathogen surveillance.

We agree with the reviewer. Given that pathogen surveillance is beyond the scope of our study and would require an individual analysis, we removed the content related to this topic. We now only mention it briefly in the following lines:

New text (Lines 389-304):

We provide new insights into the complex interplay of environmental, cultural and threatening factors shaping wildlife harvest patterns throughout Amazonia. The finer details of these interactions merit further locally focused research and avoid determinisms. While the Sustainable Development Goals and IUCN Red List of Threatened Species frameworks emphasise global policies and actions, we stress that wildlife management initiatives must be tailored to local ecological, cultural, socioeconomic, and vulnerability nuances, ideally focused on local and regional key hunted taxa and led by Amazonian Peoples. Wildlife management initiatives shaped by Amazonian Peoples' demands and cultural practices are more legitimate and likely to remain viable long-term. Indigenous Lands and Extractive Reserves usually maintain healthy populations of key hunted species⁴¹, even those once commercially over-harvested⁴². Indigenous and traditional knowledges are critical for assessing the conservation status of animal populations⁴³ and understanding key ecological parameters⁴⁴, including their varying density⁴⁵, reproductive rates⁴⁶, and population dynamics²⁶. These factors, along with deforestation³⁴ and commercial hunting pressures⁴⁷, and climate change impacts³⁵, strongly influence sustainable harvest potentials. Our predictive heatmaps offer valuable spatial insights into harvest productivity, human-wildlife interactions, and human nutrition in Amazonia, supporting more effective policies integrating conservation and public health.

Lines 314-315: do you have examples to cite here of successful policies and effective wildlife management developed from taxon-specific extraction maps?

314 harvesting. Our taxon-specific extraction maps can guide policies on effective wildlife management,

315 food health and pathogen surveillance.

New text (Lines 302-304):

Our predictive heatmaps offer valuable spatial insights into harvest productivity, human-wildlife interactions, and human nutrition in Amazonia, supporting more effective policies integrating conservation and public health.

Several examples in the literature of successful policies and effective wildlife management systems have been developed using taxon-specific extraction maps. These maps provide crucial data on species distribution, spatial spread of hunting pressure, and ecosystem health, which can inform sustainable wildlife management strategies. Here are a few examples:

Amazonia

- In the Peruvian Amazon, collared peccary pelts are exported under CITES Appendix II regulations, and Peruvian authorities adjust regional export quotas based on scientific wildlife monitoring (Bodmer et al., 2023). This scientific monitoring has been successfully supported by a communal forest management best practices certification program, which includes hunting records and participatory extraction maps, and wildlife censuses. As a result, long-term collared peccary populations have remained relatively stable (Fang et al., 2008)
- A collaborative project in the Paumari Indigenous Lands created maps to spatially visualize the number of animals hunted per species, combining wildlife hunting monitoring and participatory mapping. These maps played a decisive role in supporting the Paumari people's formal request to the Brazilian Government to re-delimitate the Paumari Indigenous lands. A working group has been established to conduct anthropological and environmental studies in their territory.
- Levi et al. (2011) used empirical hunting data to construct spatial models of spider monkey depletion across a landscape in the Peruvian Amazon. However, it is unclear whether their models have been applied to management in practice. Some authors (e.g., Novaro et al. 2000) have advocated for the application of source-sink models in subsistence hunting management, and there are examples of local hunting agreements incorporating source-sink considerations, such as the definition of no-take zones in Cazumbá-Iracema Extractive Reserve (Oliveira and Calouro 2019) and the Amanã Lake region of Brazil's Amazonas state (Loureiro et al. 2024). However, these agreements are not strictly based on taxon-specific extraction maps.
- Indigenous peoples in the Amazon state of Acre have used hunting maps, either taxon-specific or not, to support management decision in their demarcated Indigenous Lands. Constantino et al. (2012) shows how hunting maps were created during the ethnomapping process of Kaxinawa da Praia do Carapanã Indigenous Land and used by the Huni Kuin to support their environmental and territorial management plans. The Huni Kuin of Kaxinawá/Ashaninka do Rio Breu Indigenous Land mapped every hunted animal as part of their monitoring scheme and used this information to support their hunting agreements (Constantino 2020).
- The territorial zoning model, which delimits areas for direct, indirect and non-use, is mostly based on the source-sink logic and has been used in managing protected areas in Brazil that are co-managed by state agencies, traditional populations and other stakeholders. In the Piagaçu-Purus RDS, for example, the model was cited by hunters as an effective conservation strategy for protecting the species on which they depend for their food security. The tapir (*Tapirus terrestris*) is the most emblematic case, for which the hunters have specific hunting rules, which include protecting the areas they use for feeding, specifically the saltlicks, preventing outside users from accessing these areas and regularly switching the places they hunt this species (Vieira et al. 2019).
- In French Guiana, results of hunting surveys were used to support arguments for governmental policy makers for hunting ban seasons for green iguana (*Iguana iguana*), and to change regulations on tapir hunting, confirming the local hunters' feelings and demand.
- Community-based management of *Arapaima gigas* (pirarucu), the world's largest freshwater fish, has proven to be an effective strategy for the recovery of wild populations in the Amazon. Historically devastated by overfishing, pirarucu populations are now showing signs of recovery due to community-led management efforts. These efforts have significantly contributed to the regeneration of pirarucu stocks, even outside formal protected areas. Protected lakes, managed by local communities, serve as productive "high-interest savings accounts," ensuring a rare income opportunity for local

communities. These management strategies not only boost biodiversity conservation but also enhance local livelihoods. The success of pirarucu management highlights the potential of community-led initiatives to promote both ecological sustainability and socioeconomic development, underscoring the importance of including local stakeholders in conservation planning for Amazonian floodplains.

- Brazil's Indigenous Territories: In Brazil, Indigenous communities have used taxon-specific maps to regulate hunting practices within their territories, particularly for species like the peccary and tapir. These maps track hunting intensity and species distribution, allowing communities to set sustainable hunting limits, including temporal closures or restricted areas, to prevent overexploitation. These systems are aligned with traditional ecological knowledge and complement modern conservation practices, promoting both biodiversity conservation and the sustenance of Indigenous communities.

Tropical forest worldwide

- In Madagascar, the use of taxon-specific extraction maps for lemurs has helped design sustainable hunting guidelines for certain lemur species, particularly in rural communities. The maps have highlighted areas of high hunting pressure and have been used to adjust quotas and hunting seasons. As a result, conservation policies that balance local livelihoods and lemur populations have been implemented, contributing to the protection of several lemur species from overexploitation.
- In South Africa, community-based wildlife management programs have successfully used taxon-specific extraction maps to monitor game populations and hunting activities. For example, in the communal areas of the Limpopo and Eastern Cape regions, maps detailing the distribution of key species like wildebeest, antelope, and buffalo are used to create hunting quotas. This information has led to sustainable wildlife management, enhancing both conservation and local income from regulated hunting and eco-tourism.
- Tanzania has used taxon-specific maps for wildlife management in its protected areas, particularly in the Serengeti National Park. These maps allow authorities to monitor the effects of poaching and legal hunting, ensuring that certain species, such as elephants and lions, are protected while others, like wildebeest, can be harvested sustainably. This approach has helped mitigate human-wildlife conflict while preserving the ecological balance.

References

Bodmer Richard E., Pablo Puertas, Tula Fang, Miguel Antúnez, Sandro Soplín, Jhonathan Caro, Pedro Pérez, Hani R. El Bizri, Marco Arenas, José Carlos Nieto, Maire Kirkland, and Pedro Mayor (2023). Management of Subsistence Hunting of Mammals in Amazonia: A Case Study in Loreto, Peru. In: Spironello, W.R., Barnett, A.A., Lynch, J.W., Bobrowiec, P.E.D., Boyle, S.A. (eds) *Amazonian Mammals*. Springer, Cham. https://doi.org/10.1007/978-3-031-43071-8_10

Constantino PAL, Tavares RA, Kaxinawa JA, Kaxinawa FM, Kaxinawa E, Kaxinawa AS, 2012. Monitoramento e mapeamento participativo da caça na Terra Indígena Kaxinawá da Praia do Carapanã (Acre). In, *Conservação da biodiversidade com SIG* Chapter: Monitoramento e Mapeamento Participativo na Terra Indígena Kaxinawa da Praia do Carapanã (Acre) Publisher: Oficina de textos Editors: A. Paese, A. Uezu, M.L. Lorini, A. Cunha

Constantino PAL, Benchimol M, Antunes AP (2018). Designing Indigenous Lands in Amazonia: Securing indigenous rights and wildlife conservation through hunting management. *Land Use Policy*, 77, 652-660. <https://doi.org/10.1016/j>.

Constantino PAL, 2020. Challenges of Forest Citizen Involvement in Biodiversity Monitoring in Protected Areas of Brazilian Amazonia. In, *Handbook of Citizen Science in Ecology and Conservation*. Lepczyk CA, Boyle OD, Vargo TLV (Eds) Publisher: University of California Press.

Campos-Silva, J. V., & Peres, C. A. (2016). Community-based management induces rapid recovery of a high-value tropical freshwater fishery. *Scientific reports*, 6(1), 34745.

Campos-Silva, J. V., Hawes, J. E., & Peres, C. A. (2019). Population recovery, seasonal site fidelity, and daily activity of pirarucu (*Arapaima* spp.) in an Amazonian floodplain mosaic. *Freshwater Biology*, 64(7), 1255-1264.

Fang, T., Bodmer, R., Puertas, P., Pérez, P., Mayor, P. and Hayman, D. 2008. Certificación de pieles de pecaríes (*Tayassu tajacu* y *Tayassu pecari*): Una estrategia para la conservación y Manejo de Fauna en la Amazonía Peruana. 210 pp. Depósito Legal: N. 2008-08427. WUST EDICIONES.

Levi, T., Shepard Jr, G.H., Ohl-Schacherer, J., Wilmers, C.C., Peres, C.A. and Yu, D.W., 2011. Spatial tools for modeling the sustainability of subsistence hunting in tropical forests. *Ecological Applications*, 21(5), pp.1802-1818. <https://doi.org/10.1890/10-0375.1>

Loureiro, L.F., Gomes, L.P.L.N., Franco, C.L.B., Vasconcelos Neto, C.F.A.D. and Valsecchi, J., 2024. Traditional Territory in a Protected Area: Territorial Dynamics and Wildlife Management in the Amanã Sustainable Development Reserve, Amazonas, Brazil. *Sociedade & Natureza*, 36, p.e71004. <https://doi.org/10.14393/SN-v36-2024-71004x>

Novaro, A.J., Redford, K.H. and Bodmer, R.E., 2000. Effect of hunting in source-sink systems in the Neotropics. *Conservation Biology*, 14(3), pp.713-721. <https://doi.org/10.1046/j.1523-1739.2000.98452.x>

Oliveira, M.A., Calouro, A.M. 2019. Hunting agreements as a strategy for the conservation of species: the case of the Cazumbá-Iracema Extractive Reserve, state of Acre, Brazil *Oecologia Australis*, 23 (2), pp. 357-366, 10.4257/oeco.2019.2302.13

Vieira, M.A.R.M., de Castro, F., and Shepard, G. H. 2019. Who sets the rules? Institutional misfits and bricolage in hunting management in Brazil. *Human Ecology*, 47, 369-380.

Jose, J.P. & Salino, I.(2024). *Lemurs of Madagascar: A Review of Socio-Ecological Significance, Current Conservation Strategies, and Lessons for Future Conservation of the Species*. Current topics in Conservation Biology, Master of Science in Sustainable Development 2023-2025, Katholieke Universiteit Leuven, Belgium.

Du Toit, J.T. (2002). Wildlife harvesting guidelines for community-based wildlife management: a southern African perspective. *Biodiversity and Conservation* 11, 1403–1416. <https://doi.org/10.1023/A:1016263606704>

Paksi, A.K., Belani, T. O., & Hutami, A. N.(2023). The IUCN's Contribution to Supporting Nature Conservation Programs in Serengeti National Park. *Society*, 11(2), 255-274. <https://doi.org/10.33019/society.v11i2.501>

Lines 326-329: do you have examples to cite here?

326 To safeguard

327 Amazonia's forest health and sustain wildlife hunting, it is imperative to fully recognize the rights of

328 Amazonian Peoples to their lands, establish clear boundaries, and implement policies focusing on their

329 autonomy and governance.

The health of Amazonian peoples is deeply interconnected with the forests they depend on for sustenance, culture, and livelihoods. Deforestation directly impacts their well-being by reducing wildlife availability, a crucial protein and nutrient source for many Indigenous communities. As key species like peccaries and tapirs decline, nutritional deficiencies such as malnutrition and anemia become more prevalent. Additionally, forest degradation affects water quality by polluting rivers and streams with sedimentation and agricultural chemicals, increasing waterborne diseases. The loss of traditional lands also disrupts cultural and mental health, with displacement and a sense of identity loss contributing to anxiety, depression, and substance abuse, particularly among younger generations. Finally, deforestation exposes communities to greater risks of infectious diseases like malaria, dengue, and zoonotic infections, as the disturbance of natural ecosystems facilitates closer contact between humans and wildlife, further endangering public health. Please see Campos-Silva et al. (2021) for a more detailed discussion in local scale.

Moreover, protected areas managed by the Amazonian Peoples play a crucial role in maintaining usually harbor robust and ecologically functional animal populations of hunted species (Nepstad et al. 2006; Azevedo-Ramos et al. 2006; Benítez-López et al. 2017; Blackman et al. 2017; Schleicher et al. 2017; Garnett et al. 2018). Community-based management has successfully contributed to be instrumental in the recovery of animal species that were overexploited in the recent past of commercial exploitation (Campos-Silva et al. 2017, 2018), such as the manatee (Souza 2015), giant river otter (Lima et al. 2014), South American river turtle (Cantarelli et al. 2014), black caiman (Silveira & Thorbjarnarson) and pirarucu (arapaima) fish (Castello et al. 2009), which were once overexploited for commercial purposes (Antunes et al. 2016).

New text (Lines 289-304):

We provide new insights into the complex interplay of environmental, cultural and threatening factors shaping wildlife harvest patterns throughout Amazonia. The finer details of these interactions merit further locally focused research and avoid determinisms. While the Sustainable Development Goals and IUCN Red List of Threatened Species frameworks emphasise global policies and actions, we stress that wildlife management initiatives must be tailored to local ecological, cultural, socioeconomic, and vulnerability nuances, ideally focused on local and regional key hunted taxa and led by Amazonian Peoples. Wildlife management initiatives shaped by Amazonian Peoples' demands and cultural practices are more legitimate and likely to remain viable long-term. Indigenous Lands and Extractive Reserves usually maintain healthy populations of key hunted species⁴¹, even those once commercially over-harvested⁴². Indigenous and traditional knowledges are critical for assessing the conservation status of animal populations⁴³ and understanding key

ecological parameters⁴⁴, including their varying density⁴⁵, reproductive rates⁴⁶, and population dynamics²⁶. These factors, along with deforestation³⁴ and commercial hunting pressures⁴⁷, and climate change impacts³⁵, strongly influence sustainable harvest potentials. Our predictive heatmaps offer valuable spatial insights into harvest productivity, human-wildlife interactions, and human nutrition in Amazonia, supporting more effective policies integrating conservation and public health.

New text (Lines 314-326):

Amazonian systems of knowledges and practices are grounded in ontologies attributing agency, personhood, and humanity to various beings, including animals⁴⁹, which must be taken seriously in ecological assessments of traditional hunting and management systems⁵⁰. For Amazonian Peoples, relationships with wildlife are not based on resource extraction and nutrient provisioning but rather on social reciprocity governed by norms and ethical obligations⁵¹. Dietary restrictions and spatial avoidance function as wildlife management and conservation systems akin to species-specific protection, spatial zoning rules, and hunting bans⁵². These cultural practices are effective tools in regulating wildlife harvesting and the most legitimate management and conservation strategies in Amazonia⁵³. Wild meat is a critical food source and a social cornerstone that motivates Amazonian Peoples to safeguard their territories. The health of Amazonian ecosystems and wildlife is inextricably linked to the well-being of Amazonian Peoples, underscoring the importance of recognizing their land rights and supporting policies that enhance their autonomy and governance over their territories and biodiversity.

References

Antunes, A. P. et al. Empty forest or empty rivers? A century of commercial hunting in Amazonia. *Sci. Adv.* 2, e1600936 (2016).

Azevedo-Ramos C, Amaral BD, Nepstad DC, Soares Filho B, Nasi R. Integrating Ecosystem management, protected areas, and mammal conservation in the Brazilian Amazon. *Ecol Soc* 2006; **11**: 17

Benítez-López A, Santini L, Schipper AM, Busana M, Huijbregts MAJ. Intact but empty forests? Patterns of hunting-induced mammal defaunation in the tropics. *PNAS* 2017; **114**: 4123–4128.

Blackman A, Corral L, Lima ES, Asner GP. Titling indigenous communities protects forests in the Peruvian Amazon. *PNAS* 2017; **114**: 4123–4128.

Campos-Silva, JV, Peres CA, Antunes AP, Valsecchi J, Pezzuti J. Community-based population recovery of overexploited Amazonian wildlife. *Perspect Ecol Conserv* 2017; **15**: 266–270.

Campos-Silva JV, Hawes JE, Andrade PC, Peres CA. Unintended multispecies co-benefits of an Amazonian community-based conservation programme. *Nat Sustain* 2018; **1**: 650-656.

Campos-Silva, J. V., Peres, C. A., Hawes, J. E., Haugaasen, T., Freitas, C. T., Ladle, R. J., & Lopes, P. F. (2021). Sustainable-use protected areas catalyze enhanced livelihoods in rural Amazonia. *Proceedings of the National Academy of Sciences*, 118(40), e2105480118.

- Cantarelli VH, Malvasio A, Verdade LM. Brazil's *Podocnemis expansa* conservation program: Retrospective and future directions. *Chelonian Conserv Biol* 2014; **13**: 124–8.
- Castello L, Viana JP, Watkins G, Pinedo-Vasquez M, Luzadis VA. Lessons from integrating fishers of Arapaima in small-scale fisheries management at the Mamirauá Reserve, Amazon. *Environ Manag* 2009; **43**: 197–209.
- Garnett ST, Burgess ND, Fa JE, et al. A spatial overview of the global importance of indigenous lands for conservation. *Nat Sustain* 2018; **1**: 369–374.
- Lima DS, Marmontel M, Bernard E. Reoccupation of historical areas by the endangered giant river otter *Pteronura brasiliensis* (Carnivora: Mustelidae) in Central Amazonia, Brazil. *Mammalia* 2014; **78**: 177–184.
- Pimenta NC, Gonçalves ALS, Shepard GH, Macedo VW, Barnett APA. The return of giant otter to the Baniwa Landscape: A multi-scale approach to species recovery in the middle Içana River, Northwest Amazonia, Brazil. *Biol Conserv* 2018; **224**: 318–326.
- Schleicher J, Peres CA, Amano T, Llactayo W, Leader-Williams N. Conservation performance of different conservation governance regimes in the Peruvian Amazon. *Sci Rep* 2017; **7**: 11318.
- Souza DS. Amazonian manatee (*Trichechus inunguis* natterer 1883): mortality and habitat use in the Piagaçu-Purus sustainable development reserve, central Amazon, Brazil. Thesis [PhD in Freshwater Biology and Inland Fisheries] - National Institute of Amazonian Research, 2015.
- Schuster R, Germaine RR, Bennett JR, Reo NJ, Arcese P. Vertebrate biodiversity on indigenous-managed lands in Australia, Brazil, and Canada equals that in protected areas. *Environ Sci Policy* 2019; **101**: 1–6.
- Silveira R, Thorbjarnarson JB. Conservation implications of commercial hunting of black and spectacled caiman in the Mamirauá Sustainable Development Reserve, Brazil. *Biol Conserv* 1999; **88**: 103–109.

Referee #2

General Comments

This paper estimates overall harvests of terrestrial wild vertebrates in the Amazonian rainforest, employing meta-analysis based on an extensive literature set. The authors evaluated overall and species-specific harvest rates and total offtakes, listing 20 species predominantly hunted. The analysis is original and impressive, the statistical methods for estimating hunting pressure are rigorous, and the results shown are appropriate and informative. Introduction and Discussion are well-served, and the reference list is comprehensive. I only have two concerns, however.

We thank the reviewer for these positive remarks.

First, the analysis is explicit spatially but not temporally. The dates of data used for this analysis range from 1965 to 2024. Hunting patterns are assumed to have dramatically changed during the 60 years for a variety of reasons, including hunting techniques modernization and economic changes in rural areas. Why didn't the analysis involve this potential temporal variation? I think it is necessary to include this in the analysis by incorporating the data from long-term studies and, more indirectly, by associating the data with environmental and anthropogenic variables that correspond to the period of each survey.

We thank the reviewer for pointing out this issue and we agree that the analysis should consider temporal variation in anthropogenic and environmental factors for each data point. This was an issue that concerned us from the start of the analysis.

In the first version we had dealt with it using the average or median of the variables for the whole period:

Whenever a spatial variable had multiple temporal rasters, we built one with the average or median values. For instance, we obtained a single raster for both Annual NPP and GPP from the median of 22 years. In relation to the Proportion of Forest Cover variable, we used the annual mapping of 2005 land cover, since 2005 represents the averaged year of our ATH dataset. These techniques were aimed at minimising errors caused by temporal variation in spatial variables during the significant time window covered by our ATH dataset.

However, in this new version, we strived to obtain more time-specific information for all variables. Given the lack of data for some of them, and the high computational cost, we re-run the model using multiple averages for various specific periods considering the closest data available for each covariate in relation to each hunting study. We highly appreciate this suggestion as our models were greatly improved. We now added a new section in the Methods:

New text (Lines 638-652):

Temporal variation of spatial variables

None of the digital spatial variables fully cover the 1965–2024 period of the APTH dataset, preventing a year-by-year evaluation of their spatial and temporal effects on HHR and TSPO. To address this, we matched HHR and TSPO records to the closest available values of spatial variables in years where spatio-temporal data were available. This approach was feasible for modeling the effects of EVI, NPP, and GPP on HHR and TSPO by incorporating measurements at five-year intervals. We obtained Proportion of Habitat Loss data for 1985, 1990, 1995, 2000, 2005, 2010, 2015, and 2020. For EVI, NPP, and GPP, data were available from 2000 onwards at the same five-year intervals. HHR and TSPO values from 1965 to 1987 were assigned the 1985 habitat loss data; from 1988 to 1992, the 1990 data; from 1993 to 1998, the 1995 data, and so on, with post-2018 records assigned the 2020 data. A similar process was applied to EVI, NPP, and GPP, though for a shorter period: records from 1965 to 2002 were matched to the 2000 values, continuing at five-year intervals up to 2020. Despite the APTH dataset spanning 1965–2024, most hunting studies occurred around 2006±9 years (90% quantiles: 1995–2017), aligning well with the available temporal coverage of the spatial variables.

We also included cautious remarks in the main text regarding the lack of data for some variables:

New text (Lines 141-149):

While our findings provide robust insights, several limitations should be considered when interpreting these results. Although hunting data span from 1965 to 2024, spatial covariates are not uniformly available across the entire period. To partially address this, we matched HHR and TSPO records to the closest corresponding time period (see Methods). We also incorporated a variable to account for variation in recording effort across hunting studies (see Methods), which proved to be an important predictor in our HHR models. Despite potential limitations in capturing fine-scale local variability, the breadth and diversity of our dataset, combined with consistent spatiotemporal covariates, provide a strong foundation for a comprehensive assessment of wildlife harvesting dynamics throughout Amazonia.

and...

New text (Lines 207-213):

While our models adequately capture TSOP variation for key hunted taxa, they are less reliable for underrepresented species (Supplementary Data 2). In addition, hunting practices have evolved over the past 60 years, influenced by cultural and technological shifts. Increased rifle use, for example, has increased hunting efficiency, particularly for large species²⁴, while the growing use of flashlights has intensified the hunting pressure on nocturnal animals such as the paca²⁵, now the most hunted species in Amazonia. Our models also do not account for recently documented natural population cycles in white-lipped peccaries²⁶, which may influence hunting patterns over time.

Second, “potential consumption of wild meat” (LL248–269, Figure 4), is misleading. If I understand the analysis well, it assumes that all the harvest in a

given area (pixel?) is consumed within the same area, so it does not consider the trade and outside consumers (like tourists). This is a major limitation of this analysis, just as the authors also acknowledged (LL265–266), so I suggest that the authors replace the term “consumption” with “availability” and discuss the consumption with more explicit explanations of this limitation.

We agree with the reviewer and refer to it now as *available wild meat per rural inhabitant*.

Specific Comments

L48, indigenous: Capitalise “I”.

Done.

L131, human pressure (HP): The abbreviation HP is already used to refer to “Harvest Productivity” (L111), and I did not find the need for “human pressure” to have an abbreviation. Omit it?

Omitted.

Figure 1: I wanted to see the map with higher resolution (a larger map) to check the survey locations used in this study’s analysis. Can you include a more extensive map as a Supplementary Figure to show the locations with national borders?

For Figure 1, we have added a large map showing the locations of the primary and secondary hunting studies and specifying the locations with hunting effort measures used to calculate the HHR. Thus, we have removed the locations from the HP and offtake predictions maps in Figure 1 that were in the first version of the manuscript. Country borders, previously omitted, are shown now.

L173, (Fig. S2): A mistake to be “Fig. 2”?

Done.

L231, “US\$ 2.1 ... billion, based on the market value of beef in 2024”: Isn’t it overestimated? With my limited experience in the Colombian Amazon, wild meat is much cheaper than beef.

In our experience on-the-ground, the price of wild meat is usually between 60% and 75% lower than the price of beef in towns and villages in the interior of the Amazon. However, one of the most comprehensive and up-to-date references for a wildlife market in the Amazon, in Iquitos, Peru, calculated the average price for beef and wild meat in the same

markets, showing that both are sold at similar prices (i.e., beef; 4.81 (0.23 SD) USD vs wild meat; 5.79 (0.24 SD) USD; Mayor et al., 2022). The disparity in meat prices across much of Amazonia is not primarily driven by differences in meat quality, but rather by logistical constraints associated with transporting beef to remote areas and the widespread illegality of wild meat trade in the region. Given these factors, we opted to use the price of beef as a proxy to estimate the value of wild meat. Nonetheless, we have included a cautionary note in the revised text to acknowledge the limitations of this approach.

New text (Lines 219-224):

The annual monetary value of this wild meat production is approximately US\$ 2.3±0.7 (1.7–3.1) billion, based on 2024 beef market prices. Accurately assessing the economic value of wild meat remains challenging due to the often informal or illegal nature of the trade in most Amazonia. However, this hidden economic value of wild meat production suggests that traditional hunting is a significant ecosystem service for Amazonian Peoples, providing affordable, high-quality nutrition and reducing food expenditures.

New text (Lines 267-268):

Due to its high cost and the significant logistical challenges of producing, transporting, and storing in remote areas, beef is rarely consumed across most of Amazonia³⁷.

Reference

Mayor, P., El Bizri, H. R., Morcatty, T. Q., Moya, K., Bendayán, N., Solis, S., Vasconcelos Neto, C. F. A., Kirkland, M., Arevalo, O., Fang, T. G., Pérez-Peña, P. E., & Bodmer, R. E. (2021). Wild meat trade over the last 45 years in the Peruvian Amazon. *Conservation Biology*, 36(2), e13801. <https://doi.org/10.1111/cobi.13801>.

L436, Correct “Table X”.

Corrected.

L443, “geofenced”: typo

Corrected.

(Remarks on code availability): The code is not provided in the reviewer dataset.

We encountered technical difficulties in making the code fully available via Ocean Code due to our limited experience with the platform and its restricted computational allowance (a maximum of 30 processing hours per month). Given the spatial nature of our analyses, which often require several hours or even days to complete, we were unable to run and debug the entire code within the platform’s constraints. For this version, we will make available the Random Forest spatial models for HHR (Hunter Harvest Rate) and TSOP (Taxon-Specific Offtake Proportion), which form the basis of our results. We are prepared to share the remaining code and data directly with the reviewer for evaluation purposes if required.

Referee #3

This article summarizes the geographic extent, volume, and nutritional value of wild meat harvested across the nine countries of the Amazon. Its most notable feature is an extensive dataset covering 560 rural locations, including 341 with primary data, and incorporating spatial analysis of consumption patterns. The article also evaluates the nutritional significance of wild meat for rural inhabitants and provides detailed insights into the consumption patterns of the twenty most consumed species. By combining this dataset with modelling, the article offers a large-scale, regional perspective on wild meat consumption in the Amazon, which is unparalleled given that most previous studies have been localized. This analysis helps identify the challenges involved in conserving hunted species in the region.

This study is unique in its geographical scope and dataset size, enabling a comprehensive assessment of the importance of wild meat consumption for the nutrition of the rural Amazon population. By drawing from a broad sample of locations, it also highlights regional differences in wild meat consumption and explores the potential impacts on hunted fauna. No comparable studies exist for this region, and perhaps not globally.

We thank the reviewer for all the positive remarks.

The dataset consists of primary and secondary data from articles, theses, and various reports, spanning nearly six decades. As a result, the data vary in characteristics and potential accuracy, which is typical for evaluations of this scale. I believe the authors should more clearly emphasize this variation, as some methodological explanations are currently presented as if they were derived from primary sources (see, for example, lines 447-463).

We agree with the reviewer that additional detail was needed regarding the selection and validation of secondary data. This was, in fact, a concern from the outset of our research. We were aware that the different studies could have varying objectives and methodologies, potentially leading to inconsistencies in the data. For primary data studies, we contacted the coordinators of the initiatives whenever clarification was needed. In some cases, we also reached out to the authors of secondary studies, which led to some agreeing to share full datasets and join our research effort.

The methods of all secondary studies were carefully reviewed to determine whether their data could be reliably integrated into our database. This was particularly important for hunting effort data, which we used to calculate the Hunter Harvest Rate (HHR). Several effort-related metrics (e.g., number of hunters and hunting days) were excluded when their accuracy or coverage was uncertain.

While the HHR metric — animals hunted per hunter per day — provides a degree of standardization across studies (reflecting study effort rather than individual hunting effort), we took further steps to minimise potential bias. In this revised version of the manuscript,

we introduced an additional variable into the model to explicitly account for the collection effort associated with each hunting monitoring initiative, assuming that the duration of the study might affect its accuracy in reflecting the hunting activities.

Thus, we included the following description in the methods:

New text (Lines 624-627):

Hunting recording time span

A variable that controls the effort to record hunting in each study, that is, the time range in days in which hunted animals were recorded. This metric was only used to model HHR. We included this variable since we assumed that different time spans of hunting surveys could have different accuracies for data on the animals hunted. We obtained measures of the hunting recording time span from the APTH dataset.

New text (Lines 103-105):

For HHR modelling, we included a variable to account for the hunting registration effort, specifically the number of days over which hunted animals were recorded.

This variable was among the most important in our models for predicting HHR, improving our ability to better assess the effects of variables on HHR.

In this new version of the text, we have also added in the Methods section more information about how we selected, handled and validated the data, and made decisions regarding including it (or not) in our dataset, especially those from secondary sources, as follows:

New text (Lines 521-544):

Sources, selection, and validation of primary and secondary data

Given the diversity of secondary data sources spanning nearly six decades, our dataset naturally varied in objectives, methodological rigour, reporting standards, and cultural contexts. To ensure consistency, we focused on extracting comparable information across all studies.

We implemented a multi-step validation process to mitigate potential inaccuracies. Primary data served as a baseline for evaluating datasets and assessing methodological consistency, identifying discrepancies, and evaluating the reliability of secondary sources. We prioritised studies that provided clear methodological descriptions, reproducible metrics, or supplemental documentation and contacted the original authors or institutions when clarification was needed regarding sampling design and local conditions.

We scrutinized outliers and apparent inconsistencies by cross-referencing with more recent peer-reviewed sources or primary datasets from similar regions or time periods. When discrepancies were identified, we assessed whether they reflected genuine cultural differences, shifting hunting practices, or methodological shortcomings. Records showing implausible biological or cultural values were excluded.

Throughout the validation process, we remained attentive to cultural and practical factors – such as hunting laws, local traditions, and wildlife management strategies – which vary widely across regions and over time. To address this, we consulted local experts and researchers familiar with such nuances to confirm that data collection methods were

culturally appropriate and to verify that the underlying assumptions of each dataset remained valid. Only datasets meeting our standards for scientific rigour and comparability were integrated into the analysis and annotated with metadata. We also excluded studies that reported less than four hunted taxa, as these offered limited insights into species composition. Records lacking clearly described or reliable hunting effort methodologies were removed from the final dataset.

In my assessment, the article has no major flaws that would prevent its publication. However, one significant limitation is the exclusion of urban wild meat consumption, which presents a partial view of overall consumption in the Amazon. That said, the article appropriately focuses on the population subset for whom wild meat consumption is most important. Moreover, it is important to note that the absence of urban consumption data is a common issue for studies in the region, largely due to the illegality of the practice in Brazil. While some studies address urban consumption, they are few in number and may be biased, potentially impacting the article's calculation of total consumption.

We fully agree with the reviewer that urban consumption and trade are important dimensions that would greatly enrich our analysis. However, as the reviewer noted, reliable data on these aspects are largely absent or highly fragmented across the Amazon region. We now address this limitation explicitly in the revised manuscript, with a new section added to the Methods that discusses the challenges of incorporating urban data and the implications for interpreting our findings.

New text (Lines 615-623):

Wild meat trade and urban consumption data were excluded to maintain a focus on rural hunting practices and their ecological and nutritional implications. Including trade and urban consumption would have introduced complexities related to market dynamics, transportation, and intermediate processing, which are outside the scope of this study's subsistence-focused approach. Additionally, reliable trade and urban consumption data are often sparse or inconsistent in Amazonia, making integrating them into the models challenging without introducing significant uncertainty. Although the spatial process of excluding urban areas probably excludes rural or peri-urban inhabitants who may rely on wild meat for part of their diet, our database includes such people, analyzing the effects of urbanization on Harvest Productivity more reliably.

New text (Lines 84-92):

This paper examines the links between wildlife, human nutrition, and sustainable food systems in Amazonia. Through these analyses, we assess key indicators of ecosystem integrity—including species diversity, biomass availability, and habitat condition. Our findings show that deforestation and urbanisation reduce wild meat productivity and simplify the composition of hunted taxa, ultimately threatening traditional food systems and the nutritional well-being of Amazonian Peoples. Protecting Amazonia is therefore essential not only for conserving biodiversity but also for sustaining ecosystem functions and achieving multiple global SDGs, including those related to environmental and human health.

New text (Lines 262-266):

We also predicted higher offtake and lower HP near urban centres, raising concerns about the sustainability of hunting in these areas. High meat demand in densely populated peri-urban areas, coupled with declining wildlife populations³⁶, may shift rural diets toward cheaper domestic meats like chicken³⁷, which generally contains four times less iron, twice less B vitamins, and lower levels of protein and zinc than wild meat³⁸.

Another methodological concern relates to the visualization of findings in the maps, which are a central feature of the article. Many maps are difficult to read, even with a magnifying lens, and in some cases remain unclear despite magnification. I thus recommend improving the figures and the text within them, as these maps are likely the most critical aspect of the article.

We appreciate the suggestion and fully agree that the maps should be as clear as possible, given their central role in the manuscript. We have revised all the maps to enhance clarity and visual quality, and all figures are now provided in high resolution.

In Figure 1, we have included to the plate a larger overview map displaying the locations of both primary and secondary hunting studies, with specific indication of the sites for which hunting effort data were available and used to calculate the HHR. Consequently, we have removed the location dots from the HP and offtake prediction maps that were present in the initial version of the manuscript, as they are now represented in the new overview map.

APPROPRIATE USE OF STATISTICS AND TREATMENT OF UNCERTAINTIES

The methods are sufficiently detailed and, in general, well-described, but I believe two aspects could be improved. First, the article utilizes both primary and secondary data from various sources, which likely differ in quality. While some brief notes on limitations are provided, I found them insufficient. I recommend including a section that discusses the limitations of the dataset more thoroughly, particularly in terms of data quality, and spatial and temporal variation. This could be incorporated into the main text or, alternatively, in the appendix. Additionally, a more in-depth discussion of the potential limitations arising from the exclusion of urban data would be valuable.

We agree with the reviewer that clear explanations of limitations were lacking. We now added the following text both in the main text and Methods:

New text (Lines 141-149):

While our findings provide robust insights, several limitations should be considered when interpreting these results. Although hunting data span from 1965 to 2024, spatial covariates are not uniformly available across the entire period. To partially address this, we matched HHR and TSPO records to the closest corresponding time period (see Methods). We also incorporated a variable to account for variation in recording effort across hunting studies (see Methods), which proved to be an important predictor in our HHR models. Despite

potential limitations in capturing fine-scale local variability, the breadth and diversity of our dataset, combined with consistent spatiotemporal covariates, provide a strong foundation for a comprehensive assessment of wildlife harvesting dynamics throughout Amazonia.

New text (Lines 200-213):

*Cultural identity strongly influences HP and TSOP across taxa. For example, most Aruak Peoples value hunting ungulates, yet those in the Xingu River deliberately avoid them. In contrast, some Indigenous peoples in the northwestern Amazon highly value the often overlooked slender-legged treefrogs (*Osteocephalus* spp.). These patterns reflect a complex interplay of cultural preferences, environmental conditions, and ecological factors shaping hunting dynamics. Understanding these systems requires moving beyond environmental or cultural determinism toward an interdisciplinary perspective.*

While our models adequately capture TSOP variation for key hunted taxa, they are less reliable for underrepresented species (Supplementary Data 2). In addition, hunting practices have evolved over the past 60 years, influenced by cultural and technological shifts. Increased rifle use, for example, has increased hunting efficiency, particularly for large species²⁴, while the growing use of flashlights has intensified the hunting pressure on nocturnal animals such as the paca²⁵, now the most hunted species in Amazonia. Our models also do not account for recently documented natural population cycles in white-lipped peccaries²⁶, which may influence hunting patterns over time.

New text (Lines 292-321):

The fate of wild meat food systems in Amazonia

Our study highlights the essential role of traditional hunting and wild meat access in advancing several Sustainable Development Goals (SDGs) of The United Nations Convention on Biological Diversity, supporting nutritional security and health, helping reduce malnutrition, strengthening traditional food systems, and promoting sustainable wildlife use and ecosystem conservation across Amazonia (Extended Data Fig. 14). Although the sustainability of hunting in tropical forests and the risks of zoonotic diseases potentially linked to hunting have been intensively debated, we focus on understanding traditional hunting practices and wild meat access within the broader context of achieving the SDGs in Amazonia. We demonstrated that the health of the Amazon Forest is vital to sustaining traditional wild meat food systems and the nutrition of Amazonian Peoples. From this evidence, we contend that illegitimate proposals to ban, restrict, or replace wild meat without acknowledging its cultural and nutritional significance reflect a colonial mindset that undermines the autonomy and traditional food systems of Amazonian Peoples.

We provide new insights into the complex interplay of environmental, cultural and threatening factors shaping wildlife harvest patterns throughout Amazonia. The finer details of these interactions merit further locally focused research and avoid determinisms. While the Sustainable Development Goals and IUCN Red List of Threatened Species frameworks emphasise global policies and actions, we stress that wildlife management initiatives must be tailored to local ecological, cultural, socioeconomic, and vulnerability nuances, ideally focused on local and regional key hunted taxa and led by Amazonian Peoples. Wildlife management initiatives shaped by Amazonian Peoples' demands and

cultural practices are more legitimate and likely to remain viable long-term. Indigenous Lands and Extractive Reserves usually maintain healthy populations of key hunted species⁴¹, even those once commercially over-harvested⁴². Indigenous and traditional knowledges are critical for assessing the conservation status of animal populations⁴³ and understanding key ecological parameters⁴⁴, including their varying density⁴⁵, reproductive rates⁴⁶, and population dynamics²⁶. These factors, along with deforestation³⁴ and commercial hunting pressures⁴⁷, and climate change impacts³⁵, strongly influence sustainable harvest potentials. Our predictive heatmaps offer valuable spatial insights into harvest productivity, human-wildlife interactions, and human nutrition in Amazonia, supporting more effective policies integrating conservation and public health.

New text (Lines 638-652):

Temporal variation of spatial variables

None of the digital spatial variables fully cover the 1965–2024 period of the APTH dataset, preventing a year-by-year evaluation of their spatial and temporal effects on HHR and TSPO. To address this, we matched HHR and TSPO records to the closest available values of spatial variables in years where spatio-temporal data were available. This approach was feasible for modeling the effects of EVI, NPP, and GPP on HHR and TSPO by incorporating measurements at five-year intervals. We obtained Proportion of Habitat Loss data for 1985, 1990, 1995, 2000, 2005, 2010, 2015, and 2020. For EVI, NPP, and GPP, data were available from 2000 onwards at the same five-year intervals. HHR and TSPO values from 1965 to 1987 were assigned the 1985 habitat loss data; from 1988 to 1992, the 1990 data; from 1993 to 1998, the 1995 data, and so on, with post-2018 records assigned the 2020 data. A similar process was applied to EVI, NPP, and GPP, though for a shorter period: records from 1965 to 2002 were matched to the 2000 values, continuing at five-year intervals up to 2020. Despite the APTH dataset spanning 1965–2024, most hunting studies occurred around 2006±9 years (90% quantiles: 1995–2017), aligning well with the available temporal coverage of the spatial variables.

New text (Lines 615-623):

Wild meat trade and urban consumption data were excluded to maintain a focus on rural hunting practices and their ecological and nutritional implications. Including trade and urban consumption would have introduced complexities related to market dynamics, transportation, and intermediate processing, which are outside the scope of this study's subsistence-focused approach. Additionally, reliable trade and urban consumption data are often sparse or inconsistent in Amazonia, making integrating them into the models challenging without introducing significant uncertainty. Although the spatial process of excluding urban areas probably excludes rural or peri-urban inhabitants who may rely on wild meat for part of their diet, our database includes such people, analyzing the effects of urbanization on Harvest Productivity more reliably.

Second, in the methods section, it is not entirely clear whether all procedures were applied to both the primary and secondary data (e.g., but not limited to, lines 563-576). Clarifying the distinction between procedures applied to each data source would be beneficial.

We agree that this point required further clarification. As previously mentioned, we included a dedicated subsection in the Methods for each dataset, where we explain how the data were collated and handled. Once the APTH dataset was finalised, we applied the same analytical procedures uniformly across all datasets to ensure consistency in the analyses.

Third, I suggest the authors consider providing the data analysis coding to enhance the reproducibility of the results. A final note: I am not familiar with random forest models, so I am unable to evaluate this aspect properly.

We encountered technical difficulties in making the code fully available via Ocean Code due to our limited experience with the platform and its restricted computational allowance (a maximum of 30 processing hours per month). Given the spatial nature of our analyses, which often require several hours or even days to complete, we were unable to run and debug the entire code within the platform's constraints. For this version, we will make available the Random Forest spatial models for HHR (Hunter Harvest Rate) and TSOP (Taxon-Specific Offtake Proportion), which form the basis of our results. We are prepared to share the remaining code and data directly with the reviewer for evaluation purposes if required.

CONCLUSIONS

The results describe wild meat harvesting, the most commonly exploited species, and the nutritional contribution of macro- and micronutrients in the Amazon. While the conclusions that deforestation and population growth reduce access to wild meat are obviously not new, the overall assessment of the importance of wild meat in the region offers a novel perspective. The results also provide a clearer understanding of the immense challenges involved in controlling wild meat consumption in the region.

We thank the reviewer for these positive remarks.

SUGGESTED IMPROVEMENTS

Q.7 The authors have compiled an impressive spatial dataset that is both intriguing and likely to inspire further research, and they should be commended for this achievement. However, I have three main concerns, besides some more specific suggestions.

First, the unique contributions of this dataset need to be more clearly highlighted in the text. While the importance of wild animal hunting for local communities is emphasized, the article does not sufficiently explain how this dataset advances our understanding of Amazonian wild meat consumption or

provide novel insights beyond existing local studies. I believe it is crucial to clarify how this dataset enhances our understanding of issues related to wild meat consumption and nutritional effects that were previously unclear with local-scale data.

We agree with the reviewer. We added more information about the uniqueness of our data and the spatially-explicit analysis we did in the Introduction and last section of our paper, as follows:

New text (Lines 76-92):

Amazonia's unparalleled biodiversity and the deep reliance of rural peoples on it offer a unique opportunity to assess wildlife as a critical food source across a near-continental scale. While previous studies have documented hunting practices in specific localities or sub-regions, they have largely been too geographically limited to reveal broader spatial patterns or the cumulative impacts of large-scale environmental changes. In contrast, our dataset—spanning multiple Amazonian cultures and regions and covering six decades—offers the first spatially explicit, large-scale analysis of hunted animal diversity along with the environmental, cultural and threatening factors shaping wild meat productivity, availability, consumption, and nutritional contributions.

This paper examines the links between wildlife, human nutrition, and traditional food systems in Amazonia. We assess key indicators of ecosystem integrity—including species diversity, biomass availability, and environment condition. Our findings show that deforestation and urbanisation reduce wild meat productivity and simplify the composition of hunted taxa, ultimately threatening traditional food systems and the nutritional well-being of Amazonian Peoples. Protecting Amazonia is therefore essential not only for conserving biodiversity but also for sustaining ecosystem functions¹¹ and achieving multiple global the Sustainable Development Goals (SDGs) under the United Nations Convention on Biological Diversity, including those related to ecosystem and human health.

New text (Lines 276-288):

The fate of wild meat food systems in Amazonia

Our study highlights the essential role of traditional hunting and wild meat access in advancing several Sustainable Development Goals (SDGs) of The United Nations Convention on Biological Diversity, supporting nutritional security and health, helping reduce malnutrition, strengthening traditional food systems, and promoting sustainable wildlife use and ecosystem conservation across Amazonia (Extended Data Fig. 14). Although the sustainability of hunting in tropical forests and the risks of zoonotic diseases potentially linked to hunting have been intensively debated, we focus on understanding traditional hunting practices and wild meat access within the broader context of achieving the SDGs in Amazonia. We demonstrated that the health of the Amazon Forest is vital to sustaining traditional wild meat food systems and the nutrition of Amazonian Peoples. From this evidence, we contend that illegitimate proposals to ban, restrict, or replace wild meat without acknowledging its cultural and nutritional significance reflect a colonial mindset that undermines the autonomy and traditional food systems of Amazonian Peoples.

Second, although the discussion focuses on the role of wild meat in achieving Sustainable Development Goals, the most intriguing aspect of this dataset is its ability to reveal regional variability and the factors that explain it. This aspect is only minimally explored in the article. The numerous maps presented are only briefly interpreted, and the text does not fully reflect the extensive work undertaken. The authors could consider briefly summarizing the relevance to the Sustainable Development Goals in a short paragraph, and use the remaining space to provide a more in-depth discussion of the spatial variation.

We thank the reviewer for this valuable observation. We agree that one of the most compelling contributions of our dataset is indeed its potential to reveal regional variability in wild meat offtake and the underlying drivers of such patterns.

In response, we have now largely expanded the results and discussion to better interpret the spatial patterns presented in the maps, drawing attention to key regional contrasts in both harvest productivity and taxonomic composition. We also highlight potential explanatory factors, such as differences in habitat, deforestation and cultural aspects.

New text (Lines 120-149):

Random forest spatial models predicted higher HP in forested regions of Western Amazonia, along the mainstream of the Amazon River and its major tributaries, as well as in parts of the Guyana Shield and Southern-Western Amazonia (Fig. 1a, 1b). These areas are characterised by fertile soils, high primary productivity, low to moderate elevations, well-preserved forests, and relative isolation from large urban centres. Higher HP was also predicted in historically inhabited territories managed by the Waorani, Itonama, Movima, Warina, Kandoshi-Shapra, and Yurucare Indigenous peoples of Western Amazonia (Extended Data Fig. 3). Full details on the relationships between HHR and environmental, cultural and threatening factors, including variable importance, are provided in the Extended Data Figure 5 and Supplementary Data 1.

Our predicted HP patterns align with broader Amazonian ecosystem dynamics, particularly the region's geological history and the resulting variation in soil fertility, forest productivity and turnover, and species and functional composition. Enriched by Andean erosion, the younger and more fertile soils of Western Amazonia and floodplains support highly productive forests that prioritise reproductive processes (e.g., flower and fruit production) over photosynthesis¹³. This reproductive investment likely sustains larger and more diverse animal populations^{14,15} (Extended Data Fig. 6), contributing to higher HP in these regions.

Cultural identity emerged as a key predictor of HP across Amazonia (see Extended Data Fig. 5; Supplementary Data 1). With 511 distinct Indigenous Peoples¹⁶ speaking at least 335 languages¹⁷, Amazonia's remarkable cultural diversity reflects millennia of dynamic and reciprocal relationships with nature^{3,4}. Our results reveal, for the first time on a large spatial scale, how environmental, cultural and threatening factors interact to shape wildlife harvesting patterns across the region.

While our findings provide robust insights, several limitations should be considered when interpreting these results. Although hunting data span from 1965 to 2024, spatial covariates are not uniformly available across the entire period. To partially address this, we matched HHR and TSOP records to the closest corresponding time period (see Methods). We also

incorporated a variable to account for variation in recording effort across hunting studies (see Methods), which proved to be an important predictor in our HHR models. Despite potential limitations in capturing fine-scale local variability, the breadth and diversity of our dataset, combined with consistent spatiotemporal covariates, provide a strong foundation for a comprehensive assessment of wildlife harvesting dynamics throughout Amazonia.

New text (Lines 186-206):

*The lowland paca (*Cuniculus paca*) was the most hunted species by number of individuals in all sub-regions, generally followed by white-lipped peccary. However, regional differences are marked (Fig. 3; Extended Data Tables 4, 5). Ungulates dominate harvests in the Guyana Shield, Western and Southeastern Amazonia, while large rodents are more target in Central and Eastern Amazonia. Large primates, such as the woolly (*Lagothrix* spp.), spider (*Ateles* spp.), howler (*Alouatta* spp.), and capuchin (*Sapajus* spp.) monkeys, are more commonly hunted in Western Amazonia, whereas armadillos are more frequent in Eastern and Southeastern Amazonia.*

Environmental context also shapes hunting patterns. In flooded forests, Amazonian Peoples hunt higher proportions of river turtles, tortoises, caimans, howler monkeys, capybara, manatees, and waterfowl (ducks, cormorants, and egrets). In contrast, those living in upland terra firme forests, which cover 86% of Amazonia²¹, yield more tinamous, trumpeters, wood quails, guans, and armadillos are more commonly hunted (Extended Data Tables 6, 7). Many Amazonian Peoples have historically settled in transitional zones between flooded and upland areas²², taking advantage of seasonally complementary resources and adapting management strategies accordingly²³.

*Cultural identity strongly influences HP and TSOP across taxa. For example, most Aruak Peoples value hunting ungulates, yet those in the Xingu River deliberately avoid them. In contrast, some Indigenous peoples in the northwestern Amazon highly value the often overlooked slender-legged treefrogs (*Osteocephalus* spp.). These patterns reflect a complex interplay of cultural preferences, environmental conditions, and ecological factors shaping hunting dynamics. Understanding these systems requires moving beyond environmental or cultural determinism toward an interdisciplinary perspective.*

New text (Lines 245-253):

*Available wild meat per rural inhabitant was significantly lower in communities with (1) higher numbers of rural inhabitants, (2) closer proximity to urban centres, and (3) greater deforestation levels (Extended Data Fig. 13). In these more degraded areas, our TSOP spatial estimates indicate a shift in hunting patterns, with ecological generalists such as the nine-banded armadillo (*Dasypus novemcinctus*), capybara (*Hydrochoerus hydrochaeris*), guans (*Cracidae*, *Penelope* spp.), and pigeons (*Columbidae*) becoming proportionally more hunted than in better-conserved forests (Extended Data Table 8). In contrast, large atelid primates such as the woolly, spider and howler monkeys, which are quite vulnerable to the synergistic effects of deforestation and overhunting, are much less hunted in degraded areas.*

New text (Lines 289-313):

We provide new insights into the complex interplay of environmental, cultural and threatening factors shaping wildlife harvest patterns throughout Amazonia. The finer

details of these interactions merit further locally focused research and avoid determinisms. While the Sustainable Development Goals and IUCN Red List of Threatened Species frameworks emphasise global policies and actions, we stress that wildlife management initiatives must be tailored to local ecological, cultural, socioeconomic, and vulnerability nuances, ideally focused on local and regional key hunted taxa and led by Amazonian Peoples. Wildlife management initiatives shaped by Amazonian Peoples' demands and cultural practices are more legitimate and likely to remain viable long-term. Indigenous Lands and Extractive Reserves usually maintain healthy populations of key hunted species⁴¹, even those once commercially over-harvested⁴². Indigenous and traditional knowledges are critical for assessing the conservation status of animal populations⁴³ and understanding key ecological parameters⁴⁴, including their varying density⁴⁵, reproductive rates⁴⁶, and population dynamics²⁶. These factors, along with deforestation³⁴ and commercial hunting pressures⁴⁷, and climate change impacts³⁵, strongly influence sustainable harvest potentials. Our predictive heatmaps offer valuable spatial insights into harvest productivity, human-wildlife interactions, and human nutrition in Amazonia, supporting more effective policies integrating conservation and public health.

Scientific evidence demonstrates that traditional hunting in Amazonia has historically relied on sparse human populations and the limited spread of hunters across vast, conserved forests⁴⁷. This mechanism guarantees the preservation of large spatial refuges for terrestrial species, in contrast to aquatic species⁴⁷, which often require spatial zoning and local agreements to maintain healthy populations⁴⁸. While well-preserved forests combined with strong local governance can support sustained harvests through source-sink hunting dynamics, our findings show that forest degradation is increasingly threatening wild meat production, undermining unique traditional food systems. In these contexts, building agreements for hunting management may be critical to ensure Amazonian Peoples' well-being and the sustainability of the wildlife they rely on.

New text (Lines 581-590):

Despite being a relatively refined scale considering the size of the Amazon biome, this 10×10 km grain may not be fine enough for the different local contexts local-scale variations in habitat conditions, species distribution, and conservation needs. However, this scale provides a necessary balance between regional coverage and data availability, guaranteeing compatibility with widely used ecological and environmental datasets while allowing for broad-scale assessments that inform conservation planning. A finer resolution would significantly increase computational demands and face data limitations, particularly across such an extensive and heterogeneous region. The 10×10 km scale remains effective for identifying broader patterns, and its outputs can be complemented in the future by higher-resolution studies or downscaled using local ecological data to refine site-specific conservation actions.

Finally, the authors should more thoroughly acknowledge the limitations of their results and discuss how these limitations may affect the interpretation of their findings. Specifically, they could include remarks in the discussion or conclusions on the implications of missing urban consumption data. While I understand the challenges of obtaining accurate data due to the illegality or regulation of hunting in much of the region, it would be valuable to highlight how this limitation might lead to an underestimation of the data.

If space permits, another important aspect to highlight is the new research directions or questions suggested by the article's conclusions."

We agree with the reviewer that clear explanations of limitations were lacking. We now added the following text both in the main text and Methods:

New text (Lines 141-149):

While our findings provide robust insights, several limitations should be considered when interpreting these results. Although hunting data span from 1965 to 2024, spatial covariates are not uniformly available across the entire period. To partially address this, we matched HHR and TSOP records to the closest corresponding time period (see Methods). We also incorporated a variable to account for variation in recording effort across hunting studies (see Methods), which proved to be an important predictor in our HHR models. Despite potential limitations in capturing fine-scale local variability, the breadth and diversity of our dataset, combined with consistent spatiotemporal covariates, provide a strong foundation for a comprehensive assessment of wildlife harvesting dynamics throughout Amazonia.

New text (Lines 200-206):

*Cultural identity strongly influences HP and TSOP across taxa. For example, most Aruak Peoples value hunting ungulates, yet those in the Xingu River deliberately avoid them. In contrast, some Indigenous peoples in the northwestern Amazon highly value the often overlooked slender-legged treefrogs (*Osteocephalus* spp.). These patterns reflect a complex interplay of cultural preferences, environmental conditions, and ecological factors shaping hunting dynamics. Understanding these systems requires moving beyond environmental or cultural determinism toward an interdisciplinary perspective.*

New text (Lines 207-213):

While our models adequately capture TSOP variation for key hunted taxa, they are less reliable for underrepresented species (Supplementary Data 2). In addition, hunting practices have evolved over the past 60 years, influenced by cultural and technological shifts. Increased rifle use, for example, has increased hunting efficiency, particularly for large species²⁴, while the growing use of flashlights has intensified the hunting pressure on nocturnal animals such as the paca²⁵, now the most hunted species in Amazonia. Our models also do not account for recently documented natural population cycles in white-lipped peccaries²⁶, which may influence hunting patterns over time.

New text (Lines 289-304):

We provide new insights into the complex interplay of environmental, cultural and threatening factors shaping wildlife harvest patterns throughout Amazonia. The finer details of these interactions merit further locally focused research and avoid determinisms. While the Sustainable Development Goals and IUCN Red List of Threatened Species frameworks emphasise global policies and actions, we stress that wildlife management initiatives must be tailored to local ecological, cultural, socioeconomic, and vulnerability nuances, ideally focused on local and regional key hunted taxa and led by Amazonian Peoples. Wildlife management initiatives shaped by Amazonian Peoples' demands and cultural practices are more legitimate and likely to remain viable long-term. Indigenous

Lands and Extractive Reserves usually maintain healthy populations of key hunted species⁴¹, even those once commercially over-harvested⁴². Indigenous and traditional knowledges are critical for assessing the conservation status of animal populations⁴³ and understanding key ecological parameters⁴⁴, including their varying density⁴⁵, reproductive rates⁴⁶, and population dynamics²⁶. These factors, along with deforestation³⁴ and commercial hunting pressures⁴⁷, and climate change impacts³⁵, strongly influence sustainable harvest potentials. Our predictive heatmaps offer valuable spatial insights into harvest productivity, human-wildlife interactions, and human nutrition in Amazonia, supporting more effective policies integrating conservation and public health.

We totally agree with the reviewer that urban consumption and trade would be important to be considered in estimates, but as we pointed out, these data are largely lacking and sparse across Amazonia. We refer to this issue in a new text added in Methods:

New text (Lines 615-623):

Wild meat trade and urban consumption data were excluded to maintain a focus on rural hunting practices and their ecological and nutritional implications. Including trade and urban consumption would have introduced complexities related to market dynamics, transportation, and intermediate processing, which are outside the scope of this study's subsistence-focused approach. Additionally, reliable trade and urban consumption data are often sparse or inconsistent in Amazonia, making integrating them into the models challenging without introducing significant uncertainty. Although the spatial process of excluding urban areas probably excludes rural or peri-urban inhabitants who may rely on wild meat for part of their diet, our database includes such people, analyzing the effects of urbanization on Harvest Productivity more reliably.

New text (Lines 84-92):

This paper examines the links between wildlife, human nutrition, and sustainable food systems in Amazonia. Through these analyses, we assess key indicators of ecosystem integrity—including species diversity, biomass availability, and habitat condition. Our findings show that deforestation and urbanisation reduce wild meat productivity and simplify the composition of hunted taxa, ultimately threatening traditional food systems and the nutritional well-being of Amazonian Peoples. Protecting Amazonia is therefore essential not only for conserving biodiversity but also for sustaining ecosystem functions and achieving multiple global SDGs, including those related to environmental and human health.

New text (Lines 262-266):

We also predicted higher offtake and lower HP near urban centres, raising concerns about the sustainability of hunting in these areas. High meat demand in densely populated peri-urban areas, coupled with declining wildlife populations³⁶, may shift rural diets toward cheaper domestic meats like chicken³⁷, which generally contains four times less iron, twice less B vitamins, and lower levels of protein and zinc than wild meat³⁸.

Referee #4

General Comments

A: Summary of the Key Results

The paper presents a comprehensive study on sustainable wild meat harvesting in Amazonia, emphasizing its significance for nutritional security and ecosystem health. It quantifies the geographic extent, volumes, and nutritional value of wild meat harvested in Amazonia, demonstrating its critical role in meeting the dietary needs of remote rural populations. The study also highlights the negative impacts of deforestation on wild meat productivity and proposes the urgent need to preserve traditional food systems to achieve the Sustainable Development Goals (SDGs). The results indicate that wild meat plays a critical role in the nutritional security of rural and Indigenous communities in the Amazon, contributing significantly to their dietary intake of proteins and micronutrients like iron and B vitamins. Deforestation and proximity to urban areas, however, are linked to declining wild meat productivity, emphasizing the need for sustainable practices."

We thank the reviewer for these positive remarks.

B: Originality and Significance

The paper addresses a crucial and under-explored aspect of environmental conservation and food security, specifically in the Amazon region. While the study's scale and methodology are significant contributions, the topic itself is not entirely novel as similar studies have addressed wild meat consumption and its implications in other contexts. However, the large-scale dataset and the detailed spatial analysis make this study particularly valuable. The work significantly contributes to the discourse on sustainable resource management and conservation in Amazonia. While the subject of wild meat has been explored, the integration of sustainability and SDG impacts in the Amazon context is original.

We thank the reviewer for these positive remarks.

C: Data & Methodology

The methodology is robust, with a large dataset spanning several decades and a substantial geographic area. It comprises data on over 441,000 animal kills across 560 rural localities. The methodology, using Random Forest models to spatially predict hunting metrics, appears sound. However, details about some data sources (i.e., specific geographic variability) could be expanded. The use of

both primary and secondary data enhances the study's validity, though more granularity in the source of secondary data (e.g., author verification of older records) could improve reliability. The use of Random Forest models to predict hunting metrics and the comprehensive analysis of environmental and anthropogenic variables is appropriate and well-executed. The data is of high quality, and the presentation is clear, with thorough explanations of the methods used. However, more detailed information on how data from different sources were standardized and integrated would enhance the transparency of the approach.

We thank the reviewer for these positive remarks.

We agree with the reviewer that additional detail was needed regarding the selection and validation of secondary data. This was, in fact, a concern from the outset of our research. We were aware that the different studies could have varying objectives and methodologies, potentially leading to inconsistencies in the data. For primary data studies, we contacted the coordinators of the initiatives whenever clarification was needed. In some cases, we also reached out to the authors of secondary studies, which led to some agreeing to share full datasets and join our research effort.

The methods of all secondary studies were carefully reviewed to determine whether their data could be reliably integrated into our database. This was particularly important for hunting effort data, which we used to calculate the Hunter Harvest Rate (HHR). Several effort-related metrics (e.g., number of hunters and hunting days) were excluded when their accuracy or coverage was uncertain.

While the HHR metric—animals hunted per hunter per day—provides a degree of standardization across studies (reflecting study effort rather than individual hunting effort), we took further steps to minimise potential bias. In this revised version of the manuscript, we introduced an additional variable into the model to explicitly account for the collection effort associated with each hunting monitoring initiative, assuming that the duration of the study might affect its accuracy in reflecting the hunting activities.

Thus, we included the following description in the methods:

New text (Lines 624-627):

Hunting recording time span

A variable that controls the effort to record hunting in each study, that is, the time range in days in which hunted animals were recorded. This metric was only used to model HHR. We included this variable since we assumed that different time spans of hunting surveys could have different accuracies for data on the animals hunted. We obtained measures of the hunting recording time span from the APTH dataset.

New text (Lines 103-105):

For HHR modelling, we included a variable to account for the hunting registration effort, specifically the number of days over which hunted animals were recorded.

This variable was among the most important in our random forest models for predicting HHR, improving our ability to better assess the effects of variables on HHR.

In this new version of the text, we have also added in the Methods section more information about how we selected, handled and validated the data and included it (or not) in our dataset, especially those from secondary sources, as follows:

New text (Lines 521-544):

Sources, selection, and validation of primary and secondary data

Given the diversity of secondary data sources spanning nearly six decades, our dataset naturally varied in objectives, methodological rigour, reporting standards, and cultural contexts. To ensure consistency, we focused on extracting comparable information across all studies.

We implemented a multi-step validation process to mitigate potential inaccuracies. Primary data served as a baseline for evaluating datasets and assessing methodological consistency, identifying discrepancies, and evaluating the reliability of secondary sources. We prioritised studies that provided clear methodological descriptions, reproducible metrics, or supplemental documentation and contacted the original authors or institutions when clarification was needed regarding sampling design and local conditions.

We scrutinized outliers and apparent inconsistencies by cross-referencing with more recent peer-reviewed sources or primary datasets from similar regions or time periods. When discrepancies were identified, we assessed whether they reflected genuine cultural differences, shifting hunting practices, or methodological shortcomings. Records showing implausible biological or cultural values were excluded.

Throughout the validation process, we remained attentive to cultural and practical factors – such as hunting laws, local traditions, and wildlife management strategies – which vary widely across regions and over time. To address this, we consulted local experts and researchers familiar with such nuances to confirm that data collection methods were culturally appropriate and to verify that the underlying assumptions of each dataset remained valid. Only datasets meeting our standards for scientific rigour and comparability were integrated into the analysis and annotated with metadata. We also excluded studies that reported less than four hunted taxa, as these offered limited insights into species composition. Records lacking clearly described or reliable hunting effort methodologies were removed from the final dataset.

D: Appropriate Use of Statistics and Treatment of Uncertainties

The statistical methods used in the study are appropriate for the data and research questions. The treatment of uncertainties is well-managed, particularly through the use of confidence intervals and quantile predictions. The authors acknowledge the limitations of their spatial models and provide a balanced discussion of the uncertainties inherent in their estimates.

We thank the reviewer for these positive remarks.

E: Conclusions: Robustness, Validity, Reliability

The conclusions drawn are robust and supported by the data. The study effectively links wild meat harvesting to broader issues of nutritional security and ecosystem health. The discussion is well-grounded in the results, and the implications for policy and conservation are clearly articulated. However, the paper could benefit from a more explicit discussion of the potential for generalization beyond the Amazon region. Additionally, expanding on alternative conservation measures or sustainable practices could make the study more actionable.

Unfortunately, although we agree with the importance of including more information about generalization and conservation measures, we cannot include all these topics suggested by the reviewer due to limited space. Nevertheless, we have added a sentence that compares our results to those of the literature:

New text (Lines 230-239):

This level of available wild meat per rural inhabitant has the potential to meet the Dietary Reference Intakes (DRIs) for vitamin B12 across much of Amazonia, while also significantly contributing to DRIs for protein, iron, zinc, and other essential B vitamins and minerals (Fig. 4). The irreplaceable nutritional value of wild meat is particularly critical for 10.89 million rural inhabitants in Amazonia, providing a highly bioavailable source of protein, all essential amino acids, and vital micronutrients, often less accessible in plant-based foods^{27,28}. Higher wild meat availability was linked to better health among children not only in the Amazon²⁹ but also in the Congo Basin³⁰, including higher hemoglobin levels in children and increased household iron and zinc intake³¹. This is particularly noteworthy in regions where micronutrient deficiencies are widespread, compounded by malaria, intestinal parasites, and genetic disorders³².

References

The study cites a wide range of relevant literature, including both recent and seminal works on biodiversity, ecosystem services, and human nutrition. However, further integration of Indigenous knowledge sources or community-led research could strengthen the contextual relevance.

We thank the reviewer for these positive remarks. We included a few references on this topic, as below:

New text (Lines 289-304):

We provide new insights into the complex interplay of environmental, cultural and threatening factors shaping wildlife harvest patterns throughout Amazonia. The finer details of these interactions merit further locally focused research and avoid determinisms. While the Sustainable Development Goals and IUCN Red List of Threatened Species frameworks emphasise global policies and actions, we stress that wildlife management initiatives must be tailored to local ecological, cultural, socioeconomic, and vulnerability nuances, ideally focused on local and regional key hunted taxa and led by Amazonian

Peoples. Wildlife management initiatives shaped by Amazonian Peoples' demands and cultural practices are more legitimate and likely to remain viable long-term. Indigenous Lands and Extractive Reserves usually maintain healthy populations of key hunted species⁴¹, even those once commercially over-harvested⁴². Indigenous and traditional knowledges are critical for assessing the conservation status of animal populations⁴³ and understanding key ecological parameters⁴⁴, including their varying density⁴⁵, reproductive rates⁴⁶, and population dynamics²⁶. These factors, along with deforestation³⁴ and commercial hunting pressures⁴⁷, and climate change impacts³⁵, strongly influence sustainable harvest potentials. Our predictive heatmaps offer valuable spatial insights into harvest productivity, human-wildlife interactions, and human nutrition in Amazonia, supporting more effective policies integrating conservation and public health.

New text (Lines 314-326):

Amazonian systems of knowledges and practices are grounded in ontologies attributing agency, personhood, and humanity to various beings, including animals⁴⁹, which must be taken seriously in ecological assessments of traditional hunting and management systems⁵⁰. For Amazonian Peoples, relationships with wildlife are not based on resource extraction and nutrient provisioning but rather on social reciprocity governed by norms and ethical obligations⁵¹. Dietary restrictions and spatial avoidance function as wildlife management and conservation systems akin to species-specific protection, spatial zoning rules, and hunting bans⁵². These cultural practices are effective tools in regulating wildlife harvesting and the most legitimate management and conservation strategies in Amazonia⁵³. Wild meat is a critical food source and a social cornerstone that motivates Amazonian Peoples to safeguard their territories. The health of Amazonian ecosystems and wildlife is inextricably linked to the well-being of Amazonian Peoples, underscoring the importance of recognizing their land rights and supporting policies that enhance their autonomy and governance over their territories and biodiversity.

Suggested Improvements

Standardization of Data Sources: Provide more details on how data from various studies and regions were standardized for integration into the analysis.

Please see the response in C: Data & Methodology.

Spatial Resolution: Consider discussing the implications of the 10×10 km spatial resolution for local-scale conservation efforts.

We added this sentence in the Methods section:

New text (Lines 581-590):

Despite being a relatively refined scale considering the size of the Amazon biome, this 10×10 km grain may not be fine enough for the different local contexts local-scale variations in habitat conditions, species distribution, and conservation needs. However, this scale provides a necessary balance between regional coverage and data availability,

guaranteeing compatibility with widely used ecological and environmental datasets while allowing for broad-scale assessments that inform conservation planning. A finer resolution would significantly increase computational demands and face data limitations, particularly across such an extensive and heterogeneous region. The 10×10 km scale remains effective for identifying broader patterns, and its outputs can be complemented in the future by higher-resolution studies or downscaled using local ecological data to refine site-specific conservation actions.

In addition, in relation to the “*local-scale conservation efforts.*” and regional nuances, we highlight in the main text the following:

New text (Lines 120-149):

Random forest spatial models predicted higher HP in forested regions of Western Amazonia, along the mainstream of the Amazon River and its major tributaries, as well as in parts of the Guyana Shield and Southern-Western Amazonia (Fig. 1a, 1b). These areas are characterised by fertile soils, high primary productivity, low to moderate elevations, well-preserved forests, and relative isolation from large urban centres. Higher HP was also predicted in historically inhabited territories managed by the Waorani, Itonama, Movima, Warina, Kandoshi-Shapra, and Yurucare Indigenous peoples of Western Amazonia (Extended Data Fig. 3). Full details on the relationships between HHR and environmental, cultural and threatening factors, including variable importance, are provided in the Extended Data Figure 5 and Supplementary Data 1.

Our predicted HP patterns align with broader Amazonian ecosystem dynamics, particularly the region’s geological history and the resulting variation in soil fertility, forest productivity and turnover, and species and functional composition. Enriched by Andean erosion, the younger and more fertile soils of Western Amazonia and floodplains support highly productive forests that prioritise reproductive processes (e.g., flower and fruit production) over photosynthesis¹³. This reproductive investment likely sustains larger and more diverse animal populations^{14,15} (Extended Data Fig. 6), contributing to higher HP in these regions.

Cultural identity emerged as a key predictor of HP across Amazonia (see Extended Data Fig. 5; Supplementary Data 1). With 511 distinct Indigenous Peoples¹⁶ speaking at least 335 languages¹⁷, Amazonia’s remarkable cultural diversity reflects millennia of dynamic and reciprocal relationships with nature^{3,4}. Our results reveal, for the first time on a large spatial scale, how environmental, cultural and threatening factors interact to shape wildlife harvesting patterns across the region.

While our findings provide robust insights, several limitations should be considered when interpreting these results. Although hunting data span from 1965 to 2024, spatial covariates are not uniformly available across the entire period. To partially address this, we matched HHR and TSOP records to the closest corresponding time period (see Methods). We also incorporated a variable to account for variation in recording effort across hunting studies (see Methods), which proved to be an important predictor in our HHR models. Despite potential limitations in capturing fine-scale local variability, the breadth and diversity of our dataset, combined with consistent spatiotemporal covariates, provide a strong foundation for a comprehensive assessment of wildlife harvesting dynamics throughout Amazonia.

New text (Lines 186-206):

The lowland paca (Cuniculus paca) was the most hunted species by number of individuals in all sub-regions, generally followed by white-lipped peccary. However, regional differences are marked (Fig. 3; Extended Data Tables 4, 5). Ungulates dominate harvests in the Guyana Shield, Western and Southeastern Amazonia, while large rodents are more target in Central and Eastern Amazonia. Large primates, such as the woolly (Lagothrix spp.), spider (Ateles spp.), howler (Alouatta spp.), and capuchin (Sapajus spp.) monkeys, are more commonly hunted in Western Amazonia, whereas armadillos are more frequent in Eastern and Southeastern Amazonia.

Environmental context also shapes hunting patterns. In flooded forests, Amazonian Peoples hunt higher proportions of river turtles, tortoises, caimans, howler monkeys, capybara, manatees, and waterfowl (ducks, cormorants, and egrets). In contrast, those living in upland terra firme forests, which cover 86% of Amazonia²¹, yield more tinamous, trumpeters, wood quails, guans, and armadillos are more commonly hunted (Extended Data Tables 6, 7). Many Amazonian Peoples have historically settled in transitional zones between flooded and upland areas²², taking advantage of seasonally complementary resources and adapting management strategies accordingly²³.

Cultural identity strongly influences HP and TSOP across taxa. For example, most Aruak Peoples value hunting ungulates, yet those in the Xingu River deliberately avoid them. In contrast, some Indigenous peoples in the northwestern Amazon highly value the often overlooked slender-legged treefrogs (Osteocephalus spp.). These patterns reflect a complex interplay of cultural preferences, environmental conditions, and ecological factors shaping hunting dynamics. Understanding these systems requires moving beyond environmental or cultural determinism toward an interdisciplinary perspective.

New text (Lines 245-253):

Available wild meat per rural inhabitant was significantly lower in communities with (1) higher numbers of rural inhabitants, (2) closer proximity to urban centres, and (3) greater deforestation levels (Extended Data Fig. 13). In these more degraded areas, our TSOP spatial estimates indicate a shift in hunting patterns, with ecological generalists such as the nine-banded armadillo (Dasypus novemcinctus), capybara (Hydrochoerus hydrochaeris), guans (Cracidae, Penelope spp.), and pigeons (Columbidae) becoming proportionally more hunted than in better-conserved forests (Extended Data Table 8). In contrast, large atelid primates such as the woolly, spider and howler monkeys, which are quite vulnerable to the synergistic effects of deforestation and overhunting, are much less hunted in degraded areas.

New text (Lines 289-313):

We provide new insights into the complex interplay of environmental, cultural and threatening factors shaping wildlife harvest patterns throughout Amazonia. The finer details of these interactions merit further locally focused research and avoid determinisms. While the Sustainable Development Goals and IUCN Red List of Threatened Species frameworks emphasise global policies and actions, we stress that wildlife management initiatives must be tailored to local ecological, cultural, socioeconomic, and vulnerability nuances, ideally focused on local and regional key hunted taxa and led by Amazonian

Peoples. Wildlife management initiatives shaped by Amazonian Peoples' demands and cultural practices are more legitimate and likely to remain viable long-term. Indigenous Lands and Extractive Reserves usually maintain healthy populations of key hunted species⁴¹, even those once commercially over-harvested⁴². Indigenous and traditional knowledges are critical for assessing the conservation status of animal populations⁴³ and understanding key ecological parameters⁴⁴, including their varying density⁴⁵, reproductive rates⁴⁶, and population dynamics²⁶. These factors, along with deforestation³⁴ and commercial hunting pressures⁴⁷, and climate change impacts³⁵, strongly influence sustainable harvest potentials. Our predictive heatmaps offer valuable spatial insights into harvest productivity, human-wildlife interactions, and human nutrition in Amazonia, supporting more effective policies integrating conservation and public health.

Scientific evidence demonstrates that traditional hunting in Amazonia has historically relied on sparse human populations and the limited spread of hunters across vast, conserved forests⁴⁷. This mechanism guarantees the preservation of large spatial refuges for terrestrial species, in contrast to aquatic species⁴⁷, which often require spatial zoning and local agreements to maintain healthy populations⁴⁸. While well-preserved forests combined with strong local governance can support sustained harvests through source-sink hunting dynamics, our findings show that forest degradation is increasingly threatening wild meat production, undermining unique traditional food systems. In these contexts, building agreements for hunting management may be critical to ensure Amazonian Peoples' well-being and the sustainability of the wildlife they rely on.

Policy Implications: Expand on the policy recommendations, particularly on how to implement sustainable wild meat harvesting practices across different regions with varying ecological and socio-economic contexts.

Our conclusions, entitled “The fate of wild meat food systems in Amazonia”, were rewritten considering the practical consequences of our article in terms of public policies for wildlife management and the need to promote the engagement of Amazonian Peoples and value their ancestral practices of sustainable wildlife use.

New text (Lines 276-326):

The fate of wild meat food systems in Amazonia

Our study highlights the essential role of traditional hunting and wild meat access in advancing several Sustainable Development Goals (SDGs) of The United Nations Convention on Biological Diversity, supporting nutritional security and health, helping reduce malnutrition, strengthening traditional food systems, and promoting sustainable wildlife use and ecosystem conservation across Amazonia (Extended Data Fig. 14). Although the sustainability of hunting in tropical forests and the risks of zoonotic diseases potentially linked to hunting have been intensively debated, we focus on understanding traditional hunting practices and wild meat access within the broader context of achieving the SDGs in Amazonia. We demonstrated that the health of the Amazon Forest is vital to sustaining traditional wild meat food systems and the nutrition of Amazonian Peoples. From this evidence, we contend that illegitimate proposals to ban, restrict, or replace wild meat without acknowledging its cultural and nutritional significance reflect a colonial mindset that undermines the autonomy and traditional food systems of Amazonian Peoples.

We provide new insights into the complex interplay of environmental, cultural and threatening factors shaping wildlife harvest patterns throughout Amazonia. The finer details of these interactions merit further locally focused research and avoid determinisms. While the Sustainable Development Goals and IUCN Red List of Threatened Species frameworks emphasise global policies and actions, we stress that wildlife management initiatives must be tailored to local ecological, cultural, socioeconomic, and vulnerability nuances, ideally focused on local and regional key hunted taxa and led by Amazonian Peoples. Wildlife management initiatives shaped by Amazonian Peoples' demands and cultural practices are more legitimate and likely to remain viable long-term. Indigenous Lands and Extractive Reserves usually maintain healthy populations of key hunted species⁴¹, even those once commercially over-harvested⁴². Indigenous and traditional knowledges are critical for assessing the conservation status of animal populations⁴³ and understanding key ecological parameters⁴⁴, including their varying density⁴⁵, reproductive rates⁴⁶, and population dynamics²⁶. These factors, along with deforestation³⁴ and commercial hunting pressures⁴⁷, and climate change impacts³⁵, strongly influence sustainable harvest potentials. Our predictive heatmaps offer valuable spatial insights into harvest productivity, human-wildlife interactions, and human nutrition in Amazonia, supporting more effective policies integrating conservation and public health.

Scientific evidence demonstrates that traditional hunting in Amazonia has historically relied on sparse human populations and the limited spread of hunters across vast, conserved forests⁴⁷. This mechanism guarantees the preservation of large spatial refuges for terrestrial species, in contrast to aquatic species⁴⁷, which often require spatial zoning and local agreements to maintain healthy populations⁴⁸. While well-preserved forests combined with strong local governance can support sustained harvests through source-sink hunting dynamics, our findings show that forest degradation is increasingly threatening wild meat production, undermining unique traditional food systems. In these contexts, building agreements for hunting management may be critical to ensure Amazonian Peoples' well-being and the sustainability of the wildlife they rely on.

Amazonian systems of knowledges and practices are grounded in ontologies attributing agency, personhood, and humanity to various beings, including animals⁴⁹, which must be taken seriously in ecological assessments of traditional hunting and management systems⁵⁰. For Amazonian Peoples, relationships with wildlife are not based on resource extraction and nutrient provisioning but rather on social reciprocity governed by norms and ethical obligations⁵¹. Dietary restrictions and spatial avoidance function as wildlife management and conservation systems akin to species-specific protection, spatial zoning rules, and hunting bans⁵². These cultural practices are effective tools in regulating wildlife harvesting and the most legitimate management and conservation strategies in Amazonia⁵³. Wild meat is a critical food source and a social cornerstone that motivates Amazonian Peoples to safeguard their territories. The health of Amazonian ecosystems and wildlife is inextricably linked to the well-being of Amazonian Peoples, underscoring the importance of recognizing their land rights and supporting policies that enhance their autonomy and governance over their territories and biodiversity.

Impact of Climate Change: Include a more detailed analysis of how climate change might further impact wild meat availability and ecosystem health in Amazonia.

Although we acknowledge that the impacts of climate change can be severe for both wildlife and the human populations that rely on it, a comprehensive discussion of this topic falls outside the scope of the present article. For this reason, we limit our treatment of the issue to brief mentions when it was relevant in the context of our findings.

New text (Lines 254-261):

Large-scale agriculture, land grabbing, logging, mining, and infrastructure development have led to deforestation³³, increasingly undermining Amazonian People's reliance on biodiversity. The combined effects of deforestation and wildlife overharvesting have produced simplified animal assemblages, with lower species richness and fewer large-bodied species³⁴. These pressures are further compounded by recent climatic changes, including more frequent floods, droughts, and large-scale fires³³, all of which threaten both habitats and many key hunted taxa³⁵. We recommend that further studies should investigate the future impacts of deforestation, wildfire, and climate change on animal populations and on the supply of wild meat.

New text (Lines 289-304):

We provide new insights into the complex interplay of environmental, cultural and threatening factors shaping wildlife harvest patterns throughout Amazonia. The finer details of these interactions merit further locally focused research and avoid determinisms. While the Sustainable Development Goals and IUCN Red List of Threatened Species frameworks emphasise global policies and actions, we stress that wildlife management initiatives must be tailored to local ecological, cultural, socioeconomic, and vulnerability nuances, ideally focused on local and regional key hunted taxa and led by Amazonian Peoples. Wildlife management initiatives shaped by Amazonian Peoples' demands and cultural practices are more legitimate and likely to remain viable long-term. Indigenous Lands and Extractive Reserves usually maintain healthy populations of key hunted species⁴¹, even those once commercially over-harvested⁴². Indigenous and traditional knowledges are critical for assessing the conservation status of animal populations⁴³ and understanding key ecological parameters⁴⁴, including their varying density⁴⁵, reproductive rates⁴⁶, and population dynamics²⁶. These factors, along with deforestation³⁴ and commercial hunting pressures⁴⁷, and climate change impacts³⁵, strongly influence sustainable harvest potentials. Our predictive heatmaps offer valuable spatial insights into harvest productivity, human-wildlife interactions, and human nutrition in Amazonia, supporting more effective policies integrating conservation and public health.

Explicit mention of obtaining Free, Prior, and Informed Consent (FPIC) from Indigenous communities.

We provide a broad reflection on questions related to the consent of Indigenous and Traditional Peoples, their participation in the monitoring initiatives that generated the hunting data, the use of these data at both local and large scales, and their involvement and

authorship in this publication, in response to reviewer #6, whose comments focused on these matters.

As a more concise and direct response to the reviewer's specific query, we would like to emphasize that the primary data used in this paper were collected through long-term, community-based monitoring initiatives that were designed and implemented with strong engagement from Indigenous and Traditional Peoples, including community leaders and hunters. These initiatives were developed to support local decision-making on matters of direct concern to the communities themselves.

While each initiative involved distinct forms of collaboration among stakeholders, they all emerged from long-standing relationships between Indigenous and Traditional communities and non-Indigenous researchers or organizations. These collaborations were often embedded in broader sociopolitical contexts that extended well beyond academic purposes. As such, the outcomes of these partnerships, all with local communities' consent and deep involvement, have contributed to diverse interests at multiple levels.

Regarding the use of hunting data from these initiatives for academic purposes, all initiatives and methods of data collection were reviewed and approved by academic or governmental ethics committees, as detailed in Extended Data Table 9.

New text (Lines 933-964):

Research collaboration and ethics

The community-based hunting monitoring initiatives that contribute primary data to the Amazonian Peoples Traditional Hunting (APTH) dataset were developed and implemented through close partnerships with Indigenous and Traditional Peoples. These initiatives should not be seen as conventional academic research, where external researchers extract data on biodiversity use. Instead, they are grounded in sociopolitical realities relevant to Indigenous and Traditional communities and are designed to strengthen livelihoods, empower communities, support wildlife conservation and management, enhance food security, and protect cultural practices—always with respect for their priorities and autonomy. The data collected directly informs community-led decisions on sustainable wildlife and territorial management, with meaningful community engagement throughout the monitoring cycle. Results are transparently shared at the community level and used to guide practical actions on the ground. Each project is tailored to the specific needs of the communities involved, ensuring that traditional knowledge is respected, safeguarded, and valued. Our collaborative approach includes training Indigenous and Traditional hunters and researchers, fostering long-term capacity building and education. Wildlife monitoring programs are conducted under government regulation, ensuring ethical data handling and confidentiality. All research activities are formally approved by Indigenous and Traditional communities, as well as by relevant academic or governmental institutions overseeing Indigenous Lands, parks, and extractive reserves.

Data-sharing agreements were established among communities, researchers, and technicians, enabling informed local decision-making and advancing research on wildlife use, management, and conservation in Amazonia. Free, Prior, and Informed Consent (FPIC) was obtained—either orally or in writing—from all communities participating in those initiatives, ensuring ethical engagement and respect for rights, welfare, and autonomy. While some early initiatives predated formal non-indigenous and local ethics committees (e.g., before the Nagoya Protocol, 2010), agreements were always culturally

adapted, ranging from oral consensus to written contracts detailing research objectives, participant rights, and data use (Extended Data Table 9).

Six independent ethics committees reviewed and approved all primary data methods, with the 12 contributing initiatives receiving clearance from institutional review boards across Brazilian, Peruvian, and French Guianese Amazonia (Extended Data Table 9). These approvals, secured through universities and research institutions, guaranteed compliance with international ethical standards for research involving Indigenous and Traditional Peoples.

Clarification is needed on data collection protocols in areas with significant geographic or cultural variability.

We added this paragraph in the Methods:

New text (Lines 536-544):

Throughout the validation process, we remained attentive to cultural and practical factors – such as hunting laws, local traditions, and wildlife management strategies – which vary widely across regions and over time. To address this, we consulted local experts and researchers familiar with such nuances to confirm that data collection methods were culturally appropriate and to verify that the underlying assumptions of each dataset remained valid. Only datasets meeting our standards for scientific rigour and comparability were integrated into the analysis and annotated with metadata. We also excluded studies that reported less than four hunted taxa, as these offered limited insights into species composition. Records lacking clearly described or reliable hunting effort methodologies were removed from the final dataset.

More discussion on the limitations of model-based predictions is necessary, especially in heterogenous environments like the Amazon.

Please see our response to the following issue: *Spatial Resolution: Consider discussing the implications of the 10×10 km spatial resolution for local-scale conservation efforts.* We understand the importance of regional and local variations in hunting productivity and the availability of wild meat. Wildlife management cannot be reliant on general hunting patterns in the Amazon. Our large-scale work still manages to show regional patterns, but we admit that it can be risky to use our results to predict local patterns. These nuances deserve local analytical approaches. We now address these concerns at various points in the text. We have added text to address this issue, both in terms of harvest productivity and composition of hunted taxa, as follows:

New text (Lines 141-149):

While our findings provide robust insights, several limitations should be considered when interpreting these results. Although hunting data span from 1965 to 2024, spatial covariates are not uniformly available across the entire period. To partially address this, we matched HHR and TSOP records to the closest corresponding time period (see Methods). We also incorporated a variable to account for variation in recording effort across hunting studies (see Methods), which proved to be an important predictor in our HHR models. Despite

potential limitations in capturing fine-scale local variability, the breadth and diversity of our dataset, combined with consistent spatiotemporal covariates, provide a strong foundation for a comprehensive assessment of wildlife harvesting dynamics throughout Amazonia.

New text (Lines 200-213):

*Cultural identity strongly influences HP and TSOP across taxa. For example, most Aruak Peoples value hunting ungulates, yet those in the Xingu River deliberately avoid them. In contrast, some Indigenous peoples in the northwestern Amazon highly value the often overlooked slender-legged treefrogs (*Osteocephalus* spp.). These patterns reflect a complex interplay of cultural preferences, environmental conditions, and ecological factors shaping hunting dynamics. Understanding these systems requires moving beyond environmental or cultural determinism toward an interdisciplinary perspective.*

While our models adequately capture TSOP variation for key hunted taxa, they are less reliable for underrepresented species (Supplementary Data 2). In addition, hunting practices have evolved over the past 60 years, influenced by cultural and technological shifts. Increased rifle use, for example, has increased hunting efficiency, particularly for large species²⁴, while the growing use of flashlights has intensified the hunting pressure on nocturnal animals such as the paca²⁵, now the most hunted species in Amazonia. Our models also do not account for recently documented natural population cycles in white-lipped peccaries²⁶, which may influence hunting patterns over time.

New text (Lines 304-319):

We provide new insights into the complex interplay of environmental, cultural and threatening factors shaping wildlife harvest patterns throughout Amazonia. The finer details of these interactions merit further locally focused research and avoid determinisms. While the Sustainable Development Goals and IUCN Red List of Threatened Species frameworks emphasise global policies and actions, we stress that wildlife management initiatives must be tailored to local ecological, cultural, socioeconomic, and vulnerability nuances, ideally focused on local and regional key hunted taxa and led by Amazonian Peoples. Wildlife management initiatives shaped by Amazonian Peoples' demands and cultural practices are more legitimate and likely to remain viable long-term. Indigenous Lands and Extractive Reserves usually maintain healthy populations of key hunted species⁴¹, even those once commercially over-harvested⁴². Indigenous and traditional knowledges are critical for assessing the conservation status of animal populations⁴³ and understanding key ecological parameters⁴⁴, including their varying density⁴⁵, reproductive rates⁴⁶, and population dynamics²⁶. These factors, along with deforestation³⁴ and commercial hunting pressures⁴⁷, and climate change impacts³⁵, strongly influence sustainable harvest potentials. Our predictive heatmaps offer valuable spatial insights into harvest productivity, human-wildlife interactions, and human nutrition in Amazonia, supporting more effective policies integrating conservation and public health.

Clarity and Context

The paper is generally clear and well-structured. The abstract provides a succinct summary of the research, and the introduction effectively sets out the context. However, some sections, particularly those detailing the methodology,

could be streamlined for better readability. In addition, the conclusion could more clearly reiterate the main findings and their implications for policy and future research.

Overall, this paper is a significant contribution to the field of environmental conservation and sustainable resource management in Amazonia. With some refinements, it could be an even more powerful document for influencing policy and conservation practices.

We thank the reviewer for these positive remarks. Our conclusion, titled “The fate of wild meat food systems in Amazonia”, has been restructured for greater assertiveness and applicability.

Ethics

The study demonstrates a solid effort in ethically working with Indigenous communities across the Amazon region. The inclusion of Indigenous collaborators, such as hunters from multiple Indigenous groups, suggests an approach aligned with respectful and collaborative research. Data collection methods appear to prioritize community engagement, involving local researchers and community members. The monitoring schemes seem to have been designed with cultural sensitivity, which is crucial for maintaining ethical standards. However, the paper would benefit from explicitly stating whether free, prior, and informed consent (FPIC) was obtained and if there were specific protocols to ensure how Indigenous knowledge was respected and protected. Including such details would strengthen its ethical grounding.

The paper would benefit from a more explicit discussion of the ethical standards and guidelines followed during the study. For instance, it should clarify how informed consent was obtained from participants and whether any specific ethical approvals were sought from local or national bodies overseeing research involving Indigenous communities. In addition, the paper could address how the data will be used and shared in a way that benefits the communities involved, ensuring that they are not merely subjects of study but active participants in the research process.

Hunting monitoring initiatives that provided data for the APTH dataset are located in widely dispersed regions across Amazonia in highly variable periods, from the early 1990s to the present. These initiatives were designed and implemented to address local issues from each village, community, or region, aiming to improve the quality of life of Indigenous and Traditional Peoples, conserve their wildlife, enhance their food security, and preserve their livelihoods, and respecting their decisions. Because of this characteristic, Indigenous and Traditional Peoples and their collaborators have primarily used the data at the local scale, to understand local wildlife use and support the management of their territories and natural resources. Each working group, within its area of activity, has provided feedback on the results at the local level; this activity was already undertaken by each initiative prior to the start of this study.

New text (Lines 933-964):

Research collaboration and ethics

The community-based hunting monitoring initiatives that contribute primary data to the Amazonian Peoples Traditional Hunting (APTH) dataset were developed and implemented through close partnerships with Indigenous and Traditional Peoples. These initiatives should not be seen as conventional academic research, where external researchers extract data on biodiversity use. Instead, they are grounded in sociopolitical realities relevant to Indigenous and Traditional communities and are designed to strengthen livelihoods, empower communities, support wildlife conservation and management, enhance food security, and protect cultural practices—always with respect for their priorities and autonomy. The data collected directly informs community-led decisions on sustainable wildlife and territorial management, with meaningful community engagement throughout the monitoring cycle. Results are transparently shared at the community level and used to guide practical actions on the ground. Each project is tailored to the specific needs of the communities involved, ensuring that traditional knowledge is respected, safeguarded, and valued. Our collaborative approach includes training Indigenous and Traditional hunters and researchers, fostering long-term capacity building and education. Wildlife monitoring programs are conducted under government regulation, ensuring ethical data handling and confidentiality. All research activities are formally approved by Indigenous and Traditional communities, as well as by relevant academic or governmental institutions overseeing Indigenous Lands, parks, and extractive reserves.

Data-sharing agreements were established among communities, researchers, and technicians, enabling informed local decision-making and advancing research on wildlife use, management, and conservation in Amazonia. Free, Prior, and Informed Consent (FPIC) was obtained—either orally or in writing—from all communities participating in those initiatives, ensuring ethical engagement and respect for rights, welfare, and autonomy. While some early initiatives predated formal non-indigenous and local ethics committees (e.g., before the Nagoya Protocol, 2010), agreements were always culturally adapted, ranging from oral consensus to written contracts detailing research objectives, participant rights, and data use (Extended Data Table 9).

Six independent ethics committees reviewed and approved all primary data methods, with the 12 contributing initiatives receiving clearance from institutional review boards across Brazilian, Peruvian, and French Guianese Amazonia (Extended Data Table 9). These approvals, secured through universities and research institutions, guaranteed compliance with international ethical standards for research involving Indigenous and Traditional Peoples.

Specific Comments

Title: The title suggests that the article's primary focus is on food security and ecosystem health. However, the article primarily discusses estimates of consumption and the contribution of macro- and micronutrients. The connection between these results and ecosystem health is not clearly established in the text and does not seem to be a central aspect of the article. The text should either clarify this connection or the title should more accurately reflect the article's content. Additionally, one of the article's main contributions is the spatialization of the data, which is not adequately reflected in the title.

We agree with the reviewer's concern about the title. We have changed it to “Healthy forests safeguard traditional wild meat food systems in Amazonia”. Healthy forests would be a synonym for a healthy ecosystem or a conserved forest. We understand that the integrity of the forest is what guarantees the preservation of the ecosystem and its diversity of wildlife and culture. Our data supports that once the forest is lost or urbanized, harvest productivity and the availability of wild meat decrease, and the composition of hunted species simplifies. For these reasons, we opted for a short, general title with a strong message. Instead of giving specific details in the title, we opted for a broad title, as the article does.

Page 3, Lines 93-101: Could you also highlight how the study contributes to our scientific understanding or current debates? Specifically, how does this spatial approach advance our existing knowledge?

New text (Lines 76-92):

Amazonia’s unparalleled biodiversity and the deep reliance of rural peoples on it offer a unique opportunity to assess wildlife as a critical food source across a near-continental scale. While previous studies have documented hunting practices in specific localities or sub-regions, they have largely been too geographically limited to reveal broader spatial patterns or the cumulative impacts of large-scale environmental changes. In contrast, our dataset—spanning multiple Amazonian cultures and regions and covering six decades—offers the first spatially explicit, large-scale analysis of hunted animal diversity along with the environmental, cultural and threatening factors shaping wild meat productivity, availability, consumption, and nutritional contributions.

This paper examines the links between wildlife, human nutrition, and traditional food systems in Amazonia. We assess key indicators of ecosystem integrity—including species diversity, biomass availability, and environment condition. Our findings show that deforestation and urbanisation reduce wild meat productivity and simplify the composition of hunted taxa, ultimately threatening traditional food systems and the nutritional well-being of Amazonian Peoples. Protecting Amazonia is therefore essential not only for conserving biodiversity but also for sustaining ecosystem functions¹¹ and achieving multiple global the Sustainable Development Goals (SDGs) under the United Nations Convention on Biological Diversity, including those related to ecosystem and human health.

Page 3, line 120: Does this number represent the total across all decades combined, or is it the current figure?

This is the current figure of the number of hunters in Amazonia.

Page 3, line 126-129: this part is not very clear.

We agree with the reviewer, and we changed the text accordingly:

New text (Lines 120-140):

Random forest spatial models predicted higher HP in forested regions of Western Amazonia, along the mainstream of the Amazon River and its major tributaries, as well as in parts of the Guyana Shield and Southern-Western Amazonia (Fig. 1a, 1b). These areas are characterised by fertile soils, high primary productivity, low to moderate elevations, well-preserved forests, and relative isolation from large urban centres. Higher HP was also predicted in historically inhabited territories managed by the Waorani, Itonama, Movima, Warina, Kandoshi-Shapra, and Yurucare Indigenous peoples of Western Amazonia (Extended Data Fig. 3). Full details on the relationships between HHR and environmental, cultural and threatening factors, including variable importance, are provided in the Extended Data Figure 5 and Supplementary Data 1.

Our predicted HP patterns align with broader Amazonian ecosystem dynamics, particularly the region's geological history and the resulting variation in soil fertility, forest productivity and turnover, and species and functional composition. Enriched by Andean erosion, the younger and more fertile soils of Western Amazonia and floodplains support highly productive forests that prioritise reproductive processes (e.g., flower and fruit production) over photosynthesis¹³. This reproductive investment likely sustains larger and more diverse animal populations^{14,15} (Extended Data Fig. 6), contributing to higher HP in these regions.

Cultural identity emerged as a key predictor of HP across Amazonia (see Extended Data Fig. 5; Supplementary Data 1). With 511 distinct Indigenous Peoples¹⁶ speaking at least 335 languages¹⁷, Amazonia's remarkable cultural diversity reflects millennia of dynamic and reciprocal relationships with nature^{3,4}. Our results reveal, for the first time on a large spatial scale, how environmental, cultural and threatening factors interact to shape wildlife harvesting patterns across the region.

Page 3, Line 131: Is HP the acronym for both hunter productivity (see line 111) and higher predicted human pressure?

The reviewer is right, and HP was used incorrectly to refer to human pressure because the abbreviation is used in the article to refer to harvest productivity, "the potential number of individuals and biomass that can be harvested by a hunter across Amazonia."

Page 4, Line 146: Are these estimates derived from literature specifically focused on the Amazon?

Yes, these figures are derived from the literature and include estimates for the Amazon Basin only.

Page 4 lines 147-148: What specific aspects are affected by this limitation? It is unclear to mention several problematic factors without explaining how they may impact the results.

We agree that the limitations should be better explained in the paper. We included two paragraphs on limitations that read as follow:

New text (Lines 141-149):

While our findings provide robust insights, several limitations should be considered when interpreting these results. Although hunting data span from 1965 to 2024, spatial covariates are not uniformly available across the entire period. To partially address this, we matched HHR and TSOP records to the closest corresponding time period (see Methods). We also incorporated a variable to account for variation in recording effort across hunting studies (see Methods), which proved to be an important predictor in our HHR models. Despite potential limitations in capturing fine-scale local variability, the breadth and diversity of our dataset, combined with consistent spatiotemporal covariates, provide a strong foundation for a comprehensive assessment of wildlife harvesting dynamics throughout Amazonia.

New text (Lines 207-213):

While our models adequately capture TSOP variation for key hunted taxa, they are less reliable for underrepresented species (Supplementary Data 2). In addition, hunting practices have evolved over the past 60 years, influenced by cultural and technological shifts. Increased rifle use, for example, has increased hunting efficiency, particularly for large species²⁴, while the growing use of flashlights has intensified the hunting pressure on nocturnal animals such as the paca²⁵, now the most hunted species in Amazonia. Our models also do not account for recently documented natural population cycles in white-lipped peccaries²⁶, which may influence hunting patterns over time.

Page 4, line 150: ...in according or ..., according?

Changed.

Figures (all): I believe the font size in most of the figures in the main text and supplementary material should be larger. I had to use a magnifying glass to read them, and for some graphs in the appendix, even the magnifying glass was not sufficient.

We appreciate the suggestion and agree that the maps should be clear, as they represent a central feature of the article. We have improved all the maps, making them cleaner and clearer. For this version of the manuscript, we provide all the figures in high resolution.

Page 7, Lines 209-224: This section on the historical reliance of indigenous people seems out of context and does not provide sufficiently relevant information. Do you have current data to assess whether indigenous hunting patterns vary by cultural group or in comparison to non-indigenous people? Additionally, this text appears under the section titled 'Geographic Variation in Hunted Species,' but its relevance to the topic is unclear. For instance, it is not evident whether the authors intend to argue that this variation is culturally determined. While there is a map and data related to this issue, the article does not adequately explore them. This section relies on the literature, but its

connection to the dataset or study results is unclear. From what I understand, there is a spatial dataset indicating indigenous language groups (as shown in Extended Data Figure 3). However, it is not evident how this dataset was utilized or how it contributed to the findings. I would expect to see actual results regarding cultural diversity.

We agreed with the reviewer that this topic had not been well developed and thanked him for his suggestions. This was indeed a weak point in the first version of the manuscript, and we have developed it further in this new version. However, there are sensitive points that should be treated with caution. We don't want to propose deterministic cultural hunting patterns in the Amazon. This is much more complex than our data shows. Nevertheless, we have shown general patterns that do not prejudice certain social groups. We show how cultural factors are important in shaping hunting patterns in the Amazon, as they emerge as important predictors for HHR. Although it is very complex to capture and discuss the nuances of this, considering the enormous diversity of regions, habitats, zoogeographic provinces, cultures and degrees of conservation, we have endeavored to show patterns and differences wherever pertinent. We now provide much more information across the paper about how different cultures determine certain aspects of wild meat hunting, and also call for further studies that look into this issue more deeply:

New text (Lines 120-140):

Random forest spatial models predicted higher HP in forested regions of Western Amazonia, along the mainstream of the Amazon River and its major tributaries, as well as in parts of the Guyana Shield and Southern-Western Amazonia (Fig. 1a, 1b). These areas are characterised by fertile soils, high primary productivity, low to moderate elevations, well-preserved forests, and relative isolation from large urban centres. Higher HP was also predicted in historically inhabited territories managed by the Waorani, Itonama, Movima, Warina, Kandoshi-Shapra, and Yurucare Indigenous peoples of Western Amazonia (Extended Data Fig. 3). Full details on the relationships between HHR and environmental, cultural and threatening factors, including variable importance, are provided in the Extended Data Figure 5 and Supplementary Data 1.

Our predicted HP patterns align with broader Amazonian ecosystem dynamics, particularly the region's geological history and the resulting variation in soil fertility, forest productivity and turnover, and species and functional composition. Enriched by Andean erosion, the younger and more fertile soils of Western Amazonia and floodplains support highly productive forests that prioritise reproductive processes (e.g., flower and fruit production) over photosynthesis¹³. This reproductive investment likely sustains larger and more diverse animal populations^{14,15} (Extended Data Fig. 6), contributing to higher HP in these regions.

Cultural identity emerged as a key predictor of HP across Amazonia (see Extended Data Fig. 5; Supplementary Data 1). With 511 distinct Indigenous Peoples¹⁶ speaking at least 335 languages¹⁷, Amazonia's remarkable cultural diversity reflects millennia of dynamic and reciprocal relationships with nature^{3,4}. Our results reveal, for the first time on a large spatial scale, how environmental, cultural and threatening factors interact to shape wildlife harvesting patterns across the region.

New text (Lines 193-206):

Environmental context also shapes hunting patterns. In flooded forests, Amazonian Peoples hunt higher proportions of river turtles, tortoises, caimans, howler monkeys, capybara, manatees, and waterfowl (ducks, cormorants, and egrets). In contrast, those living in upland terra firme forests, which cover 86% of Amazonia²¹, yield more tinamous, trumpeters, wood quails, guans, and armadillos are more commonly hunted (Extended Data Tables 6, 7). Many Amazonian Peoples have historically settled in transitional zones between flooded and upland areas²², taking advantage of seasonally complementary resources and adapting management strategies accordingly²³.

*Cultural identity strongly influences HP and TSOP across taxa. For example, most Aruak Peoples value hunting ungulates, yet those in the Xingu River deliberately avoid them. In contrast, some Indigenous peoples in the northwestern Amazon highly value the often overlooked slender-legged treefrogs (*Osteocephalus* spp.). These patterns reflect a complex interplay of cultural preferences, environmental conditions, and ecological factors shaping hunting dynamics. Understanding these systems requires moving beyond environmental or cultural determinism toward an interdisciplinary perspective.*

New text (Lines 289-304):

We provide new insights into the complex interplay of environmental, cultural and threatening factors shaping wildlife harvest patterns throughout Amazonia. The finer details of these interactions merit further locally focused research and avoid determinisms. While the Sustainable Development Goals and IUCN Red List of Threatened Species frameworks emphasise global policies and actions, we stress that wildlife management initiatives must be tailored to local ecological, cultural, socioeconomic, and vulnerability nuances, ideally focused on local and regional key hunted taxa and led by Amazonian Peoples. Wildlife management initiatives shaped by Amazonian Peoples' demands and cultural practices are more legitimate and likely to remain viable long-term. Indigenous Lands and Extractive Reserves usually maintain healthy populations of key hunted species⁴¹, even those once commercially over-harvested⁴². Indigenous and traditional knowledges are critical for assessing the conservation status of animal populations⁴³ and understanding key ecological parameters⁴⁴, including their varying density⁴⁵, reproductive rates⁴⁶, and population dynamics²⁶. These factors, along with deforestation³⁴ and commercial hunting pressures⁴⁷, and climate change impacts³⁵, strongly influence sustainable harvest potentials. Our predictive heatmaps offer valuable spatial insights into harvest productivity, human-wildlife interactions, and human nutrition in Amazonia, supporting more effective policies integrating conservation and public health.

We concluded the article with the following:

New text (Lines 314-326):

Amazonian systems of knowledges and practices are grounded in ontologies attributing agency, personhood, and humanity to various beings, including animals⁴⁹, which must be taken seriously in ecological assessments of traditional hunting and management systems⁵⁰. For Amazonian Peoples, relationships with wildlife are not based on resource extraction and nutrient provisioning but rather on social reciprocity governed by norms and ethical obligations⁵¹. Dietary restrictions and spatial avoidance function as wildlife management and conservation systems akin to species-specific protection, spatial zoning rules, and hunting bans⁵². These cultural practices are effective tools in regulating wildlife harvesting and the most legitimate management and conservation strategies in Amazonia⁵³. Wild meat is a critical food source and a social cornerstone that motivates Amazonian Peoples to

safeguard their territories. The health of Amazonian ecosystems and wildlife is inextricably linked to the well-being of Amazonian Peoples, underscoring the importance of recognizing their land rights and supporting policies that enhance their autonomy and governance over their territories and biodiversity.

Page 9, Figure 4: unclear the source of the requirements used to calculate this.

Due to the limited space, all information about the sources of the requirement data is presented in the Methods section, as follows:

Text (Lines 763-799):

Percentage of dietary requirements furnished by wild meat

To estimate the nutritional needs of micronutrients of the Amazonian Peoples we used the dietary reference intakes (DRIs). The DRIs are a set of recommendations for nutrient intake based on the latest scientific evidence and intended to guide the amounts of nutrients that are needed to maintain health. Due to the lack of specific nutritional data for the targeted population, such as weight and food consumption, we had to rely on general references measured in grams per day (g/day) instead of grams per kilogram per day (g/kg/day) of a given nutrient. Consequently, we selected the Estimated Average Requirement (EAR) values, measured in weight per day, as our primary choice. This approach was taken because the EAR reflects the average daily nutrient intake estimated to meet the needs of half of the healthy individuals within a specific age and gender group^{88,89}. In cases where there was no EAR available, we used the Adequate Intake (AI) or Recommended Dietary Allowance (RDA) as a guide for determining the appropriate recommendation of a nutrient. For total fat, in instances where DRIs were unavailable, we utilized the Acceptable Macronutrient Distribution Range (AMDR). Considering the midpoint of the range, we converted these values to a percentage of energy. Finally, for energy (i.e., calories), we applied the Estimated Energy Requirements (EER)⁹⁰. The DRI values are presented in the Extended Data Tables 10 and 11.

First, we constructed a raster of the population size for each sex-age group (i.e., children, women, and men) by multiplying the AMPS raster with the proportion of each group relative to the total rural population in Amazonia. These proportions were derived from data available for North Brazil through the Brazilian population census⁹¹. Using this approach, we established spatially explicit demographic distributions reflecting rural Amazonia's population structure.

Then, we spatially predicted each sex-age group's minimum daily dietary requirements for seven micronutrients, two macronutrients, and energy. This was achieved by multiplying the average reference values for each nutrient or energy requirement with the corresponding sex-age population raster. The resulting Minimum Daily Dietary Requirements rasters for children, women, and men were then summed for each nutrient and energy to predict the overall Minimum Daily Dietary Requirements for Amazonian Peoples across all spatial cells. As a result, these requirements were directly proportional to the number of rural inhabitants per pixel, ensuring that the analysis accurately reflected localized nutritional needs across the region.

Finally, the daily nutrient amounts furnished by the estimated wild meat production in Amazonia were evaluated against the daily micro- and macro-nutrient requirements to determine their adequacy for a nutritionally balanced diet for Amazonian Peoples. This was done by comparing the corresponding spatial cell of a given nutrient/energy between the Daily Amounts Furnished by Wild Meat and the Minimum Daily Dietary Requirements of Amazonian Peoples. This process allowed us to spatially explicit the levels of energy and nutrients supplied by wild meat to Amazonian Peoples.

Page 9, Figure 4: This map shows the exact shape of certain indigenous territories in the Eastern Amazon. I am curious about its relevance to the results. This is just one example of spatial differences that the article's findings do not explore.

In the first version of the article, we didn't use current Indigenous and traditional territories as predictors; rather, we used historical Indigenous territories. In this new version, we have used the distribution of historical Indigenous territories, the current distribution of Indigenous and traditional territories, classified by language family, and the binary distribution of Indigenous and traditional territories. Now these patterns are likely to become even clearer, especially given the importance of some cultural factors (mainly the current distribution of indigenous and traditional territories). However, we suppose that the reviewer's careful observation of the patterns on the map in Figure 4 may be related to the conservation patterns of the Amazon rainforest in this region since it is precisely within the indigenous territories and other protected areas that the forest is conserved, while outside, the forest has been cleared. Is that it? Or, perhaps the reviewer was referring to the coloured pixels, which represented urban areas - as the figures were not of the best quality, we understand that this could generate some misunderstanding.

Page 10, Line 248-269: How do the results compare to previous studies and other regions worldwide? If the authors could summarize these references in relation to the Sustainable Development Goals, it would allow for a more direct comparison of their findings with existing literature—something that is currently lacking.

We included some brief information about findings from previous studies and other regions, as follows:

New text (Lines 230-239):

This level of available wild meat per rural inhabitant has the potential to meet the Dietary Reference Intakes (DRIs) for vitamin B12 across much of Amazonia, while also significantly contributing to DRIs for protein, iron, zinc, and other essential B vitamins and minerals (Fig. 4). The irreplaceable nutritional value of wild meat is particularly critical for 10.89 million rural inhabitants in Amazonia, providing a highly bioavailable source of protein, all essential amino acids, and vital micronutrients, often less accessible in plant-based foods^{27,28}. Higher wild meat availability was linked to better health among children not only in the Amazon²⁹ but also in the Congo Basin³⁰, including higher hemoglobin levels in children and increased household iron and zinc intake³¹. This is particularly noteworthy in regions where micronutrient deficiencies are widespread, compounded by malaria, intestinal parasites, and genetic disorders³².

We also provided a new figure (Extended Data Figure 14) summarising the contribution of traditional wild meat food systems in advancing the Sustainable Development Goals (SDGs).

Page 10, Line 262: SDGs instead of Sustainable Development Goals?

Changed

Page 10, Lines 267-269: I believe that discussing these regional differences is more compelling and important than referencing the various Sustainable Development Goals.

We agree that we should discuss regional differences further. We have now included several paragraphs addressing this as already described in the sections: “*More discussion on the limitations of model-based predictions is necessary, especially in heterogenous environments like the Amazon*” and “*Spatial Resolution: Consider discussing the implications of the 10×10 km spatial resolution for local-scale conservation efforts*” of this reviewer section.

Page 11, Line 285: Are you referring to "biodiversity" or "diversity" indicator"? Additionally, this phrase is quite convoluted.

This original sentence was removed after the revision process.

Page 11, Lines 298-299: How does this translate into the territorial percentage of additional deforestation, for example?

New text (Lines 267-275):

Due to its high cost and the significant logistical challenges of producing, transporting, and storing in remote areas, beef is rarely consumed across most of Amazonia³⁷. Paradoxically, cattle ranching remains the leading driver of deforestation in the region, contributing to the loss of approximately 0.63M km² of forest since 1978, primarily to supply domestic meat markets³⁹. Replacing the estimated edible wild meat production of 0.37 M t with beef, based on current cattle ranching yields in Amazonian traditional pastures (0.2–0.8 kg ha⁻¹ day⁻¹)⁴⁰, would require converting 7,602–63,798 km² of forest into pasture. This conversion would emit 0.3×10⁹ – 3.5×10⁹ tonnes of CO₂, equivalent to up to 10% of global annual emissions. Despite this significant environmental cost, domestic meat production still fails to ensure equitable access for rural peoples in remote areas.

Page 11, Line 303: Instead of "parks," it would probably be better to use "protected areas, including extractive reserves."

Done.

Page 11, Line 312: Is there a comma missing after “densities”?

Done.

Page 11, Line 314-315: This point sounds important, but it would be more impactful and helpful if you could specify how it contributes.

314 harvesting. Our taxon-specific extraction maps can guide policies on effective wildlife management, 315 food health and pathogen surveillance.

New text (Lines 302-304):

Our predictive heatmaps offer valuable spatial insights into harvest productivity, human-wildlife interactions, and human nutrition in Amazonia, supporting more effective policies integrating conservation and public health.

Page 11, Line 316-329: How does this relate to the findings? It might be beneficial for the authors to concentrate more on exploring regional differences, for instance, as this could provide more insight than including statements that seem unrelated to the findings or lack specific evidence.

316 Indigenous groups have inhabited Amazonia for thousands of years^{3,4}, establishing complex 317 relationships with nature that influence wildlife use and shape economic, social, and spiritual practices 318 41,51,52. This study underscores the importance of recognizing the crucial role of traditional hunting and 319 wild meat access for the health, nutritional security, biocultural heritage, and food systems of 320 Amazonian Peoples, and thus for their cultural self-determination. Consideration of these aspects are 321 essential considering the ambitious 2030 targets of the Sustainable Development Goals, which aim to 322 ensure sustainable food systems, improve health outcomes, and protect terrestrial ecosystems. 323 Therefore, to achieve the desired conservation and nutritional security targets, it is crucial to promote 324 community-led wildlife management initiatives that are aligned with cultural frameworks in Amazonia 325 and resilient to political fluctuations⁵³. The health of Amazonian Peoples is intricately linked to their 326 forests, highlighting that forest degradation negatively impacts both people and nature. To safeguard 327 Amazonia’s forest health and sustain wildlife hunting, it is imperative to fully recognize the rights of 328 Amazonian Peoples to their lands, establish clear boundaries, and implement policies focusing on their 329 autonomy and governance.

We agree with the reviewer. About this regional variation, we added these:

New text (Lines 120-140):

Random forest spatial models predicted higher HP in forested regions of Western Amazonia, along the mainstream of the Amazon River and its major tributaries, as well as in parts of the Guyana Shield and Southern-Western Amazonia (Fig. 1a, 1b). These areas are characterised by fertile soils, high primary productivity, low to moderate elevations, well-preserved forests, and relative isolation from large urban centres. Higher HP was also predicted in historically inhabited territories managed by the Waorani, Itonama, Movima, Warina, Kandoshi-Shapra, and Yurucare Indigenous peoples of Western Amazonia (Extended Data Fig. 3). Full details on the relationships between HHR and environmental, cultural and threatening factors, including variable importance, are provided in the Extended Data Figure 5 and Supplementary Data 1.

Our predicted HP patterns align with broader Amazonian ecosystem dynamics, particularly the region's geological history and the resulting variation in soil fertility, forest productivity and turnover, and species and functional composition. Enriched by Andean erosion, the younger and more fertile soils of Western Amazonia and floodplains support highly productive forests that prioritise reproductive processes (e.g., flower and fruit production) over photosynthesis¹³. This reproductive investment likely sustains larger and more diverse animal populations^{14,15} (Extended Data Fig. 6), contributing to higher HP in these regions.

Cultural identity emerged as a key predictor of HP across Amazonia (see Extended Data Fig. 5; Supplementary Data 1). With 511 distinct Indigenous Peoples¹⁶ speaking at least 335 languages¹⁷, Amazonia's remarkable cultural diversity reflects millennia of dynamic and reciprocal relationships with nature^{3,4}. Our results reveal, for the first time on a large spatial scale, how environmental, cultural and threatening factors interact to shape wildlife harvesting patterns across the region.

New text (Lines 193-206):

Environmental context also shapes hunting patterns. In flooded forests, Amazonian Peoples hunt higher proportions of river turtles, tortoises, caimans, howler monkeys, capybara, manatees, and waterfowl (ducks, cormorants, and egrets). In contrast, those living in upland terra firme forests, which cover 86% of Amazonia²¹, yield more tinamous, trumpeters, wood quails, guans, and armadillos are more commonly hunted (Extended Data Tables 6, 7). Many Amazonian Peoples have historically settled in transitional zones between flooded and upland areas²², taking advantage of seasonally complementary resources and adapting management strategies accordingly²³.

*Cultural identity strongly influences HP and TSOP across taxa. For example, most Aruak Peoples value hunting ungulates, yet those in the Xingu River deliberately avoid them. In contrast, some Indigenous peoples in the northwestern Amazon highly value the often overlooked slender-legged treefrogs (*Osteocephalus* spp.). These patterns reflect a complex interplay of cultural preferences, environmental conditions, and ecological factors shaping hunting dynamics. Understanding these systems requires moving beyond environmental or cultural determinism toward an interdisciplinary perspective.*

New text (Lines 289-304):

We provide new insights into the complex interplay of environmental, cultural and threatening factors shaping wildlife harvest patterns throughout Amazonia. The finer details of these interactions merit further locally focused research and avoid determinisms. While the Sustainable Development Goals and IUCN Red List of Threatened Species

frameworks emphasise global policies and actions, we stress that wildlife management initiatives must be tailored to local ecological, cultural, socioeconomic, and vulnerability nuances, ideally focused on local and regional key hunted taxa and led by Amazonian Peoples. Wildlife management initiatives shaped by Amazonian Peoples' demands and cultural practices are more legitimate and likely to remain viable long-term. Indigenous Lands and Extractive Reserves usually maintain healthy populations of key hunted species⁴¹, even those once commercially over-harvested⁴². Indigenous and traditional knowledges are critical for assessing the conservation status of animal populations⁴³ and understanding key ecological parameters⁴⁴, including their varying density⁴⁵, reproductive rates⁴⁶, and population dynamics²⁶. These factors, along with deforestation³⁴ and commercial hunting pressures⁴⁷, and climate change impacts³⁵, strongly influence sustainable harvest potentials. Our predictive heatmaps offer valuable spatial insights into harvest productivity, human-wildlife interactions, and human nutrition in Amazonia, supporting more effective policies integrating conservation and public health.

We concluded the article with the following:

New text (Lines 314-326):

Amazonian systems of knowledges and practices are grounded in ontologies attributing agency, personhood, and humanity to various beings, including animals⁴⁹, which must be taken seriously in ecological assessments of traditional hunting and management systems⁵⁰. For Amazonian Peoples, relationships with wildlife are not based on resource extraction and nutrient provisioning but rather on social reciprocity governed by norms and ethical obligations⁵¹. Dietary restrictions and spatial avoidance function as wildlife management and conservation systems akin to species-specific protection, spatial zoning rules, and hunting bans⁵². These cultural practices are effective tools in regulating wildlife harvesting and the most legitimate management and conservation strategies in Amazonia⁵³. Wild meat is a critical food source and a social cornerstone that motivates Amazonian Peoples to safeguard their territories. The health of Amazonian ecosystems and wildlife is inextricably linked to the well-being of Amazonian Peoples, underscoring the importance of recognizing their land rights and supporting policies that enhance their autonomy and governance over their territories and biodiversity.

Page 14, Line 431: Why is the term "traditional hunting dataset" used? Typically, "traditional" refers to specific small-scale societies with a long history in the same location and distinct cultural practices. Since the data also include agriculturalists, who are often more recent inhabitants, this term seems misleading. Also, earlier in the article, the authors refer to "Amazonian peoples" and not traditional people (see line 88). Additionally, they should clarify whether they are specifically referring to rural small-scale communities and explain which types of rural communities are excluded from/included in the sample.

We now use the name:

Amazonian Peoples Traditional Hunting Dataset (APTH Dataset)

Page 14, Line 436: Which is Table X or where is it? Moreover, there are 199 articles (please check the mistake in references 66 and 67, as they appear to be the same).

Corrected.

Page 14, Line 446: Could you briefly explain why data on wild meat trade and urban consumption were excluded?

Data on wild meat trade and urban consumption were excluded to maintain the study's focus on rural hunting practices and their ecological and nutritional implications. Including trade and urban consumption would have introduced complexities related to market dynamics, transportation, and intermediate processing, which fall outside the scope of this study's spatial and subsistence-focused approach. Additionally, reliable data on trade and urban consumption are often sparse or inconsistent, making it challenging to integrate them into the models without introducing significant uncertainty. By concentrating on rural contexts, the study ensures a clearer understanding of hunting's direct ecological and nutritional impacts, providing a solid foundation for further research on trade and urban dynamics.

New text (Lines 615-623):

Wild meat trade and urban consumption data were excluded to maintain a focus on rural hunting practices and their ecological and nutritional implications. Including trade and urban consumption would have introduced complexities related to market dynamics, transportation, and intermediate processing, which are outside the scope of this study's subsistence-focused approach. Additionally, reliable trade and urban consumption data are often sparse or inconsistent in Amazonia, making integrating them into the models challenging without introducing significant uncertainty. Although the spatial process of excluding urban areas probably excludes rural or peri-urban inhabitants who may rely on wild meat for part of their diet, our database includes such people, analyzing the effects of urbanization on Harvest Productivity more reliably.

New text (Lines 84-92):

This paper examines the links between wildlife, human nutrition, and sustainable food systems in Amazonia. Through these analyses, we assess key indicators of ecosystem integrity—including species diversity, biomass availability, and habitat condition. Our findings show that deforestation and urbanisation reduce wild meat productivity and simplify the composition of hunted taxa, ultimately threatening traditional food systems and the nutritional well-being of Amazonian Peoples. Protecting Amazonia is therefore essential not only for conserving biodiversity but also for sustaining ecosystem functions and achieving multiple global SDGs, including those related to environmental and human health.

New text (Lines 262-266):

We also predicted higher offtake and lower HP near urban centres, raising concerns about the sustainability of hunting in these areas. High meat demand in densely populated peri-urban areas, coupled with declining wildlife populations³⁶, may shift rural diets toward

cheaper domestic meats like chicken³⁷, which generally contains four times less iron, twice less B vitamins, and lower levels of protein and zinc than wild meat³⁸.

However, we discuss the effect of urbanization on the Harvest Productivity and availability of wild meat per rural inhabitant in several parts of the text, for example:

New text (Lines 99-106):

This paper examines the links between wildlife, human nutrition, and sustainable food systems in Amazonia. Through these analyses, we assess key indicators of ecosystem integrity—including species diversity, biomass availability, and habitat condition. Our findings show that deforestation and urbanisation reduce wild meat productivity and simplify the composition of hunted taxa, ultimately threatening traditional food systems and the nutritional well-being of Amazonian Peoples. Protecting Amazonia is therefore essential not only for conserving biodiversity but also for sustaining ecosystem functions and achieving multiple global SDGs, including those related to environmental and human health.

Page 15, Line 455: It is probably better to refer to it as face-to-face interviews or systematic interviews. Some authors use the term "questionnaire" exclusively for self-administered protocols.

Changed.

Page 15, Line 449-450: How do you ensure accuracy with data dating back to 1965? Additionally, how do you guarantee that the data are both culturally appropriate and scientifically rigorous, particularly when using secondary sources? It would be helpful to clearly distinguish between the primary and secondary data, as combining them without explanation may lead to confusion. Additionally, you might consider avoiding adjectives unless you provide an explanation for them.

New text (Lines 521-544):

Sources, selection, and validation of primary and secondary data

Given the diversity of secondary data sources spanning nearly six decades, our dataset naturally varied in objectives, methodological rigour, reporting standards, and cultural contexts. To ensure consistency, we focused on extracting comparable information across all studies.

We implemented a multi-step validation process to mitigate potential inaccuracies. Primary data served as a baseline for evaluating datasets and assessing methodological consistency, identifying discrepancies, and evaluating the reliability of secondary sources. We prioritised studies that provided clear methodological descriptions, reproducible metrics, or supplemental documentation and contacted the original authors or institutions when clarification was needed regarding sampling design and local conditions.

We scrutinized outliers and apparent inconsistencies by cross-referencing with more recent peer-reviewed sources or primary datasets from similar regions or time periods. When discrepancies were identified, we assessed whether they reflected genuine cultural differences, shifting hunting practices, or methodological shortcomings. Records showing implausible biological or cultural values were excluded.

Throughout the validation process, we remained attentive to cultural and practical factors – such as hunting laws, local traditions, and wildlife management strategies – which vary widely across regions and over time. To address this, we consulted local experts and researchers familiar with such nuances to confirm that data collection methods were culturally appropriate and to verify that the underlying assumptions of each dataset remained valid. Only datasets meeting our standards for scientific rigour and comparability were integrated into the analysis and annotated with metadata. We also excluded studies that reported less than four hunted taxa, as these offered limited insights into species composition. Records lacking clearly described or reliable hunting effort methodologies were removed from the final dataset.

Page 15, Line 462: Could you clarify if this refers only to primary data? If not, how was consistency ensured? Were the same protocols applied to all data, and were trained surveyors involved? It would be helpful to provide more detail on the methods used, as simply stating that consistency and accuracy were ensured may not be sufficient.

462 different communities and time periods. These methods ensured that data collection was both respectful

463 of the communities' ways of life and robust enough to provide reliable data for the ATH dataset.

For primary data. We added this detail to the section name.

New text (Line 504):

Hunting monitoring schemes and data collection methods for primary data

Page 16, Lines 515-546: In some instances, the titles in Extended Data Figures 2 and 3 do not match exactly those in the main text of the article.

For example: (i) "Annual Net Primary Productivity" – the word "Annual" is missing in the figure; (ii) "Proportion of Flooded Forest" – the figure shows "Proportion of Flooded Area" (please verify all discrepancies). Additionally, the order of appearance is inconsistent between documents.

We thank the review for this observation. We corrected them.

Page 17, Lines 539-540. The explanation in the methods section is clear; however, as I read through the article, it is not entirely clear how the data on

indigenous language zones was applied, as the relevant results are difficult to identify. It would be helpful if the authors provided more detailed information on the variations in human populations associated with hunting. In several sections on this topic, the information seems to rely more on literature than on the authors' own data.

Please see our responses to the questions above for better emphasis on the role of cultural factors in shaping hunting in Amazonia.

Page 17, Lines 557-562: I understand that you aimed to minimize the issue of temporal variation. However, I believe it would be important to highlight how this limitation may have impacted your results. While this is perhaps the most apparent limitation, there is no information provided on how it may have introduced bias into the findings.

We thank the reviewer for pointing out this issue and we agree that the analysis should consider temporal variation in anthropogenic and environmental factors for each data point. This was an issue that concerned us from the start of the analysis.

In the first version we had dealt with it using the average or median of the variables for the whole period: *“Whenever a spatial variable had multiple temporal rasters, we built one with the average or median values. For instance, we obtained a single raster for both Annual NPP and GPP from the median of 22 years. In relation to the Proportion of Forest Cover variable, we used the annual mapping of 2005 land cover, since 2005 represents the averaged year of our ATH dataset. These techniques were aimed at minimising errors caused by temporal variation in spatial variables during the significant time window covered by our ATH dataset.”*

However, in this new version, we strived to obtain more time-specific information for all variables. Given the lack of data for some of them, and the high computational cost, we re-run the model using multiple averages for various specific periods considering the closest data available for each covariate in relation to each hunting study. We highly appreciate this suggestion as our models were greatly improved. We now added a new section in the Methods:

New text (Lines 638-652):

Temporal variation of spatial variables

None of the digital spatial variables fully cover the 1965–2024 period of the APTH dataset, preventing a year-by-year evaluation of their spatial and temporal effects on HHR and TSPO. To address this, we matched HHR and TSPO records to the closest available values of spatial variables in years where spatio-temporal data were available. This approach was feasible for modeling the effects of EVI, NPP, and GPP on HHR and TSPO by incorporating measurements at five-year intervals. We obtained Proportion of Habitat Loss data for 1985, 1990, 1995, 2000, 2005, 2010, 2015, and 2020. For EVI, NPP, and GPP, data were available from 2000 onwards at the same five-year intervals. HHR and TSPO values from 1965 to 1987 were assigned the 1985 habitat loss data; from 1988 to 1992, the 1990 data; from 1993 to 1998, the 1995 data, and so on, with post-2018 records assigned the 2020 data. A similar process was applied to EVI, NPP, and GPP, though for a shorter

period: records from 1965 to 2002 were matched to the 2000 values, continuing at five-year intervals up to 2020. Despite the APTH dataset spanning 1965–2024, most hunting studies occurred around 2006±9 years (90% quantiles: 1995–2017), aligning well with the available temporal coverage of the spatial variables.

We also included cautious remarks in the main text regarding the lack of data for some variables:

New text (Lines 141-149):

While our findings provide robust insights, several limitations should be considered when interpreting these results. Although hunting data span from 1965 to 2024, spatial covariates are not uniformly available across the entire period. To partially address this, we matched HHR and TSOP records to the closest corresponding time period (see Methods). We also incorporated a variable to account for variation in recording effort across hunting studies (see Methods), which proved to be an important predictor in our HHR models. Despite potential limitations in capturing fine-scale local variability, the breadth and diversity of our dataset, combined with consistent spatiotemporal covariates, provide a strong foundation for a comprehensive assessment of wildlife harvesting dynamics throughout Amazonia.

and...

New text (Lines 207-213):

While our models adequately capture TSOP variation for key hunted taxa, they are less reliable for underrepresented species (Supplementary Data 2). In addition, hunting practices have evolved over the past 60 years, influenced by cultural and technological shifts. Increased rifle use, for example, has increased hunting efficiency, particularly for large species²⁴, while the growing use of flashlights has intensified the hunting pressure on nocturnal animals such as the paca²⁵, now the most hunted species in Amazonia. Our models also do not account for recently documented natural population cycles in white-lipped peccaries²⁶, which may influence hunting patterns over time.

Page 19, Line 642: Do you mean the average per species or the average across all species? This distinction becomes clear only later in the article.

For all species. However, by considering the amount produced and percentage of edible wild meat separately for each of the 20 dominant hunted taxa and then pooled by major taxonomic groups: mammals, birds, chelonians, and caimans, as stated:

New text (Lines 750-759):

Proportion of edible wild meat to undressed biomass

We estimated the overall annual production of edible wild meat in Amazonia (see Metrics Estimation in the Methods section for details) using data on the proportion of consumable meat contained in animal carcasses reported in the literature^{73,74}. Edible yield proportions were calculated separately for each of the 20 dominant hunted taxa and then pooled by major taxonomic groups: mammals (0.63 ± 0.11), birds (0.73 ± 0.06), chelonians (0.47 ± 0.14), and caimans (0.45 ± 0.04). These values were then multiplied by the estimated

total undressed animal biomass offtake for the corresponding taxa or groups. Based on this approach, we estimated that edible wild meat represents approximately 58.7% of the total undressed biomass harvested annually across Amazonia.

Please see Extended Data Table 11.

Page 19, Line 644: A comma is missing after “Protein”.

Done.

Page 20, Line 674: The text lacks a smooth transition between the two paragraphs and could benefit from a connecting phrase to create a more cohesive flow.

New text (Lines 780-799):

First, we constructed a raster of the population size for each sex-age group (i.e., children, women, and men) by multiplying the AMPS raster with the proportion of each group relative to the total rural population in Amazonia. These proportions were derived from data available for North Brazil through the Brazilian population census⁹¹. Using this approach, we established spatially explicit demographic distributions reflecting rural Amazonia's population structure.

Then, we spatially predicted each sex-age group's minimum daily dietary requirements for seven micronutrients, two macronutrients, and energy. This was achieved by multiplying the average reference values for each nutrient or energy requirement with the corresponding sex-age population raster. The resulting Minimum Daily Dietary Requirements rasters for children, women, and men were then summed for each nutrient and energy to predict the overall Minimum Daily Dietary Requirements for Amazonian Peoples across all spatial cells. As a result, these requirements were directly proportional to the number of rural inhabitants per pixel, ensuring that the analysis accurately reflected localized nutritional needs across the region.

Finally, the daily nutrient amounts furnished by the estimated wild meat production in Amazonia were evaluated against the daily micro- and macro-nutrient requirements to determine their adequacy for a nutritionally balanced diet for Amazonian Peoples. This was done by comparing the corresponding spatial cell of a given nutrient/energy between the Daily Amounts Furnished by Wild Meat and the Minimum Daily Dietary Requirements of Amazonian Peoples. This process allowed us to spatially explicit the levels of energy and nutrients supplied by wild meat to Amazonian Peoples.

COMMENTS TO OTHER DOCUMENTS

LIST OF LITERATURE CONTAINING HUNTING DATA IN RURAL.

Check references 66 and 67: there is only one reference.

We thank the reviewer. Corrected.

Correct reference 100: instead of Luz C. the correct is Luz AC. (sorry, I know her... ;)

Corrected.

EXTENDED DATA - ALL: I found it difficult to read the text on most maps with the naked eye, for instance in Figures 5, 6, 7, and 9. Could you increase the caption size?

We hope we've improved the figures in this new version.

EXTENDED DATA FIGURE 3: It is unclear what purpose this was intended to serve or how it fits into the analysis.

With new text and analysis highlighting the importance of cultural factors, it should be more comprehensible.

EXTENDED DATA- TABLE 1: The table (first column) is not in alphabetical order. Could you please check that?

It is ordered decreasingly as the number of animals hunted per year.

Extended Data Table 9: In Table caption is written: "Estimation was based on 256 observation" . There is an S missing in "observation".

Corrected. Thank you.

EXTENDED DATA TABLE 11: The title it "Daily values of Estimated Energy Requirements "for energy". Is this correct?

Yes. Energy refers to Calories (Kcal).

REFERENCES

The previous literature is properly referenced.

We thank the reviewer.

CLARITY AND CONTEXT

The abstract is clear, well-written, and provides an appropriate summary of the article. However, if possible, the authors might consider emphasizing the importance of understanding urban wild meat consumption as well.

We addressed this in our responses above. We thank the reviewer for all the insights, which greatly improved our work.

Referee #5

General Comments

The Amazonia is by far one of the most unique tropical forest regions on Earth, harboring high species and rich indigenous cultures. For thousands of years, documented interactions revealed that rural indigenous people have been known to depend on meat from animals in the forests as a key source of food, and most have done so in a sustainable manner. Yet, this relationship, in the form of traditional food systems, may be disrupted in recent times due to globalization, urbanization, and climate change impacts. In particular, deforestation may affect the supply and availability of wild meat to the rural people who are dependent on them for essential nutrients.

The authors are right to call for attention to human and forest ecosystem health, and hunger issues (all key Sustainable Development Goals) all highly relevant to the Amazonian peoples. Drawing on an impressive dataset collected over 40 years making up of over 400,000 individual animals harvested across at least 500 rural localities contributed from an extensive collaboration network of researchers and managers, the authors have estimated, through biological and spatially explicit modeling, an annual extraction of 0.33 Mt of edible wild meat, largely from 20 hyperdominant taxa. While this amount of wild meat is sufficient to meet some of the key nutritional requirements (both macro and micro) for rural people, high extent of deforestation in Amazonia may lead to a 60% decrease in wild meat productivity, jeopardizing both the wildlife and the traditional food systems known to preserve the Amazonian way of life for millennia.

Yet, this potentially ground-breaking and profound piece of work (and it is well-written!) hinged on the assumptions that the underlying data and models, for which the key results were estimated, are reasonable. I am outlining some key issues below that I would like the authors to address so the results could be more convincing, but it might take some effort to sort them out.

Being a large-scale spatially explicit assessment, there is a need to reasonably estimate the wild animal harvest rates and offtake levels. With this forming the basis of how wild meat contributes to food security, it becomes extremely crucial to get the initial modeling right. My biggest issue is with how the metrics (TSOP and HHR) were estimated. The authors used Random Forest models to spatially predict the metrics with twelve environmental and anthropogenic variables but these variables may be independent from the time the surveys were undertaken. The forest cover data is a good case in point. Say for example that if harvested data from the surveys stretched over two decades, and studies have shown that Amazonian forest cover has been changing rapidly. How is it possible that the harvested animal rate is estimated without accounting for the temporal changes to the numerous environmental and anthropogenic variables?

The above issue is also directly related to the key finding that higher wild meat offtake is predicted mostly around major Amazonian cities. All other spatially and demographically related findings are conditional on the assumptions made for the spatial models. While the authors acknowledge the limitations inherent to the study, particularly regarding the spatial coverage of their data (lines 147-148), they also need to consider the temporal coverage of their digital spatial variables and models. Can the authors address these issues (including having sensitivity analysis and simulations) to improve their models and outcomes? If not, the authors would need to provide an acceptable argument to defend their current approach. In my opinion, this is the most defining issue for this manuscript that would need to be resolved.

We thank the reviewer for pointing out this issue and agree that the analysis should consider temporal variation in anthropogenic and environmental factors for each hunting data point. This has been a concern since the beginning of the study, more precisely since we started the spatial analyses. Unfortunately, no spatial covariates used in the models cover the entire time span. However, we re-run the model using averages within a specific period and the closest data available for each covariate in relation to each hunting study. We highly appreciate this suggestion as our models were greatly improved.

In the first version, we dealt with it using the average or median of the variables for the whole period: *“Whenever a spatial variable had multiple temporal rasters, we built one with the average or median values. For instance, we obtained a single raster for both Annual NPP and GPP from the median of 22 years. In relation to the Proportion of Forest Cover variable, we used the annual mapping of 2005 land cover, since 2005 represents the averaged year of our ATH dataset. These techniques were aimed at minimising errors caused by temporal variation in spatial variables during the significant time window covered by our ATH dataset.”*

However, we strived to obtain more time-specific information for all variables in this new version. Given the lack of data for some of them and the high computational cost, we re-run the model using multiple averages for various specific periods considering the closest data available for each covariate in relation to each hunting study. We highly appreciate this suggestion as our models were greatly improved. We now added a new section in the Methods:

New text (Lines 638-652):

Temporal variation of spatial variables

None of the digital spatial variables fully cover the 1965–2024 period of the APTH dataset, preventing a year-by-year evaluation of their spatial and temporal effects on HHR and TSPO. To address this, we matched HHR and TSPO records to the closest available values of spatial variables in years where spatio-temporal data were available. This approach was feasible for modeling the effects of EVI, NPP, and GPP on HHR and TSPO by incorporating measurements at five-year intervals. We obtained Proportion of Habitat Loss data for 1985, 1990, 1995, 2000, 2005, 2010, 2015, and 2020. For EVI, NPP, and GPP, data were available from 2000 onwards at the same five-year intervals. HHR and TSPO values from 1965 to 1987 were assigned the 1985 habitat loss data; from 1988 to 1992, the 1990 data; from 1993 to 1998, the 1995 data, and so on, with post-2018 records assigned the 2020 data. A similar process was applied to EVI, NPP, and GPP, though for a shorter

period: records from 1965 to 2002 were matched to the 2000 values, continuing at five-year intervals up to 2020. Despite the APTH dataset spanning 1965–2024, most hunting studies occurred around 2006±9 years (90% quantiles: 1995–2017), aligning well with the available temporal coverage of the spatial variables.

We also included cautious remarks in the main text regarding the lack of data for some variables:

New text (Lines 141-149):

While our findings provide robust insights, several limitations should be considered when interpreting these results. Although hunting data span from 1965 to 2024, spatial covariates are not uniformly available across the entire period. To partially address this, we matched HHR and TSPO records to the closest corresponding time period (see Methods). We also incorporated a variable to account for variation in recording effort across hunting studies (see Methods), which proved to be an important predictor in our HHR models. Despite potential limitations in capturing fine-scale local variability, the breadth and diversity of our dataset, combined with consistent spatiotemporal covariates, provide a strong foundation for a comprehensive assessment of wildlife harvesting dynamics throughout Amazonia.

and...

New text (Lines 207-213):

While our models adequately capture TSOP variation for key hunted taxa, they are less reliable for underrepresented species (Supplementary Data 2). In addition, hunting practices have evolved over the past 60 years, influenced by cultural and technological shifts. Increased rifle use, for example, has increased hunting efficiency, particularly for large species²⁴, while the growing use of flashlights has intensified the hunting pressure on nocturnal animals such as the paca²⁵, now the most hunted species in Amazonia. Our models also do not account for recently documented natural population cycles in white-lipped peccaries²⁶, which may influence hunting patterns over time.

What is so interesting about this region is the presence of diverse indigenous cultures and their hunting practices, and target and preferred species. The authors did a good job in highlighting how the Amazonian Peoples have a deep understanding of the Amazonia ecological landscape.

But how deforestation may pit tribes (and cultures) against each other in order meet food security, which may indirectly increase conflicts among neighboring areas/tribes in the region? What are the risks of such conflicts under different scenarios?

We acknowledge that the reviewer is focusing his comment on Indigenous Peoples due to the use of the term “tribes,” so our comment will address our understanding of this specific parcel of hunting communities.

Deforestation and climate change certainly affect Indigenous communities’ food security; however, they are not only related to the depletion of hunted species. Communities’

responses to natural resources depletion, including those used for food, are complex and may vary according to several factors. The accessibility to alternative food sources, either from their forests, rivers, or land, or industrialized coming from outside communities, the social relations with neighbouring communities with available resources, family income leading to purchase power, and aid from external agencies are some of these factors that may influence how an Indigenous family or community respond to the impacts of habitat loss. Conflicts between communities over hunting grounds and prey species may happen- it is a risk. However, it is not the only possible outcome. Families may rely more on more resilient hunted species - even if less preferred, or other protein sources like fish or livestock, trade with other communities or buy from markets, or even negotiate access to hunting grounds of neighbouring communities. Although these outcomes affect Indigenous sociocultural aspects, they would represent conflicts between communities.

It is plausible that the diminishing supply of traditional food may cause shifts in diet which may increase competition and conflicts for shared species (even hyperdominance ones), hence leading to unsustainable offtakes and food system compared to the past. As stated in lines 222-224, the “dynamic interplay has led Amazonian Peoples to develop diverse and sophisticated systems of hunting rules, species preferences, and taboos, resulting in varied hunted assemblages across hundreds of cultures”, but this may be potentially disrupted by deforestation and climate change issues with time.

If hunting patterns and diets change, led by a reduction in the availability of preferred species—as predicted by the application of the Optimum Foraging Theory to Amazonian rural hunters—as a consequence of deforestation or climate change, preferred species may face increased hunting pressure if there are also more people foraging them. Also, facing new scenarios of natural resource availability and social dynamics, agreements, norms, rules, and whole systems may change to adapt with time.

While I understand that modeling under different deforestation and climate change scenario may be outside of the scope of the manuscript, some key discussion, at the minimum, on such potential conflicts may be warranted.

Although we acknowledge that climate change can have severe impacts on wildlife and the human populations that rely on it, a comprehensive discussion of this topic falls outside the scope of the present article. For this reason, we limit our treatment of the issue to brief mentions when relevant to our findings.

New text (Lines 254-261):

Large-scale agriculture, land grabbing, logging, mining, and infrastructure development have led to deforestation³³, increasingly undermining Amazonian People’s reliance on biodiversity. The combined effects of deforestation and wildlife overharvesting have produced simplified animal assemblages, with lower species richness and fewer large-bodied species³⁴. These pressures are further compounded by recent climatic changes, including more frequent floods, droughts, and large-scale fires³³, all of which threaten both habitats and many key hunted taxa³⁵. We recommend that further studies should investigate

the future impacts of deforestation, wildfire, and climate change on animal populations and on the supply of wild meat.

New text (Lines 289-304):

We provide new insights into the complex interplay of environmental, cultural and threatening factors shaping wildlife harvest patterns throughout Amazonia. The finer details of these interactions merit further locally focused research and avoid determinisms. While the Sustainable Development Goals and IUCN Red List of Threatened Species frameworks emphasise global policies and actions, we stress that wildlife management initiatives must be tailored to local ecological, cultural, socioeconomic, and vulnerability nuances, ideally focused on local and regional key hunted taxa and led by Amazonian Peoples. Wildlife management initiatives shaped by Amazonian Peoples' demands and cultural practices are more legitimate and likely to remain viable long-term. Indigenous Lands and Extractive Reserves usually maintain healthy populations of key hunted species⁴¹, even those once commercially over-harvested⁴². Indigenous and traditional knowledges are critical for assessing the conservation status of animal populations⁴³ and understanding key ecological parameters⁴⁴, including their varying density⁴⁵, reproductive rates⁴⁶, and population dynamics²⁶. These factors, along with deforestation³⁴ and commercial hunting pressures⁴⁷, and climate change impacts³⁵, strongly influence sustainable harvest potentials. Our predictive heatmaps offer valuable spatial insights into harvest productivity, human-wildlife interactions, and human nutrition in Amazonia, supporting more effective policies integrating conservation and public health.

Specific Comments

Line 126: Is HP the same as wild meat offtake?

New text (Lines 106-114):

In this context, we define Harvest Productivity (HP) as the potential number of individuals and biomass that can be harvested per hunter across Amazonia, based on how overall HHR for all taxa pooled varies with environmental, cultural and threatening factors (Fig. 1b, 1c). The total number of individual animals (Individuals Offtake) and biomass (Biomass Offtake) harvested for all taxa pooled (Fig. 1d, 1e) and for each taxon separately were spatially predicted by combining the predicted HHR rasters and the derived raster of the number of rural hunters across Amazonia. All metrics and spatial modelling procedures are described in Methods. Our estimates are presented as percentages and means \pm standard deviation (0.10–0.90 quantiles).

Line 130: Why not use per capita harvest?

Harvest Productivity (HP) is the potential number of individuals and biomass that can be harvested per hunter across Amazonia. We also used a similar term that suggested by the reviewer to per capita to refer to available wild meat per rural inhabitant.

Figure 1. Perhaps use a different color legend for each panel since the ranges are quite different.

Done. We appreciate your suggestion.

Figure 2. Appreciate the animal drawing from an indigenous artist and author!

Thank you very much. You have a keen sense of figures, and we really appreciate your suggestions. Please examine the new versions of the figures closely and let us know what you think.

Figure 3. For better visual effect on relative HP, I would suggest using a standardized color scale across all species. The point here is to offer cross-taxon comparison. While the present illustration reflects well on the spatial distribution of HP for individual species, it is not ideal for cross-species comparisons.

We agree with the reviewer. We also prefer to show the cross-comparison between rates, which we were already aware of this. However, when we made the figure with a single scale, we noticed that species with a very large HHR biomass, such as peccaries and tapirs, fully overshadowed species with a relatively low HHR biomass, such as toucans. We, therefore, opted to leave each animal with its own scale.

Lines 266-267. Is there any change in terms of amounts eaten per person observed over the course of 60 years or a lifespan and gender?

265 Amazonian population. We acknowledge, however, that our estimates of wild meat consumption

266 reflect more the availability of hunted meat rather than the actual amounts eaten per person, which vary

267 widely from 3.8 g to 668.6 g per day depending on the region

There are probably differences in consumption between these groups, although we estimated the average for the whole population. However, we have taken into account the different daily energy and nutrient requirements for each age group, sex and reproductive status

Lines 282-284. Is the point that even hyperdominant species are not spared from the climate change impact a major issue? Or is this quite unlikely considering that they are hyperdominant because they are adaptable to environmental changes, including deforestation and climate change?

282 conserved forests (>70% forest cover) (Extended Data Table 7). Recent climatic changes causing more

283 floods, droughts and large-scale fires also threaten hyperdominant species, thus impacting the

284 availability of these species to hunt³⁷.

In the first version of the manuscript, we had used the term hyperdominance to refer to taxa that are hunted in greater numbers. This differs from the original concept of hyperdominance to refer to species that are abundant in the ecosystem. To avoid confusion, we have removed the term in this version.

Lines 290-291. The bit on cultural erosion impact is not that obvious here.

290 deforestation and over-harvesting³⁸. This combination ultimately results in biocultural erosion,

291 influencing hunting practices, hunted species composition and food systems.

We thank the reviewer for this observation. We removed the term cultural erosion now.

New text (Lines 245-261):

*Available wild meat per rural inhabitant was significantly lower in communities with (1) higher numbers of rural inhabitants, (2) closer proximity to urban centres, and (3) greater deforestation levels (Extended Data Fig. 13). In these more degraded areas, our TSOP spatial estimates indicate a shift in hunting patterns, with ecological generalists such as the nine-banded armadillo (*Dasypus novemcinctus*), capybara (*Hydrochoerus hydrochaeris*), guans (*Cracidae*, *Penelope spp.*), and pigeons (*Columbidae*) becoming proportionally more hunted than in better-conserved forests (Extended Data Table 8). In contrast, large atelid primates such as the woolly, spider and howler monkeys, which are quite vulnerable to the synergistic effects of deforestation and overhunting, are much less hunted in degraded areas.*

Large-scale agriculture, land grabbing, logging, mining, and infrastructure development have led to deforestation³³, increasingly undermining Amazonian People's reliance on biodiversity. The combined effects of deforestation and wildlife overharvesting have produced simplified animal assemblages, with lower species richness and fewer large-bodied species³⁴. These pressures are further compounded by recent climatic changes, including more frequent floods, droughts, and large-scale fires³³, all of which threaten both habitats and many key hunted taxa³⁵. We recommend that further studies should investigate the future impacts of deforestation, wildfire, and climate change on animal populations and on the supply of wild meat.

Lines 294-299. This is such an important point, and a huge paradox. Obviously, the agricultural and ranching impacts, if modeled spatially, could be really useful for future management and planning, and impact assessment.

294 Paradoxically, although beef is rarely consumed in most of Amazonia due to its high cost and logistical

295 constraints, it significantly contributes to forest loss.

For the scope of this work, we deal with these issues as follows:

New text (Lines 267-275):

Due to its high cost and the significant logistical challenges of producing, transporting, and storing in remote areas, beef is rarely consumed across most of Amazonia³⁷. Paradoxically, cattle ranching remains the leading driver of deforestation in the region, contributing to the loss of approximately 0.63M km² of forest since 1978, primarily to supply domestic meat markets³⁹. Replacing the estimated edible wild meat production of 0.37 M t with beef, based on current cattle ranching yields in Amazonian traditional pastures (0.2–0.8 kg ha⁻¹ day⁻¹)⁴⁰, would require converting 7,602–63,798 km² of forest into pasture. This conversion would emit $0.3 \times 10^9 - 3.5 \times 10^9$ tonnes of CO₂, equivalent to up to 10% of global annual emissions. Despite this significant environmental cost, domestic meat production still fails to ensure equitable access for rural peoples in remote areas.

Lines 308-311. Are these from Goal 15 the only targets? How about other goals such as health, equity etc?

We also provided a new figure (Extended Data Figure 14) summarising the contribution of traditional wild meat food systems in advancing the Sustainable Development Goals (SDGs).

Line 436. Table X?

Corrected.

Lines 438-444. Re: ATH dataset. How do we know the hunted taxa are correctly identified? Surely, some primary/secondary data are more accurate than others. Do the authors take data quality and accuracy into account during the modeling?

Indeed, we acknowledge that taxa identification is a challenge, particularly considering that data was collected by so many different people, part of different monitoring initiatives, with different degrees of validation and recording methodologies. One shared characteristic of all initiatives was that recording was made by community members that had strong relationships and knowledge of hunting practices and prey species. So, the primary data was subject to local monitors and hunters identification of taxa. Our long-term collaboration shows that hunters have a very consistent knowledge on the identification of the most hunted species. Some initiatives had external taxa identification validation conducted by biologists. Usually, common names were recorded in Indigenous, Latin or English languages. The database stored the original recordings. Also, in different fields, the database standardized all common names to Latin language and English species names.

To avoid inconsistencies and subjectivity, we have tentatively assumed that the identity of the taxa hunted in a given locality refers to what the geographical distribution polygons of the fauna provided by the IUCN determine in that locality.

New text (Lines 545-563):

Taxonomic reclassification and geographic distribution of hunted animal taxa

We extracted the list of mammal, bird, reptile and amphibian species for the 647 localities from the IUCN spatial database²², assuming that the taxonomic identity of all recorded hunted species aligns with the taxonomic and geographic distribution currently recognized by the IUCN22. We then reviewed all 1,789 original taxa from the APTH dataset to correct potential misclassifications, including outdated taxonomy, vernacular identifications, misidentifications, or typographical errors. Taxonomic entries were updated to the most refined, species-specific refined classifications, with taxa categorized into species wherever possible.

*Following this review, we identified 438 species, 51 higher taxa and an 'undetermined' category, resulting in at least 490 distinct species (Extended Data Table 2). The 'undetermined' category includes unidentified species or aggregations for which TSOP could not be reliably calculated. Higher taxa represent broader taxonomic groupings retained from the original sources, such as *Mazama* spp., which includes lower taxa like *M. americana*, *M. nemorivaga* and *M. gouazoubira*, already listed separately in the APTH dataset.*

All allopatric species were aggregated at the genus level to improve analytical coherence, and taxa with very low sample sizes were grouped into broader categories at the family, order, or subclass level. This classification process resulted in 173 analytically focused hunted taxa, plus the 'undetermined' category, which were used in TSOP and related analyses (Extended Data Tables 1,2).

Line 448. Where can we find the documentation of the hunting monitoring schemes? Can we use this to evaluate the quality of the data?

We appreciate the interest in the monitoring schemes that produced the primary data used in this study. These schemes have been described in several publications and reports that can be easily accessed. We understand that the quality of data can be evaluated to some extent by analysing the documentation on the monitoring schemes. For example, aspects such as the purpose and design of the schemes, selection of people involved, capacity building processes, sampling schemes, analysis data at the local scale, interpretation and decisions to respond to communities interests, data validation, and involvement of external collaborating academic researchers are likely to influence the quality of hunting data. Please see Extended Data Table 9 for monitoring details and corresponding references.

Line 468-469. Is the estimation dependent on the time of the survey? This is similar to the key issue highlighted above.

467 body mass (Extended Data Table 1). Additionally, we gathered from the literature 2,027 observations

468 on the densities of 329 hunted animal species, which represent 130 of the 172 hunted taxa. For these

469 130 taxa, we estimated the average number of individuals found per 100 km² (Extended Data Table 1).

We compiled data from the literature on the density of species representing the hunted taxa and estimated the density using the following method:

New text (Lines 564-573):

The Taxon-Specific Average Body Mass and Density

We compiled 4,019 observations on body mass for 477 animal species, representing all 173 hunted taxa. These data were sourced directly from the APTH dataset, from primary data and from the scientific literature. Based on these records, we estimated the average body mass for each of the 173 taxa (Extended Data Fig. 10; Extended Data Table 1). Additionally, we collected from published sources a total of 2,024 observations on population density for 330 hunted animal species, representing 139 of the 173 hunted taxa. For these 139 taxa, we estimated the average number of individuals per 100 km² (Extended Data Fig. 14; Extended Data Table 1). Details on the estimation procedures for both body mass and population density are provided in the Metrics Estimation section of the Methods.

New text (Lines 800-806):

Metrics estimation

To enhance the accuracy of our metrics (e.g., Taxon-Specific Body Mass, Taxon-Specific Density, Proportion of Hunters to Consumers, Proportion of Edible Wild Meat to Undressed Biomass and Wild Meat Nutritional Composition), we run each metric 1000 times, drawing each estimated quantity from a Normal statistical distribution defined by its corresponding mean and standard error. After obtaining 1000 values for each metric, we took the mean, standard deviation and 90% quantiles of these 1000 values to produce a 90% confidence interval.

Line 513. Again, are these time dependent or independent to the survey? They do not seem so but perhaps the authors can explain.

While the HHR metric — animals hunted per hunter per day — provides a degree of standardization across studies (reflecting study effort rather than individual hunting effort), we took further steps to minimise potential bias. In this revised version of the manuscript, we introduced an additional variable into the model to explicitly account for the collection effort associated with each hunting monitoring initiative, assuming that the duration of the study might affect its accuracy in reflecting the hunting activities.

Thus, we included the following description in the methods:

New text (Lines 624-627):

Hunting recording time span

A variable that controls the effort to record hunting in each study, that is, the time range in days in which hunted animals were recorded. This metric was only used to model HHR. We included this variable since we assumed that different time spans of hunting surveys could have different accuracies for data on the animals hunted. We obtained measures of the hunting recording time span from the APTH dataset.

New text (Lines 103-105):

For HHR modelling, we included a variable to account for the hunting registration effort, specifically the number of days over which hunted animals were recorded.

This variable was among the most important in our random forest models for predicting HHR, improving our ability to better assess the effects of variables on HHR.

Lines 557-562. I am unsure if the use of data from 2005, which is the averaged year of ATH dataset, is the most rigorous approach. The variation from the time dependent survey is an important issue, which have not been addressed directly.

We thank the reviewer for pointing out this issue and we agree that the analysis should consider temporal variation in anthropogenic and environmental factors for each data point. This was an issue that concerned us from the start of the analysis.

In the first version we had dealt with it using the average or median of the variables for the whole period: *“Whenever a spatial variable had multiple temporal rasters, we built one with the average or median values. For instance, we obtained a single raster for both Annual NPP and GPP from the median of 22 years. In relation to the Proportion of Forest Cover variable, we used the annual mapping of 2005 land cover, since 2005 represents the averaged year of our ATH dataset. These techniques were aimed at minimising errors caused by temporal variation in spatial variables during the significant time window covered by our ATH dataset.”*

However, in this new version, we strived to obtain more time-specific information for all variables. Given the lack of data for some of them, and the high computational cost, we re-run the model using multiple averages for various specific periods considering the closest data available for each covariate in relation to each hunting study. We highly appreciate this suggestion as our models were greatly improved. We now added a new section in the Methods:

New text (Lines 638-652):

Temporal variation of spatial variables

None of the digital spatial variables fully cover the 1965–2024 period of the APTH dataset, preventing a year-by-year evaluation of their spatial and temporal effects on HHR and TSPO. To address this, we matched HHR and TSPO records to the closest available values of spatial variables in years where spatio-temporal data were available. This approach was feasible for modeling the effects of EVI, NPP, and GPP on HHR and TSPO by incorporating measurements at five-year intervals. We obtained Proportion of Habitat Loss data for 1985, 1990, 1995, 2000, 2005, 2010, 2015, and 2020. For EVI, NPP, and GPP, data were available from 2000 onwards at the same five-year intervals. HHR and TSPO values from 1965 to 1987 were assigned the 1985 habitat loss data; from 1988 to 1992, the 1990 data; from 1993 to 1998, the 1995 data, and so on, with post-2018 records assigned the 2020 data. A similar process was applied to EVI, NPP, and GPP, though for a shorter period: records from 1965 to 2002 were matched to the 2000 values, continuing at five-year intervals up to 2020. Despite the APTH dataset spanning 1965–2024, most hunting studies occurred around 2006±9 years (90% quantiles: 1995–2017), aligning well with the available temporal coverage of the spatial variables.

We also included cautious remarks in the main text regarding the lack of data for some variables:

New text (Lines 141-149):

While our findings provide robust insights, several limitations should be considered when interpreting these results. Although hunting data span from 1965 to 2024, spatial covariates are not uniformly available across the entire period. To partially address this, we matched HHR and TSOP records to the closest corresponding time period (see Methods). We also incorporated a variable to account for variation in recording effort across hunting studies (see Methods), which proved to be an important predictor in our HHR models. Despite potential limitations in capturing fine-scale local variability, the breadth and diversity of our dataset, combined with consistent spatiotemporal covariates, provide a strong foundation for a comprehensive assessment of wildlife harvesting dynamics throughout Amazonia.

Referee #6

General Comments

As noted in the decision letter, this referee provided comments to the editor on the ethical considerations of this study. These included noting that data collected about Indigenous and local people needs to be done with their free, prior and informed consent. This means that you need to include a description of how relevant Indigenous perspectives or oversight guided what research questions were posed, how data was analysed, and if (and how) you considered the ethical, cultural and local contextual issues required.

The ‘extended Data Table 12. Primary data collection and ethical procedure’ is useful background material as is the data availability statement, inclusion and ethics, ethical approval for primary data collection and FPIC procedures (line 800-852)

This means that extended data Table 12 (lines 800-852) would need to include an extra column that confirms human research ethical approvals from relevant institutions also cover the re-use of data set for the purpose of this study. Another column also needs to be added to confirm that conditions underpinning the Amazon Traditional Hunting (ATH) dataset have also been met (see line 800-814).

You also need to state where this ATH is housed and who owns this data. The procedures for reviewing a ‘reasonable’ request (line 804-81) for data sets are sound but our referee was concerned that the authors can make this judgement (line 803) rather than an Indigenous data governance committee."

Finally, we ask that you clarify how reasonable requests for ATH datasets are reviewed and authorised.

The first aspect raised by this reviewer concerns the need for free, prior, and informed consent (FPIC) from Indigenous and Traditional Peoples, specifically regarding collecting data about their ways of life. The other aspect concerns considering Indigenous and Traditional perspectives in defining the research (guiding questions, analysis methods, and ethical sensitivity regarding different cultures and local contexts).

The data collection process for the primary data used in this study certainly meets these requirements and goes beyond them. Formally, as part of the academic bureaucracy, all hunting monitoring initiatives that collected primary data were subject to evaluation by ethics boards or similar bodies, which deemed the data collection consistent with the rights of the Indigenous and Traditional Peoples in accordance with national laws and universities’ norms. However, we believe these bodies, while legitimate to ensure academic and governmental responsibilities, might not be sufficient to assess whether a given research project is ethically aligned with a specific Indigenous and Traditional Peoples. Indeed, Indigenous and Traditional Peoples are often absent from such boards.

Therefore, we emphasize that all primary data were collected from participatory or community-based monitoring initiatives, often long-term and integrated into territorial and natural resource management programs or training processes. These initiatives were built

and conducted by NGOs or government agencies in collaboration with Indigenous and Traditional Peoples and their associations based on their demands to address important local challenges. In most cases, representatives of these Indigenous and Traditional communities, particularly leaders indicated by the community members, participated in various stages of the design and improvement of these initiatives. Many of these wildlife management and monitoring initiatives, and included in the primary data, were demanded by Indigenous and Traditional leaders.

This means that the hunting monitoring initiatives that provided data for the database are located in widely dispersed regions across the Amazon in highly variable periods, from the early 1990s to the present. These initiatives were designed and implemented to address local issues from each village, community, or region, aiming to improve the quality of life of Indigenous and Traditional Peoples, conserve their wildlife, enhance their food security, and preserve their livelihoods, while respecting their decisions. Because of this characteristic, Indigenous and Traditional Peoples and their collaborators have primarily used the data at the local scale, to understand local wildlife use and support the management of their territories and natural resources. Each working group, within its area of activity, has provided feedback on the results at the local level; this activity was already undertaken by each initiative prior to the start of this study.

Below are some examples of processes and previous publications on the initiatives included in the current research that illustrate how hunting data collection and local peoples' participation are ethically sound:

- In 2008, one of the first articles in the Brazilian Amazon on hunting management and collaboration between Indigenous Peoples and scientists, with Indigenous authorship, was published. This was the first analysis of hunting data collected through a monitoring initiative developed in collaboration between indigenous peoples in Acre, the association representing them, the supporting NGO, and associated researchers (Constantino et al. 2008). In 2012, another book chapter using hunting data from other Indigenous lands in Acre followed the same principles (Constantino et al. 2012). It is crucial to highlight that this monitoring initiative was built within a context of collaboration spanning over four decades between Indigenous peoples and an NGO to train Indigenous social actors to improve management practices in their formally recognized territories. In this context, hunting data were consistently used by Indigenous peoples—often without external support—to make decisions about wildlife and territory management, as discussed in another book chapter (Constantino 2020). Three hunting monitoring initiatives that contributed data to the database used in this article were analyzed for their role in empowering Indigenous peoples and traditional communities through their involvement in biodiversity monitoring initiatives (Constantino et al. 2012). The authors of this article, together with the Brazilian federal government, organized an international seminar in 2014 with broad participation from Indigenous and traditional community representatives from several countries (Constantino et al. 2015a), leading to the development of the Manaus Charter, which outlines principles for participatory monitoring and management of natural resources (Constantino et al. 2015b). These principles were adopted to develop a national government program in Brazil for participatory biodiversity monitoring (Programa Monitora), currently implemented in over 150 protected areas in Brazil (Constantino et al. 2019). Hunting data from other communities involved in this national program are integrated into the database used in this article. Additionally, the contribution of some authors to the debate on Indigenous rights, knowledge, and data

ownership is reflected in proposals for the involvement of non-academics in international forums such as the Convention on Biological Diversity (CBD) (Danielsen et al. 2024).

- In the Peruvian Amazon, fish and wild meat have been traditional sources of animal protein for the rural population and have been traded in urban markets (Bodmer and Pezo, 2001; Mayor et al., 2021). Following the implementation of CITES in 1974, different Latin American countries adopted different approaches to mammal hunting in the Amazon. Peru, through laws enacted in 1976 (no. 21.147), 2000 (no. 27.308), and 2011 (no. 29.763), allowed subsistence hunting of certain wild species by indigenous hunters. For over three decades in the Peruvian Amazon, community-based wildlife management systems have been successfully developed through research and community engagement, resulting in examples of sustainable hunting of wild mammal species (Silvius et al., 2004; Bodmer et al., 2008; Townsend, 2000). This has led to an expansion in the number of community-managed areas, in the form of Indigenous territories, Community-Managed Protected Areas, and Community-Co-managed Protected Areas (Mayor et al., 2020). Consequently, regional and national governments have strongly supported the creation of Community-Managed Areas, where Indigenous Peoples participate in sustainable use and management. Beginning in 2007, natural resource management plans were implemented with local communities, which include river turtle breeding, bushmeat hunting, palm fruit harvesting, and fishing (Pérez-Pena et al., 2022; SERNANP 2011; SERNANP 2015). The Tamshiyacu-Tahuayo, Ampiyacu-Apayacu, Alto Nanay-Pintuyacu-Chambira, and Maijuna Regional Community Conservation Areas, as well as the Pacaya-Samiria, Pucacuro, Allpahuayo-Mishana, and Matsés National Reserves, co-managed by communities, have implemented management plans during this period (Bodmer et al., 2023). Community-based wildlife management is more successful when management plans are adapted to the cultural and socioeconomic characteristics of local communities (Manfredo et al. 2008). Thus, the expansion of these co-managed areas covers more than 5.4 million hectares and is home to the indigenous Achuar, Huitoto, Kichwa, Kukama, Kukamilla, Maijuna, Matsés, and Yagua cultures (Mayor et al., 2020). For instance, wildlife management plans in Pucacuro National Reserve and the Ampiyacu-Apayacu Regional Conservation Area in the Peruvian Amazon have been approved that legally permit the sale of wild meat in urban markets for the first time since the ban on the urban wild meat trade in 1976 (Mayor et al., 2021). Currently, each engaged family receives US\$ 484,18 for every 100 kg of wild meat sold. La Reserva Nacional Pucacuro tiene al 2024, 215 beneficiarios directos, los cuales representan los cazadores inscritos en el padrón de socios, 57 cuentan con Contrato de aprovechamiento y 158 con Acuerdo de Actividad Menor. Los contratos esta representado por 2 asociaciones y acuerdos, por 5 comunidades indígenas kichwas. En una superficie de aprovechamiento de 136,045.3 ha, en el año 2024, se extrajo de forma sostenible 1,382 kg de carne de monte de las especies Tayassu pecari, Cuniculus paca, Pecari tajacu sajino y Mazama americana, con 577.5 kg, 468.5 kg, 311 kg y 25 kg, respectivamente. El volumen promedio de carne aprovechada por cazador en cada año varió entre 10.0 y 42.0kg. En total, la cantidad de carne vendida asciende a S/37,338.59 (USD 10191,17). El beneficio anual promedio por cazador también varió desde los S/157.3 (USD/42.9) y los S/1469.1 (USD/401) por familia, dependiendo de la comunidad indígena. Estos números suponen entre el 1.7 y 6.6% de la Canasta Básica Familiar anual (INEI, 2021)
- The collaboration with the Paumari people was initially built to support their territory's management. The design of the wildlife and hunting monitoring project has assessed the state of conservation and the sustainability of hunting from a demographic and spatial

perspective. The data from this project has not only underpinned the Paumari people's political demand for the re-delimitation of the lands, which are recognized as being too small to guarantee the sustainability of the resources but has also been fundamental in strengthening this request to the Brazilian government (Antunes et al. 2015; Antunes 2023, Antunes 2024). Currently, the anthropological and environmental study to redraw the boundaries of the Paumari Indigenous Lands is in the planning phase. An oral and written agreement to use this data to carry out research into hunting and wildlife has been made between the Paumari people and the coordinator (APA). The main Paumari researchers for developing this project authored this article.

- In the Piagaçu-Purus Sustainable Development Reserve, wildlife use monitoring was carried out with volunteer traditional hunters from 10 regional riverine communities. First, the researchers responsible for conducting the study called meetings in each of these communities with local authorities and hunters to present the project proposal and ask for their collective consent. In those communities that agreed to the study, the researchers then consulted each resident who self-recognized as a frequent (subsistence) hunter if they were interested in recording data on their hunting events on pre-established forms, which were easy to use even for people with low reading levels, voluntarily and anonymously. Over the course of x years, the researchers accompanied the volunteers during regular household visits (no more than 2 months apart), maintaining a relationship of trust with them. At the end of each year of monitoring, the data collected was analyzed by the researchers and presented to the residents of each community, aiming to stimulate discussions about hunting management in the region and improve the system based on feedback from the hunters themselves (Vieira et al. 2019).
- In the Juruá River, particularly within the Uacari Sustainable-use reserve, the hunting monitoring was performed through the Biodiversity and Natural Resource Use Monitoring Program for Protected Areas (PROBUC). This program was established by the government of the State of Amazonas with the aim of strengthening the management of Protected Areas in the region, promoting active participation of local communities in the collection and analysis of environmental data. This program stands out for its co-production approach, where local leaders played a crucial role in monitoring and decision-making regarding territorial conservation. This program has proven to be an effective platform for exchanging traditional knowledge and scientific methodologies, fostering the integration of local knowledge with scientific methods. Approved by the deliberative council of the reserves, the program ensured that communities were formally involved in decisions about the use and protection of natural resources, ensuring more inclusive, effective, and sustainable management of protected areas in Amazonas.
- Monitoring in the Iriri and Riozinho do Anfrísio Extractive Reserves was conducted under the Brazilian Biodiversity Monitoring Program (Programa Monitora). First, we held meetings in key communities along major rivers to consult with locals about their interest in the monitoring initiative and determine which targets they would like to monitor. All communities gave their verbal informed consent, indicating fishing and hunting as priorities for monitoring. Subsequently, we trained local monitors nominated by the communities to collect hunting and fishing data regularly. Monitors signed a consent form and were provided with financial support and fuel to reach localities where questionnaires were deployed. Throughout the project, we held yearly meetings in key communities to discuss and interpret the program results collectively (De Paula et al. 2022, Ponce-Martins et al. 2022).

- In the specific case of the hunting and game consumption data from the indigenous communities at the Xingu Indigenous Territory in Mato Grosso State, and from the Tenharim Indigenous Lands in the Amazonas State, the information was part of a study that aimed to attest the link to the land and natural resources, and a mandatory information to be collected during the official process of formal recognition of Indigenous Lands in Brazil. Additionally, previous meetings were held to explain the purposes of the study, and each interviewee was asked and gave their verbal consent.

Although the principles of engagement developed through these initiatives have been adopted at larger scales and in broader contexts, this article represents the first attempt to jointly analyze the hunting data from various monitoring initiatives at the Amazon scale. Hunting data were only examined for years from the perspective of local issues. It is essential to emphasize that 19 representatives of Indigenous and Traditional Peoples from the long-term collaborative monitoring initiatives whose data is included in the database used in this article are co-authors of this article, one of them being a representative of the National Council of Extractivists (CNS) and one of the Coordination of Indigenous Organizations of the Brazilian Amazon (Coiab). Indeed, issues may differ from those contextualized locally at this large scale. Patterns observed at this continental scale might not reflect particular cultural specificities that are relevant locally. Also, it is important to notice that there is no traditional knowledge associated with hunting practices, animals, or their cosmological relations of any specific ethnic group that participated in the individual monitoring initiatives discussed in this paper. Nevertheless, we believe that the knowledge generated at this continental scale addresses some of the large-scale issues Indigenous and Traditional Peoples discuss at many arenas, such as food security, nutrition, and territorial and natural resource management rights, as each monitoring initiative we included in the manuscript is held through engaged long term relationship with Indigenous and Traditional Peoples, and also ensured by the Indigenous and Traditional Peoples representatives authoring this paper.

The integrated multiregional and diachronic dimension of this study provides a holistic interpretation of hunting and its benefits for food security. These results have significant regional and national significance. Thus, the feedback loop is established at the level of Indigenous, Traditional, and public institutions. The direct results of this study will be delivered based on the design of reports discussed in meetings with Indigenous and Traditional Peoples federations and the responsible sectoral ministries in each participating country. Additionally, the study will make it possible to share results with sectoral authorities, which will support decision-making.

We understand the concerns raised by Reviewer #6, and we have been working most of our professional lives to support Indigenous and Traditional Peoples. This publication is just another piece of knowledge contributing to the long-standing collaboration for the autonomy of Indigenous and Traditional Peoples in the Amazon. Thus, the hunting data in the database, resulting from long-term participatory or community-based monitoring initiatives used in this article, differ significantly from traditional research conducted exclusively by non-Indigenous academics or external to local communities. Given the context in which these data were generated, and the article has been written, we believe it is difficult to question the ethical alignment and legitimacy of the relevance of these data to the interests of Indigenous and Traditional Peoples. External assessments, the way they are currently structured on this matter, will always fall short of the ethical importance of the work and the commitment of these authors to Indigenous and Traditional Peoples. So, the

reviewer's suggestion to create two additional columns in Table 12 (now Table 9) would not add significant information.

On the other hand, we appreciate the suggestion to improve the data-sharing decision-making process. So, we are establishing a committee composed of representatives of the Indigenous and Traditional Peoples, regional organizations, and the non-indigenous and non-traditional peoples collaborating with them. These organizations are: **Coordination of Indigenous Organizations of the Brazilian Amazon (COIAB)** (representing the Indigenous Peoples of Brazilian Amazonia), **National Council of Extractivist Populations (CNS)** (representing the Traditional non-indigenous peoples of Brazilian Amazonia), and RedeFauna (Rede de Pesquisa em Diversidade, Conservação e Uso da Fauna na Amazônia). This board will meet upon request to share the data.

Finally, we submitted the manuscript to organizations with broad Indigenous and traditional community representation in the Amazon region to evaluate whether the article's scope and methods ensured that the studied communities had full autonomy to provide or withhold Free, Prior, and Informed Consent (FPIC) regarding any project or research affecting their territories, rights, and ways of life—while respecting their culture, self-determination, and equity.

So far, we have obtained the letter of support from the CNS (see document attached below) and are waiting for the evaluation from COIAB. COIAB itself liaises with AIDSESEP and FOAG, the Indigenous representation from Peru and Guyana, respectively, to evaluate the manuscript's methods and consider joining the evaluation committee.

Conselho Nacional das Populações Extrativistas - CNS

Brasília/DF, 12 de maio de 2025.

Dear Editor and Reviewers of Nature,

The National Council of Extractivist Populations (CNS), which represents traditional extractivist populations in Brazil, has the mission of representing, organizing and guaranteeing the territories of collective use of traditional extractivist populations, articulating, proposing and demanding policies and promoting socio-economic, environmental and cultural sustainability for present and future generations, based on traditional extractivist populations, in accordance with Decree 6.040/2007, which recognizes the rights of traditional peoples and communities in Brazil, with a focus on working with grassroots organizations in the Brazilian Amazon, expresses its support for the publication of the article entitled "Healthy forests safeguard traditional wild meat food systems in Amazonia", which analyses data on hunting practiced by indigenous peoples and traditional communities in Amazonia.

Studies like this, which highlight the importance of sustainable wildlife use and management for our peoples, align fully with our efforts to preserve the traditional relationship we maintain with nature. The findings of this research have the potential to strengthen advocacy at both national and Amazonian scales, underscoring the need to protect our territories to ensure the continuity of our sustainable traditional practices—practices that are vital for our peoples' food security and nutritional well-being.

Furthermore, we commit to collaborating with the article's authors and other representative organizations of Indigenous peoples and traditional communities to establish an Evaluation Committee. This committee will assess potential requests from external researchers for access to the hunting dataset compiled in the study's Database, ensuring any data sharing is conducted ethically and respectfully.

We appreciate the opportunity to contribute to this scientific dialogue and reaffirm our commitment to advancing knowledge that values Indigenous and traditional practices.

Sincerely,

Júlio Barbosa de Aquino
Presidente -CNS

National Council of Extractivist Populations (CNS)

SNQ 111, BLOCO 1, BRASÍLIA/DF
CEP: 70.754-090
E - MAIL: cns.secretarianacional@gmail.com

New text (Lines 919-964):

Data Availability Statement

The Amazonian Peoples Traditional Hunting dataset (APTH) built and analyzed in this study is not publicly available due to sensitivity and privacy concerns related to hunters and their communities. However, the authors can make the data available upon request for research purposes only, where the researcher provides a detailed written document outlining the study's objectives and a signed letter of commitment to the ethical and exclusive use of the data for the specified research. Each proposal must be explicitly approved by all contributors to the dataset and Indigenous and Traditional representatives, who will review the request for raw data, digital variables, or other information used in the paper. Conditions on re-use include acknowledgment of the data source and adherence to ethical standards outlined in the agreement. If a contributor or Indigenous or Traditional representative chooses not to share their data for a particular study, those specific data will be removed without impeding the overall research. Contributors retain full autonomy over using their own data and can withdraw from the agreement at any time without prior notice. Requests should be sent to aapardalis@gmail.com.

Research collaboration and ethics

The community-based hunting monitoring initiatives that contribute primary data to the Amazonian Peoples Traditional Hunting (APTH) dataset were developed and implemented through close partnerships with Indigenous and Traditional Peoples. These initiatives should not be seen as conventional academic research, where external researchers extract data on biodiversity use. Instead, they are grounded in sociopolitical realities relevant to Indigenous and Traditional communities and are designed to strengthen livelihoods, empower communities, support wildlife conservation and management, enhance food security, and protect cultural practices—always with respect for their priorities and autonomy. The data collected directly informs community-led decisions on sustainable wildlife and territorial management, with meaningful community engagement throughout the monitoring cycle. Results are transparently shared at the community level and used to guide practical actions on the ground. Each project is tailored to the specific needs of the communities involved, ensuring that traditional knowledge is respected, safeguarded, and valued. Our collaborative approach includes training Indigenous and Traditional hunters and researchers, fostering long-term capacity building and education. Wildlife monitoring programs are conducted under government regulation, ensuring ethical data handling and confidentiality. All research activities are formally approved by Indigenous and Traditional communities, as well as by relevant academic or governmental institutions overseeing Indigenous Lands, parks, and extractive reserves.

Data-sharing agreements were established among communities, researchers, and technicians, enabling informed local decision-making and advancing research on wildlife use, management, and conservation in Amazonia. Free, Prior, and Informed Consent (FPIC) was obtained—either orally or in writing—from all communities participating in those initiatives, ensuring ethical engagement and respect for rights, welfare, and autonomy. While some early initiatives predated formal non-indigenous and local ethics committees (e.g., before the Nagoya Protocol, 2010), agreements were always culturally adapted, ranging from oral consensus to written contracts detailing research objectives, participant rights, and data use (Extended Data Table 9).

Six independent ethics committees reviewed and approved all primary data methods, with the 12 contributing initiatives receiving clearance from institutional review boards across

Brazilian, Peruvian, and French Guianese Amazonia (Extended Data Table 9). These approvals, secured through universities and research institutions, guaranteed compliance with international ethical standards for research involving Indigenous and Traditional Peoples.

References

Antunes AP, Muhlen EMV, Rossoni FC, Ventincinque EM. Monitoramento de fauna Paumari nas Terras Indígenas Paumari. (Operação Amazônia Nativa, Cuiabá, 2015).

Antunes AP. Manejo Territorial de Fauna em Terras Indígenas da Amazônia. Relatório Final. (PCI/INPA/MCTI, 2023)

Antunes AP. Wildlife participatory management in two indigenous territories of Amazonia. Final Report. (National Geographic Society, 2024).

Bodmer RE, Puertas P, Fang T (2008) Co-managing wildlife in the Amazon and the salvation of the Pacaya-Samiria National Reserve in Peru. In: Manfredo M, Vaske J, Brown P, Decker D, Duke E (eds) Wildlife and society. The science of human dimensions. Island Press, Washington, DC, pp 104–116

Bodmer, R. E., & Pezo, E. (2001). Rural development and sustainable wildlife use in the tropics. *Conservation Biology*, 15, 1163–1170.

Bodmer, R.E., Pablo Puertas, Tula Fang, Miguel Antúnez, Sandro Soplín, Jhonathan Caro, Pedro Pérez, Hani R. El Bizri, Marco Arenas, José Carlos Nieto, Maire Kirkland, and Pedro Mayor (2023). Management of Subsistence Hunting of Mammals in Amazonia: A Case Study in Loreto, Peru. In: Spironello, W.R., Barnett, A.A., Lynch, J.W., Bobrowiec, P.E.D., Boyle, S.A. (eds) *Amazonian Mammals*. Springer, Cham. https://doi.org/10.1007/978-3-031-43071-8_10

Constantino PAL, Tavares RA, Kaxinawa, JA, Kaxinawa FM, Kaxinawa E, Kaxinawa AS, 2012. Monitoramento e mapeamento participativo da caça na Terra Indígena Kaxinawá da Praia do Carapanã (Acre). In, *Conservação da biodiversidade com SIG* Chapter: Monitoramento e Mapeamento Participativo na Terra Indígena Kaxinawa da Praia do Carapanã (Acre). Publisher: Oficina de textos Editors: A. Paese, A. Uezu, M.L. Lorini, A. Cunha

Constantino PAL, Benchimol M, Antunes AP (2018). Designing Indigenous Lands in Amazonia: Securing indigenous rights and wildlife conservation through hunting management. *Land Use Policy*, 77, 652-660. <https://doi.org/10.1016/j>.

Constantino PAL, 2020. Challenges of Forest Citizen Involvement in Biodiversity Monitoring in Protected Areas of Brazilian Amazonia. In, *Handbook of Citizen Science in Ecology and Conservation*. Lepczyk CA, Boyle OD, Vargo TLV (Eds) Publisher: University of California Press.

Fang, T.G.; et al. Wild meat trade over the last 45 years in the Peruvian Amazon. *Conserv. Biol.* 2022, 36, e13801.

Manfredo MJ, Vaske JJ, Brown PJ, Decker DJ, Duke EA (2008) *Wildlife and society: the science of human dimensions*. Island Press, Washington, p 350

Mayor P, Álvarez J, Garcia J, Bodmer RE (2020) *Pueblos Indígenas de la Amazonía Peruana*, 2nd edn. CETA/Fundamazonia, Iquitos

Mayor, P.; El Bizri, H.R.; Morcatty, T.Q.; Moya, K.; Bendayán, N.; Solis, N.; Vasconcelos-Neto, C.F.A.; Kirkland, M.; Arevalo, O.; Pérez-Pena, P.; Riveros-Montalván, M.; Tapia del Aguila, C.; Pizarro García, J.; Alvitez, C.B.; Alemán, E.M.L.; Saavedra, E.A.N.; Baca, Y.B. Abundance of the yellow-spotted river turtle *Podocnemis unifilis* in the Pacaya Samiria National Reserve, north of the Peruvian Amazon. *Cienc. Amaz.* 2022, 10, 87–100.

de Paula, M. J., Carvalho, E. A., Lopes, C. K. M., et al. Hunting sustainability within two eastern Amazon Extractive Reserves. *Environmental Conservation*. 49(2), 90-98. (2022). doi:10.1017/S0376892922000145

Ponce-Martins, M., Lopes, C.K.M., de Carvalho-Jr, E.A.R., dos Reis Castro, F.M., de Paula, M.J. and Pezzuti, J.C.B. Assessing the contribution of local experts in monitoring Neotropical vertebrates with camera traps, linear transects and track and sign surveys in the Amazon. *Perspectives in Ecology and Conservation*, 20(4), pp.303-313 (2022).

SERNANP (2015) *Plan Maestro de la Reserva Nacional Pucacuro*. SERNANP, Ministerio del Ambiente, Lima

SERNANP. *Integrated Management Plan for Aquatic Turtles in the Samiria Watershed, Pacaya Samiria National Reserve (PSNR) 2011–2015*; SERNANP: Iquitos, Peru, 2011.

Silvius KM, Bodmer RE, Fragoso JMV (2004) *People in nature: wildlife conservation in South and Central America*. Columbia University Press, New York, p 464

Townsend W (2000) The sustainability of subsistence hunting by the Sirionó Indians of Bolivia. In: Robinson J, Bennett E (eds) *Hunting for sustainability in tropical forests*. Columbia University Press, New York, pp 267–281

Vieira, M.A.R.M., de Castro, F., and Shepard, G. H. 2019. Who sets the rules? Institutional misfits and bricolage in hunting management in Brazil. *Human Ecology*, 47, 369-380.

Referees' comments:

Referee #1 (Remarks to the Author):

Thank you for carefully addressing, in great detail, each of my comments and questions.

I do not have any additional questions.

We sincerely thank Reviewer #1 for its valuable comments and suggestions.

Referee #2 (Remarks to the Author):

HONGO Shun

Research Institute for Humanity and Nature

Thank you for seriously considering my comments on the previous version of the manuscript; I think it is now largely improved by rigorously addressing the temporal variation of the data. I have one question and one suggestion remaining, however.

First, the authors defined the HHR (hunter harvest rates) as “the average number of animals hunted per hunter per day in each locality”. However, I would like to confirm whether HHR from your primary and secondary datasets are calculated to include days when hunters did not engage in hunting in the denominator. If I understand well, HHR of a single locality should be calculated as: (The Total Number (or Biomass) of Animals Observed to be Harvested) / [(Number of Hunters Observed) * (Number of Days Surveyed)]. To accurately estimate the annual harvest rate and available wild meat, the Number of Days Surveyed must include not only the days hunters entered the forest for hunting but also the days when they did not engage in hunting (e.g.,

days hunters stayed home or engaged in other activities) because hunters are unlikely to go hunting everyday. Otherwise, the estimates would be overestimated. I did not find clear explanations to address this question. So, could you add a brief sentence to explain how the authors calculated HHR explicitly? Similarly, I imagine that most secondary data employed ad libitum sampling in terms of the choice of hunters. Consequently, the researchers of the secondary data may have tended to choose “frequent hunters” who engage in hunting relatively frequently. If this is the case, HHR would be overestimated even if it is correctly calculated due to the lack of data on non-frequent hunters. This point may be discussed as a limitation of the study and a future recommendation for producing hunting data.

We sincerely thank Shun Hongo for his valuable comments and suggestions.

We agree with the reviewer, and we have indeed included all surveyed days regardless of whether the hunter went out hunting or not. We have clarified these issues in the text as follows:

New text (Lines 133-143):

“While our findings provide robust insights, some limitations warrant consideration. Several studies lack clear descriptions of hunter selection criteria—whether participants were chosen randomly or focused on primary hunters within each community—and often do not specify whether reported data represent all hunting activities or only a subset. Moreover, while hunting records span from 1965 to 2024, spatial covariates are not consistently available for the entire period. To address this, we matched HHR and TSOP records to the closest corresponding time frame (see Methods) and included a variable to account for variation in recording effort across studies (see Methods), which emerged as an important predictor in our HHR models. Despite potential constraints in capturing fine-scale local variability, the breadth of our dataset and the consistency of spatiotemporal covariates, provide a solid basis for a comprehensive assessment of wildlife harvesting dynamics across Amazonia.”

New text (Lines 733-740):

“After cropping the MARUPIARA Dataset to the geographical boundaries of Amazonia, we obtained georeferenced observations with information on the number of individual animals hunted per hunter per day—Overall Individual Animals Hunter Harvest Rate (HHR). We extracted the values of all spatial variables for the 301 georeferenced localities (both from primary and secondary data) with HHR measures. We determined the Overall Individual Animals HHR for each spatial cell as the total number of animals hunted of all taxa in each locality divided by the sum of hunters accountable for catching those animals during the monitored period in days. This includes days when hunters neither went hunting nor harvested any animals.”

Second, I feel “The fate of wild meat food systems in Amazonia” is too long and speculative. This section mostly discusses the sustainability of the current wild meat hunting system in Amazonia. However, this study estimated only the current state of local hunting but did not examine trade, consumption, or animal abundance. So, this study has limited resources for discussing sustainability. I understand that this

revision is based on the response to Reviewer #1's comment (rebuttal file, pp. 7-8), but personally, I am opposed to this addition. I believe that this addition weakens the arguments of the manuscript. I suggest reducing the section and creating a supplementary discussion by merging some of the discussions in this section with the responses on pp. 7-8.

The topic "The fate of wild meat food systems in Amazonia" (now renamed as "Managing wild meat food systems") concludes the article with a discussion of the practical and political implications of our research. We agree with the observation of Shun Hongo that our results alone do not allow for an assessment of the sustainability of hunting in biological terms— and, indeed, we do not make such a claim at any point in the manuscript. On the contrary, we emphasize—drawing on the broader scientific literature—that the sustainability of hunting depends on a variety of ecological, political, cultural and human pressure factors. We argue that collaborative wildlife management initiatives, which incorporate Indigenous and traditional knowledge and practices, can play a key role in promoting sustainable hunting. Such approaches support both biodiversity conservation and the rights of Amazonian peoples. Given the importance of these issues for wild meat food systems and wildlife management, we have chosen to retain the structure and framing of this topic as presented in the final version of the manuscript.

Referee #3 (Remarks to the Author):

Carla Morsello

This article presents a large-scale analysis of wild meat consumption across the Amazon, drawing on data from 560 rural locations—341 of which are based on primary data. It includes spatial analysis, models consumption patterns, and assesses the nutritional importance of wild meat for rural populations. By combining this dataset with modelling, the article offers a regional perspective on wild meat consumption in the Amazon that is unparalleled, given that most previous studies have been localized. This analysis also helps identify the challenges involved in conserving hunted species in the region.

The authors have appropriately addressed my comments and have notably: highlighted the novelty of the study; improved the description of spatial patterns; enhanced the figures; and clarified the procedures applied to both primary and secondary data. I am therefore satisfied and would like to compliment the authors on this impressive paper. There are a few minor typos I have found below.

Minor points

Line 90: Delete the "the" here: "the" Sustainable Development Goals (SDGs)

Line 189: "targeted" instead of "target"

Line 195-197: Review this phrase: "In contrast, those living in upland terra firme forests, which cover 86% of Amazonia, yield more tinamous, trumpeters, wood quails, guans, and armadillos are more commonly hunted"

Line 583: Unclear: "different local contexts local-scale variations"?

Lines 621-622: I did not understand this phrase: “Although the spatial process of excluding urban areas probably excludes rural or peri-urban inhabitants”. How does excluding urban areas lead to the exclusion of rural inhabitants?

We sincerely thank Carla Morsello for her valuable comments and suggestions. We have addressed all her points.

Referee #4 (Remarks to the Author):

I appreciate the thorough and detailed reply from the authors regarding the ethical considerations raised in my initial review. It is evident that significant thought and effort have been dedicated to addressing these issues.

The response provides extensive clarification on the participatory nature of data collection, the longstanding collaborations with Indigenous and Traditional Peoples, and the ways in which community members have been involved in the design, implementation, and feedback processes of the monitoring initiatives. I particularly welcome the authors’ acknowledgment that formal institutional ethics approvals, while important, may not be sufficient in isolation, and that they have sought to ensure ethical alignment through community involvement and leadership.

I also appreciate the authors’ positive response to the concern regarding data-sharing governance. The establishment of a formal committee including representatives from COIAB, CNS, and RedeFauna to oversee data access is a welcome and important step toward ensuring Indigenous data sovereignty and shared decision-making.

Finally, the proactive submission of the manuscript to Indigenous representative organizations for independent evaluation, along with the supporting letter from CNS and the pending feedback from COIAB and its regional partners, further demonstrates a strong commitment to ethical and culturally sensitive research practices.

In conclusion, I believe the authors have made a sincere and meaningful effort to respond to the concerns raised.

We sincerely thank Reviewer #4 for its valuable comments and suggestions.

We mention in the section “Data and Code Availability Statement” the process of independent evaluation of COIAB and CNS organizations in the text, and we explain in the section “Research collaboration and ethics” the commitment made by both organizations to participate in the Evaluation Committee for any requests to conduct future research. The formal endorsements of both organizations are included in Supplementary *Methods 6, 7*.

New text (Lines 892-897):

“Each research proposal must receive explicit approval from the Evaluation Committee, which is composed of all dataset contributors, along with representatives from COIAB (Coordination of Indigenous Organizations of the Brazilian Amazon), CNS (National Council

of Extractive Populations), and the RedeFauna research network. The Committee is responsible for reviewing all requests for access to raw data, digital variables, or any other information used in this study."

New text (Lines 934-939):

"The primary data collection methods were approved by the main representative organizations of Indigenous Peoples and Traditional communities in the Brazilian Amazon— respectively, the Coordination of Indigenous Organizations of the Brazilian Amazon (COIAB) and the National Council of Extractive Populations (CNS). Both organizations have formally endorsed the content of this article and have committed to participating in the Evaluation Committee for future research utilizing the MARUPIARA Dataset (see Supplementary Methods 6, 7)."

Referee #5 (Remarks to the Author):

I am one of the original reviewers for the manuscript and I am happy to provide further inputs to the revision.

By and large, I commend the authors for providing high quality responses and edits to the issues that I have raised. They have made a serious attempt to improve the manuscript.

However, there remains two issues that I feel need to be directly addressed in the manuscript. They are regarding the following.

1) "What is so interesting about this region is the presence of diverse indigenous cultures and their hunting practices, and target and preferred species. The authors did a good job in highlighting how the Amazonian Peoples have a deep understanding of the Amazonia ecological landscape. But how deforestation may pit tribes (and cultures) against each other in order meet food security, which may indirectly increase conflicts among neighboring areas/tribes in the region? What are the risks of such conflicts under different scenarios?"

The authors responded appropriately to my comment. However, it would benefit the readers if the authors could include their responses into the main text.

2) "It is plausible that the diminishing supply of traditional food may cause shifts in diet which may increase competition and conflicts for shared species (even hyperdominance ones), hence leading to unsustainable offtakes and food system compared to the past. As stated in lines 222-224, the "dynamic interplay has led Amazonian Peoples to develop diverse and sophisticated systems of hunting rules, species preferences, and taboos, resulting in varied hunted assemblages across hundreds of cultures", but this may be potentially disrupted by deforestation and climate change issues with time."

Similarly, I would really appreciate it if the authors could include their responses into the main text. Compared to the responses in the above comment, this is much less developed. If the authors could work on a better response to my comment and include it in the main text, that would really benefit the readers.

Otherwise, this has been a very high quality manuscript and an important topic to work on.

We sincerely thank Reviewer #5 for its valuable comments and suggestions.

Both social conflicts over hunting areas resulting from deforestation and the climate crisis are issues of great relevance not only for hunting management but above all for the territorial management of indigenous and traditional lands as a whole. However, these are specific issues that deserve more in-depth local studies and even more in-depth discussion, which unfortunately do not fit within the scope of our article. In response to Reviewer 5, we chose to only mention the issue of climate change in the first round of revisions. In the current version, we also only mention disputes over hunting grounds, without, however, going into detail about this. Please see below:

New text (Lines 256-264):

“Large-scale agriculture, land grabbing, logging, mining, infrastructure development, and urbanization have led to deforestation³⁴, increasingly undermining Amazonian Peoples’ reliance on biodiversity. The combined effects of deforestation and wildlife overharvesting have produced simplified animal assemblages, with lower species richness and fewer large-bodied species³⁵. These pressures are further compounded by recent climatic changes, including more frequent floods, droughts, and large-scale fires³⁴, all of which threaten both habitats and many key hunted taxa³⁶. We recommend that further studies investigate the future impacts of deforestation, wildfires, and climate change on animal populations, the supply of wild meat, and the territorial dynamics of hunting grounds.”

New text (Lines 315-321):

“Although well-preserved forests combined with strong local governance can sustain harvests through source-sink hunting dynamics⁴⁵, our findings indicate that forest degradation increasingly threatens wild meat productivity, alters hunted species composition and erodes unique traditional food systems. In this context, establishing locally agreed hunting management frameworks may be essential for safeguarding the well-being of Amazonian Peoples, ensuring the long-term sustainability of the wildlife on which they depend, and reducing the risk of social conflict over hunting grounds.”